# Partial Identification of Counterfactual Distributions

## Abstract

This paper investigates the problem of bounding counterfactual queries from a combination of observational data and qualitative assumptions about the underlying data-generating model. These assumptions are usually represented in the form of a causal diagram (Pearl, 1995). We show that all counterfactual distributions (over finite observed variables) in an arbitrary causal diagram could be generated by a special family of structural causal models (SCMs), compatible with the same causal diagram, where unobserved (exogenous) variables are discrete, taking values in a finite domain. This entails a reduction in which the space where the original, arbitrary SCM lives can be mapped to a dual, more well-behaved space where the exogenous variables are discrete, and more easily parametrizable. Using this reduction, we translate the bounding problem in the original space into an equivalent optimization program in the new space. Solving such programs leads to optimal bounds over unknown counterfactuals. Finally, we develop effective Monte Carlo algorithms to approximate these optimal bounds from a finite number of observational data. Our algorithms are validated extensively on synthetic datasets.

## 1 Introduction

This paper studies the problem of inferring counterfactual queries from the combination of non-experimental data (e.g., observational studies) and qualitative assumptions about the data-generating process. These assumptions are represented in the form of a *causal diagram* [32], which is a directed acyclic graph where arrows indicate the potential existence of functional relationships among corresponding variables; some variables are unobserved. This problem arises in diverse fields such as artificial intelligence, statistics, cognitive science, economics, and the health and social sciences. For example, when investigating the gender discrimination in college admission, one may ask "what would the admission outcome be for a female applicant had she been a male?" Such a counterfactual query contains conflicting information: in the real world the applicant is female, in the hypothetical world she was not. Therefore, it is not immediately clear how to design effective experimental procedures for evaluating counterfactuals, let alone how to compute them from observations alone.

The problem of identifying counterfactual distributions from the combination of data and a causal diagram has been studied in the causal inference literature. First, there exist a complete proof system for reasoning about counterfactual queries [19]. While such a system, in principle, is sufficient in evaluating any identifiable counterfactual expression, it lacks a proof guideline which determines the feasibility of such evaluation efficiently. There are algorithms to determine whether a counterfactual distribution is inferrable from all possible controlled experiments [41]. There exist also algorithms for identifying path-specific effects from experimental data [1] and observational data [42].

In practice, however, the combination of quantitative knowledge and observed data does not always permit one to point-identify the target counterfactual queries. Partial identification methods concern with deriving informative bounds over the target counterfactual probability, even when the target

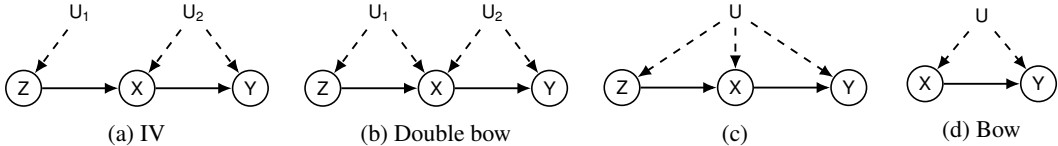

Figure 1: DAGs (a-d) containing a treatment $X$, an outcome $Y$, an ancestor $Z$, and exogenous variables $U$; $Z$ in (a) is also referred to as an instrumental variable.

itself is non-identifiable. Several algorithms have been developed to bound counterfactuals from the combination of observational and experimental data [30, 36, 3, 4, 14, 35, 23, 24, 16, 25, 49].

In this work, we build on the approach introduced by Balke & Pearl in [3], which involves direct discretization of the exogenous domains, also referred to as the principal stratification [17, 34]. Consider the causal diagram of Fig. 1a, where $X, Y, Z$ are binary variables in $\{0, 1\}$; $U$ is an unobserved variable taking values in an arbitrary continuous domain. [3] showed that domains of $U$ could be discretized into 16 equivalent classes without changing the original counterfactual distributions and the graphical structure in Fig. 1a. For instance, despite it being induced by an arbitrary distribution $P^*(u)$ over a continuous domain of the exogenous variable $U$, the observational distribution $P(x, y|z)$ must be reproduced by a generative model of the form $P(x, y|z) = \sum_u P(x|u, z)P(y|x, u)P(u)$, where $P(u)$ is a discrete distribution over a finite exogenous domain $\{1, \ldots, 16\}$.

Using the finite-state representation of unobserved variables, [4] derived tight bounds on treatment effects under the condition of noncompliance in Fig. 1a. [11, 21] applied the parsimony of finite-state representation in a Bayesian framework, to obtain credible intervals for the posterior distribution of causal effects in noncompliance settings. Despite their optimal guarantees, these bounds are only applicable to the specific noncompliance setting in Fig. 1a. For the most general cases, a systematic procedure for bounding counterfactual queries in arbitrary causal diagrams is still missing.

Our goal in this paper is to overcome these challenges. We investigate the expressive power of *discrete structural causal models* (SCMs) [33] where each unobserved variable is drawn from a discrete distribution, takes values in a finite set of states. We show that when inferring about counterfactual distributions (over finite observed variables) in an arbitrary causal diagram, one could restrict domains of unobserved variables to a finite space without loss of generality. This observation allows us to develop novel partial identification algorithms to bound unknown counterfactual probabilities from the observational data. More specifically, our contributions are as follows. (1) We introduce a special family of discrete SCMs, with finite unobserved domains, and show that it could represent all categorical counterfactual distributions in an arbitrary causal diagram. (2) Using this result, we translate the original partial identification task into equivalent polynomial programs. Solving such programs leads to informative bounds over unknown counterfactual probabilities, which are provably optimal. (3) We develop an effective Monte Carlo algorithm to approximate optimal counterfactual bounds from a finite number of observational data. Finally, our algorithms are validated extensively on synthetic datasets. Given space constraints, all proofs are provided in Appendices A and B.

## 1.1 Preliminaries

We introduce in this section some basic notations and definitions that will be used throughout the paper. We use capital letters to denote variables ($X$), small letters for their values ($x$) and $\Omega_X$ for their domains. For an arbitrary set $\mathbf{X}$, let $|\mathbf{X}|$ be its cardinality. For convenience, we denote by $P(\boldsymbol{x})$ probabilities $P(\mathbf{X} = \boldsymbol{x})$; for an arbitrary subdomain $\mathcal{X} \subseteq \Omega_X$, $P(\mathcal{X}) \equiv P(X \in \mathcal{X})$. Finally, the indicator function $\mathbb{1}_{\mathbf{X}=\boldsymbol{x}}$ returns 1 if an event $\mathbf{X} = \boldsymbol{x}$ holds true; otherwise $\mathbb{1}_{\mathbf{X}=\boldsymbol{x}} = 0$.

The basic semantical framework of our analysis rests on *structural causal models* (SCMs) [33, Ch. 7]. An SCM $M$ is a tuple $\langle \mathbf{V}, \mathbf{U}, \mathbf{F}, P \rangle$ where $\mathbf{V}$ is a set of endogenous variables and $\mathbf{U}$ is a set of exogenous variables. $\mathbf{F}$ is a set of functions where each $f_V \in \mathbf{F}$ decides values of an endogenous variable $V \in \mathbf{V}$ taking as argument a combination of other variables in the system. That is, $v \leftarrow f_V(pa_V, u_V), Pa_V \subseteq \mathbf{V}, U_V \subseteq \mathbf{U}$. Exogenous variables $U \in \mathbf{U}$ are mutually independent, values of which are drawn from the exogenous distribution $P(\boldsymbol{u})$. Naturally, $M$ induces a joint distribution $P(\boldsymbol{v})$ over endogenous variables $\mathbf{V}$, called the *observational distribution*. Each SCM is associated with a causal diagram $\mathcal{G}$ (e.g., Fig. 1), which is a directed acyclic graph (DAG) where

solid nodes represent endogenous variables $\boldsymbol{V}$, empty nodes represent exogenous variables $\boldsymbol{U}$ and arrows represent the arguments $Pa_V, U_V$ of each function $f_V$.

An intervention on an arbitrary subset $\boldsymbol{X} \subseteq \boldsymbol{V}$, denoted by $\mathrm{do}(\boldsymbol{x})$, is an operation where values of $\boldsymbol{X}$ are set to constants $\boldsymbol{x}$, regardless of how they are ordinarily determined. For an SCM $M$, let $M_{\boldsymbol{x}}$ denote a submodel of $M$ induced by intervention $\mathrm{do}(\boldsymbol{x})$. For any subset $\boldsymbol{Y} \subseteq \boldsymbol{V}$, the *potential response* $\boldsymbol{Y}_{\boldsymbol{x}}(\boldsymbol{u})$ is defined as the solution of $\boldsymbol{Y}$ in the submodel $M_{\boldsymbol{x}}$ given $\boldsymbol{U} = \boldsymbol{u}$. Drawing values of exogenous variables $\boldsymbol{U}$ following the probability measure $P$ induces a *counterfactual variable* $\boldsymbol{Y}_{\boldsymbol{x}}$. Specifically, the event $\boldsymbol{Y}_{\boldsymbol{x}} = \boldsymbol{y}$ (for short, $\boldsymbol{y}_{\boldsymbol{x}}$) can be read as "$\boldsymbol{Y}$ would be $\boldsymbol{y}$ had $\boldsymbol{X}$ been $\boldsymbol{x}$". For any subsets $\boldsymbol{Y}, \ldots, \boldsymbol{Z}, \boldsymbol{X}, \ldots, \boldsymbol{W} \subseteq \boldsymbol{V}$, the distribution over counterfactuals $\boldsymbol{Y}_{\boldsymbol{x}}, \ldots, \boldsymbol{Z}_{\boldsymbol{w}}$ is defined as:

$$P\left(\boldsymbol{y}_{\boldsymbol{x}}, \ldots, \boldsymbol{z}_{\boldsymbol{w}}\right) = \int_{\Omega_U} \mathbb{1}_{\boldsymbol{Y}_{\boldsymbol{x}}(\boldsymbol{u})=\boldsymbol{y}} \wedge \cdots \wedge \mathbb{1}_{\boldsymbol{Z}_{\boldsymbol{w}}(\boldsymbol{u})=\boldsymbol{z}} \, dP(\boldsymbol{u}). \tag{1}$$

Distributions of the form $P(\boldsymbol{y}_{\boldsymbol{x}})$ is called the *interventional distribution*; when the treatment set $\boldsymbol{X} = \emptyset$, $P(\boldsymbol{y})$ coincides with the *observational distribution*. Throughout this paper, we assume that endogenous variables $\boldsymbol{V}$ are discrete and finite; while exogenous variables $\boldsymbol{U}$ could take any (continuous) value. The counterfactual distribution $P\left(\boldsymbol{y}_{\boldsymbol{x}}, \ldots, \boldsymbol{z}_{\boldsymbol{w}}\right)$ defined above is thus a categorical distribution. For a more detailed survey on SCMs, we refer readers to [33, Ch. 7].

## 2 Discretization of Structural Causal Models

For a DAG $\mathcal{G}$ with endogenous $\boldsymbol{V}$ and exogenous variables $\boldsymbol{U}$, let $\boldsymbol{P}^*$ denote the collection of all counterfactual distributions over variables $\boldsymbol{V}$. Formally,

$$\boldsymbol{P}^* = \left\{ P\left(\boldsymbol{y}_{\boldsymbol{x}}, \ldots, \boldsymbol{z}_{\boldsymbol{w}}\right) \mid \forall \boldsymbol{Y}, \ldots, \boldsymbol{Z}, \boldsymbol{X}, \ldots, \boldsymbol{W} \subseteq \boldsymbol{V} \right\}. \tag{2}$$

Let $\mathscr{M}$ be the family of all the SCMs compatible with the causal diagram $\mathcal{G}$, i.e., $\mathscr{M} = \{\forall M \mid \mathcal{G}_M = \mathcal{G}\}$[1]. Counterfactual distributions in $\mathcal{G}$ are defined as the collection $\{\boldsymbol{P}_M^* : \forall M \in \mathscr{M}\}$ that contains all counterfactual probabilities induced by SCMs $M$ in the candidate family $\mathscr{M}$. In this section, we will show that counterfactual distributions in any causal diagram $\mathcal{G}$ could be generated by an alternative family of "generic" SCMs compatible with $\mathcal{G}$, which we will define later.

**Definition 1** (Counterfactual-Equivalence). For a DAG $\mathcal{G}$, let $\mathscr{M}, \mathscr{N}$ be two sets of SCMs compatible with $\mathcal{G}$. $\mathscr{M}$ and $\mathscr{N}$ are said to be *counterfactually equivalent* (for short, ctf-equivalent) if for any $M \in \mathscr{M}$, there exists an alternative $N \in \mathscr{N}$ such that $\boldsymbol{P}_M^* = \boldsymbol{P}_N^*$, and vice versa.

Our analysis rests on a special family of SCMs where values of each exogenous variable are drawn from a discrete distribution over a finite set of states.

**Definition 2.** An SCM $M = \langle \boldsymbol{V}, \boldsymbol{U}, \boldsymbol{F}, P \rangle$ is said to be a discrete SCM if

1. Values of every $U \in \boldsymbol{U}$ are drawn from a discrete distribution $P(u)$ over a domain $\Omega_U$; let $\theta_u$ denote the probability $P(U = u)$, for any $u \in \Omega_U$. f

2. Values of every $V \in \boldsymbol{V}$ are decided by function $v \leftarrow f_V(pa_V, u_V) \equiv \xi_V^{(pa_V, u_V)}$, where for $\forall pa_V, u_V, \xi_V^{(pa_V, u_V)}$ is a constant in the finite domain $\Omega_V$.

Given a causal diagram $\mathcal{G}$, our goal is to construct a family of discrete SCMs $\mathscr{N}$ that is counterfactually equivalent to the original family of SCMs $\mathscr{M}$. Our construction utilizes a special type of clustering of nodes in the diagram, called the confounded component [45].

**Definition 3.** For an DAG $\mathcal{G}$, a subset $\boldsymbol{C} \subseteq \boldsymbol{V}$ is a c-component if any pair $X, Y \in \boldsymbol{C}$ is connected in $\mathcal{G}$ by a *bi-directed path* of the form $V_1 \leftrightarrow V_2 \leftrightarrow \cdots \leftrightarrow V_n$, $n = 1, 2, \ldots$, where (1) $V_1 = X$, $V_n = Y$; (2) $\{V_1, \ldots, V_n\} \subseteq \boldsymbol{V}$; and (3) each $V_i \leftrightarrow V_j$ is a sequence $V_i \leftarrow U_k \rightarrow V_j$ and $U_k \in \boldsymbol{U}$.

A c-component $\boldsymbol{C}$ in $\mathcal{G}$ is maximal if there exists no other c-component that contains $\boldsymbol{C}$. We denote by $\mathcal{C}(\mathcal{G})$ the collection of all maximal c-components in $\mathcal{G}$. Naturally, c-components in $\mathcal{C}(\mathcal{G})$ form a partition over endogenous variables $\boldsymbol{V}$, which, in turn, defines a partition $\{\cup_{V \in \boldsymbol{C}} U_V \mid \forall \boldsymbol{C} \in \mathcal{C}(\mathcal{G})\}$ over exogenous variables $\boldsymbol{U}$. Therefore, for every $U \in \boldsymbol{U}$, there must exist a unique c-component in $\mathcal{C}(\mathcal{G})$, denoted by $\boldsymbol{C}_U$, such that $U \in \cup_{V \in \boldsymbol{C}_U} U_V$. For example, exogenous variables $U_1, U_2$ in Fig. 1a corresponds to c-components $\boldsymbol{C}_{U_1} = \{Z\}$ and $\boldsymbol{C}_{U_2} = \{X, Y\}$ respectively; while the causal diagram of Fig. 1b only has a single c-component $\{X, Y, Z\}$.

---

[1] We will use the subscript $M$ to represent the restriction to a specific SCM $M$. Therefore, $\mathcal{G}_M$ represents the causal diagram associated with SCM $M$; so does the collection of counterfactuals $\boldsymbol{P}_M^*$.

**Theorem 1.** *For a DAG $\mathcal{G}$, consider the following conditions[2]: (1) $\mathcal{M}$ is the set of all SCMs compatible with $\mathcal{G}$; (2) $\mathcal{N}$ is the set of all discrete SCMs compatible with $\mathcal{G}$ where for every $U \in \boldsymbol{U}$, its cardinality $|\Omega_U| = \prod_{V \in \boldsymbol{C}_U} |\Omega_{Pa_V} \mapsto \Omega_V|$, i.e., the number of functions mapping from $Pa_V$ to $V$ for every variable $V$ in the c-component $\boldsymbol{C}_U$. Then, $\mathcal{M}$ and $\mathcal{N}$ are counterfactually equivalent.*

Thm. 1 establishes the expressive power of discrete SCMs in representing counterfactual distributions in a causal diagram $\mathcal{G}$. It implies that the counterfactual distribution $P(\boldsymbol{y_x}, \ldots, \boldsymbol{z_w})$ in any SCM $M$ could be generated using a generic model as follows, for $d_U = \prod_{V \in \boldsymbol{C}_U} |\Omega_{Pa_V} \mapsto \Omega_V|$,

$$P(\boldsymbol{y_x}, \ldots, \boldsymbol{z_w}) = \sum_{U \in \boldsymbol{U}} \sum_{u=1,\ldots,d_U} \mathbb{1}_{\boldsymbol{Y_x}(\boldsymbol{u})=\boldsymbol{y}} \wedge \cdots \wedge \mathbb{1}_{\boldsymbol{Z_w}(\boldsymbol{u})=\boldsymbol{z}} \prod_{U \in \boldsymbol{U}} \theta_u. \tag{3}$$

Among above quantities, $\theta_u$ are parameters of the exogenous distribution $P(u)$ over a finite domain $\{1, \ldots, d_U\}$. Counterfactual variables $\boldsymbol{Y_x}(\boldsymbol{u})$ are recursively defined as follows:

$$\boldsymbol{Y_x}(\boldsymbol{u}) = \{Y_{\boldsymbol{x}}(\boldsymbol{u}) \mid \forall Y \in \boldsymbol{Y}\}, \text{ where } Y_{\boldsymbol{x}}(\boldsymbol{u}) = \begin{cases} \boldsymbol{x}_Y & \text{if } Y \in \boldsymbol{X} \\ \xi_Y^{(\{V_{\boldsymbol{x}}(\boldsymbol{u}) | V \in Pa_Y\}, u_Y)} & \text{otherwise} \end{cases} \tag{4}$$

where $\boldsymbol{x}_Y$ is the value assigned to variable $Y$ in constants $\boldsymbol{x}$. As an example, consider the causal diagram $\mathcal{G}$ described in Fig. 1b where $X, Y, Z$ are binary variables in $\{0, 1\}$. Since $\mathcal{G}$ has a single c-component $\{X, Y, Z\}$, exogenous variables $U_1, U_2$ must share the same cardinality $d$ in the proposed family of discrete SCMs $\mathcal{N}$. It follows from Thm. 1 the counterfactual distribution $P(z, x_{z'}, y_{x'})$ in any SCM compatible with $\mathcal{G}$ could be written as follows:

$$P(z, x_{z'}, y_{x'}) = \sum_{u_1, u_2=1}^{d} \mathbb{1}_{\xi_Z^{(u_1)}=z} \wedge \mathbb{1}_{\xi_X^{(z',u_1,u_2)}=x} \wedge \mathbb{1}_{\xi_Y^{(x',u_2)}=y} \theta_{u_1} \theta_{u_2}, \tag{5}$$

where $\xi_Z^{(u_1)}, \xi_X^{(z,u_1,u_2)}, \xi_Y^{(x,u_2)}$ are parameters taking values in $\{0, 1\}$; $\theta_{u_i}, i = 1, 2$, are probabilities of the discrete distribution $P(u_i)$ over the finite domain $\{1, \ldots, d\}$. The cardinality $d = |\Omega_Z| \times |\Omega_Z \mapsto \Omega_X| \times |\Omega_X \mapsto \Omega_Y| = 32$. The total cardinalities of domains for $U_1, U_2$ are thus $2d = 64$.

**Comparison with related work**   One could naïvely apply the discretization procedure in [3] and obtain a family of discrete SCMs that are sufficient in representing distributions in an causal diagram. However, such parametrization is not necessarily complete. To witness, consider again the causal diagram in Fig. 1b with binary $X, Y, Z$. Applying the discretization in [3] leads to a family of discrete SCMs compatible with a different diagram in Fig. 1c where the cardinality of exogenous variable $U$ is equal to $d = 32$ (see Appendix D for details). However, this parametrization fails to capture some critical constraints over counterfactual distributions since it does not maintain the original structure of the causal diagram. For instance, counterfactual variables $Z$ and $Y_x$ in the original diagram of Fig. 1b are independent due to independence restrictions [33, Ch. 7.3.2]; while $Z$ and $Y_x$ in Fig. 1c are generally correlated due to the presence of unobserved confounder $U$. Compared with [3], the discretization method in Thm. 1 captures *all* constraints over counterfactual distributions while requiring only a factor of $|\boldsymbol{U}|$ increase in the cardinality of exogenous domains.

More recently, [15] proved a special case of Thm. 1 for interventional distributions in a specific class of causal diagrams that satisfy the running intersection property. When there is no direct arrow between endogenous variables, [38] showed that the observational distribution in a diagram could be represented using finite-state exogenous variables. Thm. 1 generalizes these results by showing that, for the first time, *all* counterfactual distributions in an *arbitrary* causal diagram could be generated using discrete exogenous variables taking values from a finite domain, without any loss of generality.

## 2.1 Partial identification of Counterfactual Distributions

To demonstrate the expressive power of discrete SCMs, we investigate the problem of partial identification of counterfactual distributions. For an SCM $M^* = \langle \boldsymbol{V}, \boldsymbol{U}, \boldsymbol{F}, P \rangle$, we are interested in evaluating an arbitrary counterfactual probability $P(\boldsymbol{y_x}, \ldots, \boldsymbol{z_w})$. The detailed parametrization of $M^*$ is unknown. Instead, the learner only has access to the causal diagram $\mathcal{G}$ and the observational distribution $P(\boldsymbol{v})$ induced by $M^*$. Our goal is to derive an informative bound $[l, r]$ from the combination of $\mathcal{G}$ and $P(\boldsymbol{v})$ that contains the actual counterfactual probability $P(\boldsymbol{y_x}, \ldots, \boldsymbol{z_w})$.

---

[2]For every $V \in \boldsymbol{V}$, $\Omega_{Pa_V} \mapsto \Omega_V$ is the set of all functions mapping from domains $\Omega_{Pa_V}$ to $\Omega_V$.

Let $\mathcal{N}$ denote the family of discrete SCMs defined in Thm. 1 which are compatible with the causal diagram $\mathcal{G}$. We derive a bound $[l, r]$ over $P(\boldsymbol{y_x}, \ldots, \boldsymbol{z_w})$ from the observational data $P(\boldsymbol{v})$ by solving the following optimization problem:

$$[l, r] = \min / \max \left\{ P_N(\boldsymbol{y_x}, \ldots, \boldsymbol{z_w}) \mid \forall N \in \mathcal{N}, P_N(\boldsymbol{v}) = P(\boldsymbol{v}) \right\} \qquad (6)$$

For instance, consider again the double-bow diagram $\mathcal{G}$ in Fig. 1b. The observational distribution $P(x, y, z)$ in any discrete SCM in $\mathcal{N}$ could be written as:

$$P(x, y, z) = \sum_{u_1, u_2 = 1}^{d} \mathbb{1}_{\xi_Z^{(u_1)} = z} \wedge \mathbb{1}_{\xi_X^{(z, u_1, u_2)} = x} \wedge \mathbb{1}_{\xi_Y^{(x, u_2)} = y} \theta_{u_1} \theta_{u_2}. \qquad (7)$$

One could derive a bound over the counterfactual distribution $P(z, x_{z'}, y_{x'})$ from the observational data $P(x, y, z)$ by solving polynomial programs which optimize the objective Eq. (5) over parameters $\theta_{u_1}, \theta_{u_2}, \xi_Z^{(u_1)}, \xi_X^{(z, u_1, u_2)}, \xi_Y^{(x, u_2)}$, subject to the observational constraints Eq. (7).

As a corollary, it follows immediately from Thm. 1 that the solution $[l, r]$ of the optimization problem Eq. (6) is guaranteed to be a valid bound over the unknown counterfactual $P(\boldsymbol{y_x}, \ldots, \boldsymbol{z_w})$.

**Corollary 1** (Soundness)**.** *Given a DAG $\mathcal{G}$ and an observational distribution $P(\boldsymbol{v})$, let $\mathscr{M}$ be the set of all SCMs compatible with $\mathcal{G}$ and let $\mathscr{M}_o = \{\forall M \in \mathscr{M} \mid P_M(\boldsymbol{v}) = P(\boldsymbol{v})\}$. For the solution $[l, r]$ of Eq. (6), $P_M(\boldsymbol{y_x}, \ldots, \boldsymbol{z_w}) \in [l, r]$ for any SCM $M \in \mathscr{M}_o$.*

Since the underlying SCM $M^* \in \mathscr{M}_o$, Corol. 1 implies that the derived bound $[l, r]$ must contain the actual counterfactual probability $P(\boldsymbol{y_x}, \ldots, \boldsymbol{z_w})$. Our next result shows that such a bound $[l, r]$ is provably tight, i.e., it cannot be improved without additional assumptions.

**Corollary 2** (Tightness)**.** *Given a DAG $\mathcal{G}$ and an observational distribution $P(\boldsymbol{v})$, let $\mathscr{M}$ be the set of all SCMs compatible with $\mathcal{G}$ and let $\mathscr{M}_o = \{\forall M \in \mathscr{M} \mid P_M(\boldsymbol{v}) = P(\boldsymbol{v})\}$. For the solution $[l, r]$ of Eq. (6), there exist SCMs $M_1, M_2 \in \mathscr{M}_o$ such that $P_{M_1}(\boldsymbol{y_x}, \ldots, \boldsymbol{z_w}) = l$, $P_{M_2}(\boldsymbol{y_x}, \ldots, \boldsymbol{z_w}) = r$.*

Corol. 2 confirms the tightness of the bound $[l, r]$ obtained from Eq. (6). Suppose there exists a valid bound $[l', r']$ strictly contained in $[l, r]$. One could construct from Corol. 2 an SCM $M$ compatible with the causal diagram $\mathcal{G}$ and the observational distribution $P(\boldsymbol{v})$, but its counterfactual probability $P(\boldsymbol{y_x}, \ldots, \boldsymbol{z_w})$ lies outside $[l', r']$, which is a contradiction.

The optimization problem of Eq. (6) is reducible to equivalent polynomial programs (see Appendix E). Despite the soundness and tightness of derived bounds, solving such programs may take exponentially long in the most general case [29]. Our focus here is upon the causal inference aspect of the problem and like earlier discussions we do not specify which solvers are used [3, 4]. In some cases of interest, effective approximate planning methods for polynomial programs do exist. Investigating these methods is an ongoing subject of research [26, 31, 48, 28, 27].

# 3 Bayesian Approach for Partial Identification

This section describes an effective algorithm to approximate the optimal counterfactual bound in Eq. (6), provided with finite samples $\bar{\boldsymbol{v}} = \left\{ \boldsymbol{v}^{(n)} \right\}_{n=1}^{N}$ drawn from the observational distribution $P(\boldsymbol{v})$, and prior distributions over parameters $\theta_u$ and $\xi_V^{(pa_V, u_V)}$ (possibly uninformative).

We first introduce Markov Chain Monte Carlo (MCMC) algorithms that sample the posterior distribution $P(\theta_{\text{ctf}} \mid \bar{\boldsymbol{v}})$ over a counterfactual probability $\theta_{\text{ctf}} = P(\boldsymbol{y_x}, \ldots, \boldsymbol{z_w})$. More specifically, for every $V \in \boldsymbol{V}$, $\forall pa_V, u_V$, parameters $\xi_V^{(pa_V, u_V)}$ are drawn uniformly over the finite domain $\Omega_V$. For every $U \in \boldsymbol{U}$, exogenous probabilities $\theta_u$ are drawn from a generalized Dirichlet distribution [12]. We will take the view of a stick-breaking construction [40] which successively breaks pieces off a unit-length stick with size proportional to random draws from a Beta distribution. Parameters $\theta_u$ are proportions of each of the pieces relative to its original size. Formally,

$$\forall u = 1, 2, \ldots, d_U, \qquad \theta_u = \mu_u \prod_{i=1}^{u-1} (1 - \mu_i), \qquad \mu_u \sim \texttt{Beta}\left(\alpha_U^{(u)}, \beta_U^{(u)}\right), \qquad (8)$$

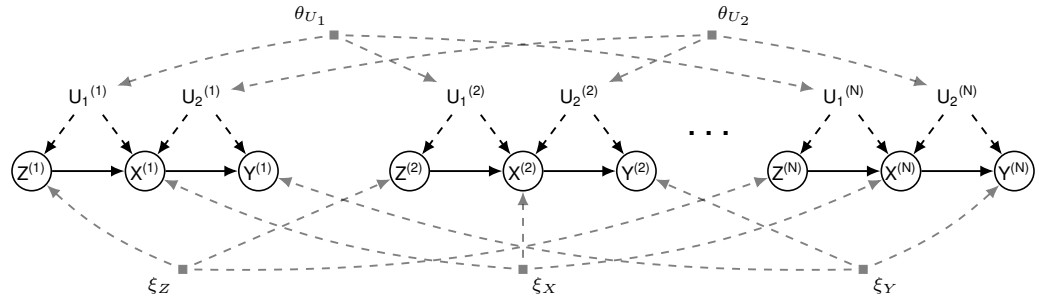

Figure 2: The data-generating process for the observational data $\left\{ X^{(n)}, Y^{(n)}, Z^{(n)} \right\}_{n=1}^{N}$ in an SCM associated with the causal diagram in Fig. 1b. For every exogenous variable $U \in \boldsymbol{U}$, $\theta_U = \{\theta_u \mid \forall u\}$. For every endogenous variable $V \in \boldsymbol{V}$, $\xi_V = \left\{ \xi_V^{(pa_V, u_V)} \mid \forall pa_V, u_V \right\}$.

where $d_U = \prod_{V \in \boldsymbol{C}_U} |\Omega_{Pa_V} \mapsto \Omega_V|$ and $\alpha_U^{(u)}, \beta_U^{(u)} > 0$ are hyperparameters. Finally, we truncate this construction by setting $\mu_{d_U} = 1$. Note from Eq. (8) that all parameters $\theta_u$ for $u > d_U$ are equal to zero. As an example, Fig. 2 shows a graphical representation of the data-generating process over parameters $\theta_u$ and $\xi_V^{(pa_V, u_V)}$ associated with SCMs in Fig. 1b, spanning over $N$ observations.

Gibbs sampling is a well-known MCMC algorithm that allows one to sample posterior distributions. For convenience, we introduce the following notations. Let parameters $\boldsymbol{\theta} = \{\theta_u \mid \forall U \in \boldsymbol{U}, \forall u\}$ and $\boldsymbol{\xi} = \left\{ \xi_V^{(pa_V, u_V)} \mid \forall V \in \boldsymbol{V}, \forall pa_V, u_V \right\}$. The set $\bar{\boldsymbol{U}} = \left\{ \boldsymbol{U}^{(n)} \right\}_{n=1}^{N}$ are exogenous variables affecting $N$ observations $\bar{\boldsymbol{V}} = \left\{ V^{(n)} \right\}_{n=1}^{N}$; we use $\bar{u}$ to represent their realizations. Our blocked Gibbs sampler works by iteratively drawing values from the conditional distributions of variables as follows [22]. Detailed derivations of complete conditional distributions are shown in Appendix F.

**Sampling** $P\left(\bar{u} \mid \bar{v}, \boldsymbol{\theta}, \boldsymbol{\xi}\right)$. Exogenous variables $\boldsymbol{U}^{(n)}$, $n = 1, \dots, N$, are mutually independent given parameters $\boldsymbol{\theta}, \boldsymbol{\xi}$. We could draw each $\left(\boldsymbol{U}^{(n)} \mid \boldsymbol{\theta}, \boldsymbol{\xi}, \bar{\boldsymbol{V}}\right)$ corresponding to the $n$th observation independently. The complete conditional for $\boldsymbol{U}^{(n)}$ is given by

$$P\left(\boldsymbol{u}^{(n)} \mid \boldsymbol{v}^{(n)}, \boldsymbol{\theta}, \boldsymbol{\xi}\right) \propto \prod_{V \in \boldsymbol{V}} \mathbb{1}_{\xi_V^{\left(pa_V^{(n)}, u_V^{(n)}\right)} = v^{(n)}} \prod_{U \in \boldsymbol{U}} \theta_u. \tag{9}$$

**Sampling** $P\left(\boldsymbol{\xi}, \boldsymbol{\theta} \mid \bar{v}, \bar{u}\right)$. Parameters $\boldsymbol{\xi}, \boldsymbol{\theta}$ are independent given $\bar{\boldsymbol{V}}, \bar{\boldsymbol{U}}$. Therefore, we will derive complete conditional $\boldsymbol{\xi}, \boldsymbol{\theta}$ separately. Note that in discrete SCMs, the $n$th observation of variable $V \in \boldsymbol{V}$ is decided by $v^{(n)} \leftarrow \xi_V^{(pa_V, u_V)}$ given $pa_V^{(n)} = pa_V, u_V^{(n)} = u_V$. Thus, draw values of each $\xi_V^{(pa_V, u_V)} \in \boldsymbol{\xi}$ from the complete conditional defined as:

$$P\left(\xi_V^{(pa_V, u_V)} \mid \bar{v}, \bar{u}\right) = \begin{cases} \mathbb{1}_{\xi_V^{(pa_V, u_V)} = v^{(i)}} & \text{if } \exists i, \text{ s.t. } pa_V^{(i)} = pa_V, u_V^{(i)} = u_V, \\ 1/|\Omega_V| & \text{otherwise.} \end{cases} \tag{10}$$

Let $n_u = \sum_{n=1}^{N} \mathbb{1}_{u^{(n)} = u}$ records the number of values in $u^{(n)}$ that are equal to $u$. By the conjugacy of the generalized Dirichlet distribution, the complete conditional of $\theta_u$ is given by, for every $U \in \bar{\boldsymbol{U}}$,

$$\forall u = 1, 2, \dots d_U, \quad \theta_u = \mu_u \prod_{i=1}^{u-1} (1 - \mu_i), \quad \mu_u \sim \texttt{Beta}\left(\alpha_U^{(u)} + n_u, \beta_U^{(u)} + \sum_{k=u+1}^{d_U} n_k\right). \tag{11}$$

Doing so eventually produces values drawn from the posterior distribution over $\left(\boldsymbol{\theta}, \boldsymbol{\xi}, \bar{\boldsymbol{U}} \mid \bar{\boldsymbol{V}}\right)$. Given parameters $\boldsymbol{\theta}, \boldsymbol{\xi}$, we compute the counterfactual probability $\theta_{\text{ctf}} = P(\boldsymbol{y_x}, \dots, \boldsymbol{z_w})$ following the three-step algorithm in [33] which consists of abduction, action, and prediction. Thus computing $\theta_{\text{ctf}}$ from each draw $\boldsymbol{\theta}, \boldsymbol{\xi}, \bar{\boldsymbol{U}}$ eventually gives us the draw from the posterior distribution $P\left(\theta_{\text{ctf}} \mid \bar{v}\right)$.

### 3.1 Collapsed Gibbs Sampling

We also describe an alternative sampler that applies to stick-breaking priors with a known Pólya urn characterization. Formally, consider stick-breaking priors in Eq. (8) with hyperparameters

$\alpha_U^{(u)} = \alpha_U/d_U$ and $\beta_U^{(u)} = (d_U - u)\alpha_U/d_U$ for some real $\alpha_U > 0$. Let $\bar{U}_{-n}$ denote the set difference $\bar{U} \setminus U^{(n)}$; so does $\bar{V}_{-n} = \bar{V} \setminus V^{(n)}$. Our collapsed Gibbs sampler first iteratively draws values from the conditional distribution of $(U^{(n)} \mid \bar{U}_{-n}, \bar{V})$, $n = 1, \ldots, N$, as follows.

**Sampling $P(u^{(n)} \mid \bar{v}, \bar{u}_{-n})$.** At each iteration, draw $U^{(n)}$ from the conditional given by

$$P\left(u^{(n)} \mid \bar{v}, \bar{u}_{-n}\right) \propto \prod_{V \in \boldsymbol{V}} P\left(v^{(n)} \mid pa_V^{(n)}, u_V^{(n)}, \bar{v}_{-n}, \bar{u}_{-n}\right) \prod_{U \in \boldsymbol{U}} P\left(u^{(n)} \mid \bar{v}_{-n}, \bar{u}_{-n}\right). \quad (12)$$

Among quantities in the above equation, for every $V \in \boldsymbol{V}$,

$$P\left(v^{(n)} \mid pa_V^{(n)}, u_V^{(n)}, \bar{v}_{-n}, \bar{u}_{-n}\right) = \begin{cases} \mathbb{1}_{v^{(n)}=v^{(i)}} & \text{if } \exists i \neq n,\, pa_V^{(i)} = pa_V^{(n)},\, u_V^{(i)} = u_V^{(n)}, \\ 1/|\Omega_V| & \text{otherwise.} \end{cases} \quad (13)$$

For every $U \in \boldsymbol{U}$, let $\bar{u}_{-n}$ be a set of exogenous samples $\{u^{(1)}, \ldots, u^{(n-1)}, u^{(n+1)}, \ldots, u^{(N)}\}$. Let $\{u_1^*, \ldots, u_K^*\}$ denote $K$ unique values that samples in $\bar{u}_{-n}$ take on.

$$P\left(u^{(n)} \mid \bar{v}_{-n}, \bar{u}_{-n}\right) = \begin{cases} \dfrac{n_k^* + \alpha_U/d_U}{\alpha_U + N - 1} & \text{if } u^{(n)} = u_k^*, \text{ for } k = 1, \ldots, K \\ \dfrac{\alpha_U(1 - K/d_U)}{\alpha_U + N - 1} & \text{if } u^{(n)} \notin \{u_1^*, \ldots, u_K^*\} \end{cases}. \quad (14)$$

where $n_k^* = \sum_{i \neq n} \mathbb{1}_{u^{(i)}=u_k^*}$ records the number of values in $u^{(i)} \in \bar{u}_{-n}$ that are equal to $u_k^*$.

Doing so eventually produces exogenous variables drawn from the posterior distribution of $(\bar{U} \mid \bar{V})$. We then sample parameters from the posterior distribution of $(\boldsymbol{\theta}, \boldsymbol{\xi} \mid \bar{U}, \bar{V})$; the complete conditional $P(\boldsymbol{\xi}, \boldsymbol{\theta} \mid \bar{v}, \bar{u})$ are given in Eqs. (10) and (11). Finally, computing $\theta_{\text{ctf}}$ from each sample $\boldsymbol{\theta}, \boldsymbol{\xi}$ gives us a draw from the posterior distribution $P(\theta_{\text{ctf}} \mid \bar{v})$.

When the cardinality $d_U$ of exogenous domains is high, the collapsed Gibbs sampler described here is more computational efficient than the blocked sampler, since it does not iteratively draw parameters $\boldsymbol{\theta}, \boldsymbol{\xi}$ in the high-dimensional space. Instead, the collapsed sampler only draws $\boldsymbol{\theta}, \boldsymbol{\xi}$ once after samples drawn from the distribution of $(\bar{U} \mid \bar{V})$ converge. On the other hand, when the cardinality $d_U$ is reasonably low, the blocked Gibbs sampler is preferable since it exhibits better convergence [22].

### 3.2 Credible Intervals over Counterfactual Probabilities

Given a MCMC sampler, one could bound the counterfactual probability $\theta_{\text{ctf}}$ by computing credible intervals from the posterior distribution $P(\theta_{\text{ctf}} \mid \bar{v})$.

**Definition 4.** Fix $\alpha \in [0, 1)$. A $100(1 - \alpha)\%$ credible interval $[l_\alpha, r_\alpha]$ for $\theta_{\text{ctf}}$ is given by

$$l_\alpha = \sup\{x \mid P(\theta_{\text{ctf}} \leq x \mid \bar{v}) = \alpha/2\}, \qquad r_\alpha = \inf\{x \mid P(\theta_{\text{ctf}} \leq x \mid \bar{v}) = 1 - \alpha/2\}. \quad (15)$$

For a $100(1 - \alpha)\%$ credible interval $[l_\alpha, r_\alpha]$, any counterfactual probability $\theta_{\text{ctf}}$ that is compatible with observational data $\bar{v}$ lies between the interval $l_\alpha$ and $r_\alpha$ with probability $1 - \alpha$. Credible intervals have been widely applied for computing bounds over counterfactuals provided with finite observations [20, 47, 37, 8, 46]. As the number of observational data $N$ grows (to infinite), the $100\%$ credible interval $[l_0, r_0]$ eventually converges to the optimal asymptotic bound $[l, r]$ in Eq. (6) [11].

Let $\{\theta^{(t)}\}_{t=1}^{T}$ be $T$ samples drawn from $P(\theta_{\text{ctf}} \mid \bar{v})$. One could compute the $100(1 - \alpha)\%$ credible interval for $\theta_{\text{ctf}}$ using the following consistent estimators [39]:

$$\hat{l}_\alpha(T) = \theta^{(\lceil (\alpha/2)T \rceil)}, \qquad \hat{r}_\alpha(T) = \theta^{(\lceil (1-\alpha/2)T \rceil)}, \quad (16)$$

where $\theta^{(\lceil (\alpha/2)T \rceil)}, \theta^{(\lceil (1-\alpha/2)T \rceil)}$ are the $\lceil (\alpha/2)T \rceil$th smallest and the $\lceil (1 - \alpha/2)T \rceil$th smallest of $\{\theta^{(t)}\}$[3]. Our next results establish non-asymptotic deviation bounds for the empirical estimates of credible intervals defined in Eq. (16) for finite samples.

**Lemma 1.** *Fix $T > 0$ and $\delta \in (0, 1)$. Let function $f(T, \delta) = \sqrt{2T^{-1}\ln(4/\delta)}$. With probability at least $1 - \delta$, estimators $\hat{l}_\alpha(T), \hat{r}_\alpha(T)$ for any $\alpha \in [0, 1)$ is bounded by*

$$\hat{l}_\alpha(T) \in \left[l_{\alpha-f(T,\delta)}, l_{\alpha+f(T,\delta)}\right], \qquad \hat{r}_\alpha(T) \in \left[r_{\alpha+f(T,\delta)}, r_{\alpha-f(T,\delta)}\right]. \quad (17)$$

---

[3]For any real $\alpha \in \mathbb{R}$, $\lceil \alpha \rceil$ denotes the smallest integer $n \in \mathbb{Z}$ larger than $\alpha$, i.e., $\lceil \alpha \rceil = \min\{n \in \mathbb{Z} \mid n \geq \alpha\}$.

269 We summarize our algorithm, CREDIBLEIN-
270 TERVAL, in Alg. 1. It takes a credible level
271 $\alpha$ and tolerance levels $\delta, \epsilon$ as inputs. In par-
272 ticular, CREDIBLEINTERVAL repeatedly draw
273 $T \geq \lceil 2\epsilon^{-2} \ln(4/\delta) \rceil$ samples from $P(\theta_{\text{ctf}} \mid \bar{\boldsymbol{v}})$.
274 It then computes estimates $\hat{l}_\alpha(T), \hat{h}_\alpha(T)$ from
275 drawn samples following Eq. (16) and return
276 them as the output. It follows immediately from
277 Lem. 1 that such a procedure efficiently approx-
278 imates a $100(1-\alpha)\%$ credible interval.

---

**Algorithm 1:** CREDIBLEINTERVAL
1: **Input:** Credible level $\alpha$, tolerance level $\delta, \epsilon$.
2: **Output:** An credible interval $[l_\alpha, h_\alpha]$ for $\theta_{\text{ctf}}$.
3: Let $T = \lceil 2\epsilon^{-2} \ln(4/\delta) \rceil$.
4: Draw samples $\left\{ \theta^{(1)}, \dots, \theta^{(T)} \right\}$ from the posterior distribution $P(\theta_{\text{ctf}} \mid \bar{\boldsymbol{v}})$.
5: Return interval $\left[ \hat{l}_\alpha(T), \hat{r}_\alpha(T) \right]$ (Eq. (16)).

---

279 **Corollary 3.** *Fix $\delta \in (0,1)$ and $\epsilon > 0$. With probability at least $1-\delta$, the interval $[\hat{l}, \hat{r}] =$*
280 CREDIBLEINTERVAL$(\alpha, \delta, \epsilon)$ *for any $\alpha \in [0,1)$ is bounded by $\hat{l} \in [l_{\alpha-\epsilon}, l_{\alpha+\epsilon}]$ and $\hat{r} \in [r_{\alpha+\epsilon}, r_{\alpha-\epsilon}]$.*

281 Corol. 3 implies that any counterfactual parameter $\theta_{\text{ctf}}$ compatible with observational data $\bar{\boldsymbol{v}}$ falls
282 between $[\hat{l}, \hat{r}] =$ CREDIBLEINTERVAL$(\alpha, \delta, \epsilon)$ with probability $P\left( \theta_{\text{ctf}} \in [\hat{l}, \hat{r}] \mid \bar{\boldsymbol{v}} \right) \approx 1 - \alpha \pm \epsilon$. As
283 the tolerance rate $\epsilon \to 0$, $[\hat{l}, \hat{r}]$ converges to a $100(1-\alpha)\%$ credible interval with high probability.

## 4 Simulations and Experiments

285 We demonstrate our algorithms on various simulated SCM instances and a real world patient dataset
286 collected from the International Stroke Trial (IST) [10]. Overall, we found that simulation results sup-
287 port our findings and the proposed bounding strategy consistently dominates state-of-art algorithms.
288 When target distributions are identifiable (Experiment 1), our bounds collapse to the actual, unknown
289 counterfactual probabilities. For non-identifiable settings, our algorithm obtains sharp asymptotic
290 bounds when closed-form solutions already exist (Experiments 2 & 3); and improves over state-of-art
291 bounds in other more general cases where the optimal strategy is unknown (Experiment 4).

292 In all experiments, we evaluate our proposed bounding strategy based on credible intervals (*ci*). In
293 particular, we draw $4 \times 10^3$ samples from the posterior distribution over the target counterfactual
294 $\left( \theta_{\text{ctf}} \mid \bar{V} \right)$. This allows us to compute $100\%$ credible interval over $\theta_{\text{ctf}}$ within error $\epsilon = 0.05$, with
295 probability at least $1-\delta = 0.95$. As the baseline, we also include the actual counterfactual probability
296 $\theta^*$. For details on simulation setups and additional experiments, we refer readers to Appendix C.

297 **Experiment 1: Frontdoor Graph**    This experiment evaluates our sam-
298 pling algorithm on interventional probabilities that are identifiable from
299 the observational data. Consider the "Frontdoor" graph described in
300 Fig. 3 where $X, Y, W$ are binary variables in $\{0, 1\}$; $U_1, U_2 \in \mathbb{R}$. In this
301 case, the interventional distribution $P(y_x)$ is identifiable from $P(x, w, y)$
302 through the frontdoor adjustment [33, Thm. 3.3.4]. We collect $N = 10^5$
303 observational samples $\bar{V} = \{X^{(n)}, Y^{(n)}, W^{(n)}\}_{n=1}^N$ from a randomly

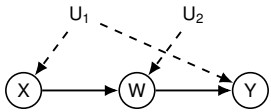

Figure 3: Frontdoor

304 generated SCM. Fig. 4a shows samples drawn from the posterior distribution of the target probability
305 $\left( P(Y_{x=0} = 1) \mid \bar{V} \right)$. The analysis reveals that these samples collapse to the actual interventional
306 probability $P(Y_{x=0} = 1) = 0.5085$, which confirms the identifiability of $P(y_x)$ in Fig. 3.

307 **Experiment 2: Instrumental Variables (IV)**    This experiment evaluates our bounding strategy in
308 non-identifiable settings, while closed-form solutions for the optimal bounds over target probabilities
309 already exist. Consider first the "IV" diagram in Fig. 1a where $X, Y, Z \in \{0, 1\}$ and $U_1, U_2 \in \mathbb{R}$.
310 The non-identifiability of $P(y_x)$ from the observational data $P(x, y, z)$ with the instrument $Z$ and the
311 unobserved confounding between $X$ and $Y$ has been acknowledged in [5]. For binary $X, Y, Z$, [2]
312 derived closed-form, sharp bounds over $P(y_x)$ (labelled as *opt*). We collect $N = 10^5$ observational
313 samples $\bar{V} = \{X^{(n)}, Y^{(n)}, Z^{(n)}\}_{n=1}^N$ from a randomly generated SCM instance. Fig. 4b shows
314 samples drawn from the posterior distribution of $\left( P(Y_{x=0} = 1) \mid \bar{V} \right)$. As a baseline, we also include
315 the optimal bound *opt*, and posterior samples obtained from the Gibbs sampler of [11], which utilizes
316 the canonical partitions of exogenous domains in [2] (*bp*). The analysis reveals that our algorithm
317 derives the valid bound over the actual probability $P(Y_{x=0} = 1) = 0.3954$; the $100\%$ credible
318 interval converges to the optimal IV bound $l = 0.1468, r = 0.6617$.

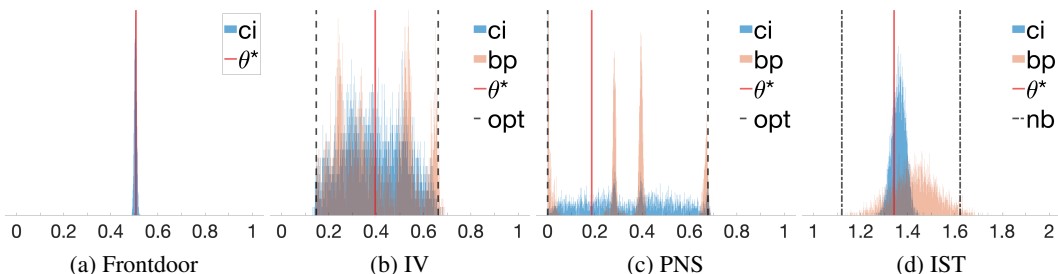

Figure 4: Histogram plots for samples drawn from the posterior distribution over target counterfactual probabilities. For all plots (a - d), *ci* represents our proposed algorithms; *bp* stands for Gibbs samplers using the representation of canonical partitions [2]; $\theta^*$ is the actual counterfactual probability. (b, c) *opt* represents the optimal asymptotic bound, if exists. (d) *nb* stands for the natural bounds [30].

**Experiment 3: Probability of Necessity and Sufficiency (PNS)** We now study the problem of evaluating the *probability of necessity and sufficiency* $P(Y_{x=1} = 1, Y_{x=0} = 0)$ from the observational data $P(x, y)$ in the "Bow" diagram of Fig. 1d where $X, Y \in \{0, 1\}$ and $U \in \mathbb{R}$. The sharp bound for $P(Y_{x=1} = 1, Y_{x=0} = 0)$ from $P(x, y)$ was introduced in [44] (labelled as *opt*). We collect $N = 10^5$ observational samples $\bar{V} = \{X^{(n)}, Y^{(n)}\}_{n=1}^N$ from an SCM instance. Fig. 4c shows samples drawn from the posterior distribution of $\left(P(Y_{x=1} = 1, Y_{x=0} = 0) \mid \bar{V}\right)$. As a baseline, we also include the optimal bound *opt*, and posterior samples obtained from the Gibbs sampler which discretizes the exogenous domains using canonical partitions [2] (*bp*). The analysis reveals that our $100\%$ credible interval (*ci*) matches the optimal PNS bound $l = 0, r = 0.6775$, i.e., the proposed strategy achieves the sharp bound over the counterfactual probability $P(Y_{x=1} = 1, Y_{x=0} = 0) = 0.1867$.

**Experiment 4: International Stroke Trials (IST)** IST was a large, randomized, open trial of up to 14 days of antithrombotic therapy after stroke onset [10]. In particular, the treatment $X$ is a pair $(i, j)$ where $i = 0$ stands for no aspirin allocation, 1 otherwise; $j = 0$ stands for no heparin allocation, 1 for median-dosage, and 2 for high-dosage. The primary outcome $Y \in \{0, \ldots, 3\}$ is the health of the patient 6 months after the treatment, where 0 stands for death, 1 for being dependent on the family, 2 for the partial recovery, and 3 for the full recovery.

To emulate the presence of unobserved confounding, we filter the experimental data with selection rules $f_X^{(Z)}$, $Z \in \{0, \ldots, 9\}$, following a procedure in [49]. Doing so allows us to obtain $N = 3 \times 10^3$ synthetic observational samples $\bar{V} = \{X^{(n)}, Y^{(n)}, Z^{(n)}\}_{n=1}^N$ that are compatible with the "Double bow" diagram of Fig. 1b. We are interested in evaluating the treatment effect $E[Y_{x=(1,0)}]$ for only assigning aspirin $X = (1, 0)$. Fig. 4d shows samples drawn from the posterior distribution of $\left(E[Y_{x=(1,0)}] \mid \bar{V}\right)$. As a baseline, we also include a naïve generalization of the discretization procedure (*bp*) [2] (see Appendix D) and the natural bounds [36, 30] estimated at the $95\%$ confidence level (*nb*) [49]. Posterior samples of *ci* and *bp* are drawn using our proposed collapsed sampler due to the high-dimensional latent space. The analysis reveals that all algorithms achieve bounds that contain the actual, target causal effect $E[Y_{x=(1,0)}] = 1.3418$. Our bounding strategy obtains a $100\%$ credible interval $l_{ci} = 1.2604, r_{ci} = 1.4687$, which consistently improves over all the other algorithms ($l_{bp} = 1.1121, r_{bp} = 1.8073, l_{nb} = 1.1195, r_{nb} = 1.6221$).

## 5 Conclusion

This paper investigated the problem of partial identification of counterfactual distributions, which concerns with bounding unknown counterfactual probabilities from the combination of the observational data and qualitative assumptions of the data-generating process, represented in the form of a directed acyclic causal diagram. We studied a special family of SCMs with discrete exogenous variables, taking values from a finite set of unobserved states, and showed that it could represent *all* counterfactual distributions (over finite observed variables) in an arbitrary causal diagram. That is, this new family of discrete SCMs is counterfactual equivalent to the original family of candidate SCMs compatible with the causal diagram. Using this result, we developed a novel algorithm to derive bounds over counterfactual probabilities from finite observations, which are provably tight.

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
