over finite observed domains. For convenience, we will focus on the following equivalent definition of discrete SCMs which will facilitate the understanding of the proof.

**Definition 5.** An SCM $M = \langle \boldsymbol{V}, \boldsymbol{U}, \boldsymbol{F}, P \rangle$ is said to be a discrete SCM if

1. For each exogenous $U \in \boldsymbol{U}$, its domain $\Omega_U$ is discrete and at most countable;
2. For each endogenous $V \in \boldsymbol{V}$, its domain $\Omega_V$ is discrete and finite;
3. Values of each endogenous $V \in \boldsymbol{V}$ are given by $v \leftarrow h_{u_V}(pa_V)$ where $h_{u_V}$ is a function mapping from finite domains of $Pa_V$ to $V$.

For every $V \in \boldsymbol{V}$, we denote by $\mathscr{H}_V$ a hypothesis class containing all function mapping from domains of $Pa_V$ to $V$, i.e., $\mathscr{H}_V = \Omega_{Pa_V} \mapsto \Omega_V$.

The main challenge in our proof is to show that given an arbitrary SCM $M$ with arbitrary exogenous domains, one could construct a discrete SCM $N$, with bounded cardinality of exogenous domains, such that $N$ and $M$ induces the same counterfactual distributions and the causal diagram. To illustrate this idea, consider the sample "Bow" graph in Fig. 1d where $X, Y$ are binary variables in $\{0, 1\}$. Since $Y$ is not a descendant of $X$, counterfactual variable $X_y = X$ for any $y \in \Omega_Y$, i.e., intervening on $Y$ has no causal effect on $X$ [18]. It is thus sufficient to consider the counterfactual distribution $P(x, y_{x=0}, y_{x=1})$. Let functions in the hypothesis class $\mathcal{H}_X$ be ordered by $h_X^{(1)} = 0$ and $h_X^{(2)} = 1$; and let functions in the hypothesis class $\mathcal{H}_Y$ be ordered by:

$$h_Y^{(1)}(x) = 0, \qquad h_Y^{(2)}(x) = x, \qquad h_Y^{(3)}(x) = \neg x, \qquad h_Y^{(4)}(x) = 1. \qquad (18)$$

Let $\mathscr{M}$ be the set of all SCMs compatible with $\mathcal{G}$ and let $\mathscr{N}$ be the set of all discrete SCMs compatible with $\mathcal{G}$ and discrete exogenous domain $|\Omega_U| \leq 8$. To prove the counterfactual equivalence between $\mathscr{M}$ and $\mathscr{N}$, it suffices to show that for any $M \in \mathscr{M}$, one could construct an $N \in \mathscr{N}$ so that $P_M(x, y_{x=0}, y_{x=1}) = P_N(x, y_{x=0}, y_{x=1})$. The construction procedure is described as follows. Let the exogenous $U$ in $N$ be a pair $(U_X, U_Y)$ where $U_X \in \{1, 2\}$ and $U_Y \in \{1, \ldots, 4\}$; values of $X$ are given by $x \leftarrow h_X^{(u_X)}$; values of $Y$ are given by $y \leftarrow h_Y^{(u_Y)}(x)$. It is verifiable that in such $N$, the counterfactual distribution $P(x, y_{x=0}, y_{x=1})$ equates to, for all $i, j, k \in \{0, 1\}$,

$$P_N(X = i, Y_{x=0} = j, Y_{x=1} = k) = P_N(U_X = i + 1, U_Y = 2j + k + 1). \qquad (19)$$

For any SCM $M \in \mathscr{M}$, let the exogenous distribution $P_N(u_X, u_Y)$ be, for all $i, j, k \in \{0, 1\}$,

$$P_N(U_X = i + 1, U_Y = 2j + k + 1) = P_M(X = i, Y_{x=0} = j, Y_{x=1} = k). \qquad (20)$$

It follows from Eqs. (19) and (20) that $M$ and $N$ coincide in the counterfactual distribution $P(x, y_{x=0}, y_{x=1})$. That is, when inferring counterfactual distributions in Fig. 1d with binary $X, Y$, we could assume that the exogenous variable $U$ is finite and discrete, without any loss of generality.

For the remainder of this section, we will generalize the construction described above to arbitrary causal diagrams. Our analysis rests on the framework of structural causal models and the measure-theoretic probability theory. Formally, each $U \in \boldsymbol{U}$ is associated with a probability space $\langle \Omega_U, \mathcal{F}_U, P_U \rangle$ where $\Omega_U$ is a sample space containing all possible outcomes; $\mathcal{F}_U$ is an event space containing subsets of $\Omega_U$; and $P_U$ is a probability measure mapping from events $\mathcal{F}_U$ to reals in $[0, 1]$. Values of exogenous variables $\boldsymbol{U}$ are drawn following the product measure $P \equiv \otimes_{U \in \boldsymbol{U}} P_U$. We refer readers to [6, 7] for a detailed introduction to the measure-theoretic probability theory.

### A.1  Canonical Partitions of Exogenous Domains

Our proof for Thm. 1 relies on a family of canonical models which any SCM could be reduced to while maintaining counterfactual distributions and the network structure encoded in the induced causal diagram. Fix an endogenous $V \in \boldsymbol{V}$. Given any configuration $U_V = u_V$, the induced function $f_V(\cdot, u_V)$ must correspond to a unique element in the hypothesis class $\mathscr{H}_V$. Naturally, such a mapping leads to a finite partition over the exogenous domain $\Omega_{U_V}$.

**Definition 6.** For an SCM $M = \langle \boldsymbol{V}, \boldsymbol{U}, \boldsymbol{F}, P \rangle$, for each $V \in \boldsymbol{V}$, let functions in $\mathscr{H}_V$ be ordered by $\{h_V^{(i)}\}_{i \in \boldsymbol{I}_V}$ where $\boldsymbol{I}_V = \{1, \ldots, m_V\}, m_V = |\mathscr{H}_V|$. A collection $\left\{\mathcal{U}_V^{(i)}\right\}_{i \in \boldsymbol{I}_V}$ is said to be *canonical partitions* of (exogenous domains of) $V$ if for all $i \in \boldsymbol{I}_V, \mathcal{U}_V^{(i)} = \left\{\forall u_V \mid f_V(\cdot, u_V) = h_V^{(i)}\right\}$.

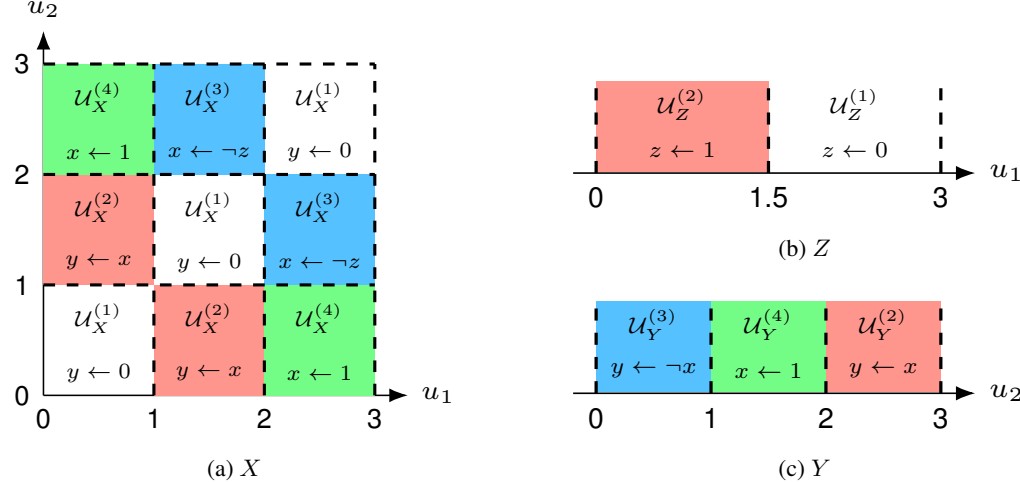

Figure 5: Canonical partitions of exogenous domains of $X, Y$ and $Z$. In (a), each canonical partition $\mathcal{U}_X^{(i)}$ is covered by a finite set of (almost) disjoint cells (e.g., $[2,3] \times [0,1]$).

As $U_V$ varies along its domain, regardless of how complex the variation is, its only effect is to switch the functional relationship between $Pa_V$ and $V$ among elements in the class $\mathscr{H}_V$. Formally,

**Lemma 2.** *For an SCM $M = \langle \boldsymbol{V}, \boldsymbol{U}, \boldsymbol{F}, P \rangle$, for each $V \in \boldsymbol{V}$, $f_V \in \boldsymbol{F}$ could be decomposed as:*

$$f_V(pa_V, u_V) = \sum_{i \in \boldsymbol{I}_V} h_V^{(i)}(pa_V) \mathbb{1}_{u_V \in \mathcal{U}_V^{(i)}}. \tag{21}$$

*Proof.* By the definition of the canonical partitions $\mathcal{U}_V^{(i)}$, $i = 1, \ldots, m_V$, for any $u_V \in \mathcal{U}_V^{(r_V)}$, $f_V(\cdot, u_V) = h_V^{(r_V)}(\cdot)$. Fix $Pa_V = pa_V$. We have $f_V(pa_V, u_V) = h_V^{(r_V)}(pa_V)$. Since $\mathcal{U}_V^{(i)}$, $i = 1, \ldots, m_V$, form a partition over domains $\Omega_{U_V}$, given the same $pa_V, u_V$, the r.h.s. of Eq. (21) must equate to $h_V^{(r_V)}(pa_V)$, which completes the proof. $\square$

As an example, consider an SCM $M$ associated with the "Double bow" graph of Fig. 1b where $X, Y, Z$ are binary variables in $\{0, 1\}$; $U_1, U_2$ are continuous values in $[0, 3]$. More specifically,

$$
\begin{aligned}
&U_i \sim \text{Unif}(0, 3), i = 1, 2, & z &\leftarrow f_Z(u_1) = \mathbb{1}_{u_1 \leq 1.5}, \\
&x \leftarrow f_X(z, u_1, u_2) = \mathbb{1}_{z \leq u_1 \leq z+2} \oplus \mathbb{1}_{z \leq u_2 \leq z+2}, & y &\leftarrow f_Y(x, u_2) = \mathbb{1}_{x \leq u_2 \leq x+2},
\end{aligned} \tag{22}
$$

where $\oplus$ is the "xor" operator. We show in Fig. 5 the canonical partitions induced by functions $f_X, f_Y$ and $f_Z$ respectively. To illustrate, Table 1 describes how the functional mapping between $X$ and $Y$ switches among $\mathcal{H}_Y$ as values of $U_2$ move across canonical partitions.

|  | $0 \leq U_2 < 1$ | $1 \leq U_2 \leq 2$ | $2 < U_2 \leq 3$ |
|---|---|---|---|
| $X = 0$ | $Y = 1$ | $Y = 1$ | $Y = 0$ |
| $X = 1$ | $Y = 0$ | $Y = 1$ | $Y = 1$ |

Table 1: Output of $f_Y(x, u_2)$ in Eq. (22). For any $u_2$, $f_Y(x, u_2)$ never equates to $h_Y^{(1)}(x) = 0$.

The decomposition of Lem. 2 implies that function $f_Y$ could be written as follows:

$$f_Y(x, u_2) = \mathbb{1}_{u_2 \in [0,1)} x + \mathbb{1}_{u_2 \in [1,2]} \neg x + \mathbb{1}_{u_2 \in (2,3]} 1. \tag{23}$$

A natural question as this point is whether one could (1) discretize the exogenous domains of $U_1, U_2$ following canonical partitions of $X, Y, Z$ and (2) replace the original $U_1, U_2$ with a discrete exogenous variable $U$ with cardinality of $2 \times 4 \times 4 = 32$. Fig. 1c shows the causal diagram of the modified discrete SCM. However, such a discretization procedure does not maintain the network structure

of the original causal diagram in Fig. 1b, thus failing to encoding some critical constraints over counterfactual distributions. For instance, variables $Z$ and $Y_x$ are independent since they are solutions of exogenous variables $U_1$ and $U_2$ respectively; $U_1, U_2$ are mutually independent. On the other hand, for any discrete SCM of Fig. 1c, such an independence relationship does not necessarily hold: $Z$ and $Y_x$ could be correlated since they are solutions of the same exogenous variable $U$.

## A.2 Decomposing Canonical Partitions

Previous example calls for a more fine-grained decomposition of canonical partitions. To begin the discussion, we introduce a special type of subdomains called cells.

**Definition 7** (Cell). For an SCM $M = \langle \boldsymbol{V}, \boldsymbol{U}, \boldsymbol{F}, P \rangle$, for each $V \in \boldsymbol{V}$, $\mathcal{R}_V$ is said to be a *cell* in domain $\Omega_{U_V}$ if $\mathcal{R}_V = \times_{U \in U_V} \mathcal{R}_{V,U}$ where $\mathcal{R}_{V,U} \subseteq \Omega_U$, for every $U \in \boldsymbol{U}$.

By definition, for $|U_V| = 1$, any subset of $\Omega_{U_V}$ is a cell (e.g., see Fig. 5). However, it is not always the case when $|U_V| \geq 2$. For instance, $\mathcal{U}_Y^{(4)}$ in Fig. 5a is not a cell. To see this, let $\mathcal{R}_{Y,U_1} = \mathcal{R}_{Y,U_2} = [0,1) \cup (2,3]$. It is verifiable that $\mathcal{U}_Y^{(4)} \neq \mathcal{R}_{Y,U_1} \times \mathcal{R}_{Y,U_2}$ since $\mathcal{R}_{Y,U_1} \times \mathcal{R}_{Y,U_2}$ consists of subsets $[0,1)^2$ and $(2,3]^2$ which is contained in $\mathcal{U}_Y^{(1)4}$.

Arbitrary subsets $A$, $B$ of an event space are said to be *almost disjoint* if their intersection has measure zero, i.e., $P(A \cap B) = 0$. Our next result shows that each canonical partition could be decomposed into a countable union of almost disjoint cells.

**Definition 8** (Covering). For an SCM $M = \langle \boldsymbol{V}, \boldsymbol{U}, \boldsymbol{F}, P \rangle$, for any $V \in \boldsymbol{V}$, let $\mathcal{U}_V$ be an arbitrary subset of $\Omega_{U_V}$. A countable set of cells $\left\{ \mathcal{R}_V^{(j)} \right\}_{j \in \boldsymbol{J}_V}$ is said to be a *covering* of $\mathcal{U}_V$ if (1) for any $i \neq j$, $\mathcal{R}_V^{(i)}$ and $\mathcal{R}_V^{(j)}$ are almost disjoint; (2) $\mathcal{U}_V \subseteq \cup_{j \in \boldsymbol{J}_V} \mathcal{R}_V^{(j)}$; (3) $P(\mathcal{U}_V) = \sum_{j \in \boldsymbol{J}_V} P\left( \mathcal{R}_V^{(j)} \right)$.

**Lemma 3.** *For an SCM $M = \langle \boldsymbol{V}, \boldsymbol{U}, \boldsymbol{F}, P \rangle$, there exists a covering $\left\{ \mathcal{R}_V^{(j)} \right\}_{j \in \boldsymbol{J}_V}$ for each canonical partition $\mathcal{U}_V^{(i)}$, for any $i \in \boldsymbol{I}_V$, any $V \in \boldsymbol{V}$.*

*Proof.* We now consider a stronger statement showing that any subset $\mathcal{U}_V \subseteq \Omega_{U_V}$ has a covering. For any $\mathcal{A} \subseteq \Omega_{U_V}$, define a set of countable collections $\mathcal{C}(\mathcal{A})$ with cells $\mathcal{R}_V \in \Omega_{U_V}$:

$$\mathcal{C}(\mathcal{A}) = \{ \mathcal{C} \subseteq \mathcal{F}_{U_V} \mid \mathcal{C} \text{ is at most countable and } \mathcal{A} \subseteq \cup_{\mathcal{R}_V \in \mathcal{C}} \mathcal{R}_V \}. \tag{24}$$

By definition of product measure $P$ [6, Theorem 9.2], we have:

$$P(\mathcal{U}_V) = \inf \left\{ \sum_{\mathcal{R}_V \in \mathcal{C}} P(\mathcal{R}_V) \mid \forall \mathcal{C} \in \mathcal{C}(\mathcal{U}_V) \right\}. \tag{25}$$

We could thus obtain a countable set $\mathcal{C}$ of cells $\mathcal{R}_V \in \Omega_{U_V}$ such that

$$\mathcal{U}_V \subseteq \cup_{\mathcal{R}_V \in \mathcal{C}} \mathcal{R}_V, \qquad P(\mathcal{U}_V) = \sum_{\mathcal{R}_V \in \mathcal{C}} P(\mathcal{R}_V). \tag{26}$$

What remains is to show that every pair $\mathcal{R}_V^{(i)}, \mathcal{R}_V^{(j)} \in \mathcal{C}$ are almost disjoint. This is equivalent to proving the following statement:

$$P(\cup_{\mathcal{R}_V \in \mathcal{C}} \mathcal{R}_V) = \sum_{\mathcal{R}_V \in \mathcal{C}} P(\mathcal{R}_V). \tag{27}$$

It is sufficient to show that

$$P(\cup_{\mathcal{R}_V \in \mathcal{C}} \mathcal{R}_V) \geq \sum_{\mathcal{R}_V \in \mathcal{C}} P(\mathcal{R}_V). \tag{28}$$

Suppose now the above equating does not hold. There must exist a set $\mathcal{C}' \in \mathcal{C}(\cup_{\mathcal{R}_V \in \mathcal{C}} \mathcal{R}_V)$ such that

$$P(\cup_{\mathcal{R}_V \in \mathcal{C}} \mathcal{R}_V) = \sum_{\mathcal{R}_V \in \mathcal{C}'} P(\mathcal{R}_V) < \sum_{\mathcal{R}_V \in \mathcal{C}} P(\mathcal{R}_V). \tag{29}$$

---

[4]For convenience, we use $[a, b]^2$ to represent the Cartesian product of intervals $[a, b] \times [a, b]$.

609      By the definition of $\mathcal{C}\left(\mathcal{U}_V\right)$ in Eq. (24), we also have $\mathcal{C}' \in \mathcal{C}\left(\mathcal{U}_V\right)$. This means that

$$P\left(\mathcal{U}_V\right) \leq \sum_{\mathcal{R}_V \in \mathcal{C}'} P\left(\mathcal{R}_V\right) < \sum_{\mathcal{R}_V \in \mathcal{C}} P\left(\mathcal{R}_V\right), \tag{30}$$

610      which is a contradiction to Eq. (26). This means that set $\mathcal{C}$ forms a covering $\left\{\mathcal{R}_V^{(j)}\right\}_{j \in \boldsymbol{J}_V}$ over
611      domains of $\mathcal{U}_V$, where $\boldsymbol{J}_V$ is a countable indexing set. $\qquad\square$

612      Consider the partition $\mathcal{U}_X^{(1)}$ in Fig. 5. Let cells $\mathcal{R}_X^{(j)} = [j-1, j]^2$, $j = 1, 2, 3$. It is verifiable that
613      $\mathcal{U}_X^{(1)} \subseteq \cup_{j=1,2,3} \mathcal{R}_X^{(j)}$. Since finite points in $\Omega_{U_1} \times \Omega_{U_2}$ (e.g., $u_1 = u_2 = 1$) has measure zero,

$$P\left(\mathcal{U}_X^{(1)}\right) = P\left((U_1, U_2) \in [0,1)^2 \cup [1,2]^2 \cup (2,3]^2\right) = \sum_{j=1,2,3} P\left(\mathcal{R}_X^{(j)}\right). \tag{31}$$

614      By Def. 8, $\left\{\mathcal{R}_X^{(1)}, \mathcal{R}_X^{(2)}, \mathcal{R}_X^{(3)}\right\}$ is thus a covering of $\mathcal{U}_X^{(1)}$. The characterization of canonical partitions
615      and coverings permits us to decompose counterfactual distributions in the canonical form as follows.

616      **Lemma 4.** *For an SCM $M = \langle \boldsymbol{V}, \boldsymbol{U}, \boldsymbol{F}, P \rangle$, let $\boldsymbol{I} = \times_{V \in \boldsymbol{V}} I_V$. For $\boldsymbol{Y}, \dots, \boldsymbol{Z}, \boldsymbol{X}, \dots, \boldsymbol{W} \subseteq \boldsymbol{V}$[5],*

$$P\left(\boldsymbol{y_x}, \dots, \boldsymbol{z_w}\right) = \sum_{\boldsymbol{i}} \mathbb{1}_{\boldsymbol{Y_x}(\boldsymbol{i})=\boldsymbol{y}} \wedge \cdots \wedge \mathbb{1}_{\boldsymbol{Z_w}(\boldsymbol{i})=\boldsymbol{z}} P\left(\bigwedge_{V \in \boldsymbol{V}} \mathcal{U}_V^{(i)}\right), \tag{32}$$

617      *where variables of the form $\boldsymbol{Y_x}(\boldsymbol{i})$ is defined as:*

$$\boldsymbol{Y_x}(\boldsymbol{i}) = \{Y_{\boldsymbol{x}}(\boldsymbol{i}) \mid \forall Y \in \boldsymbol{Y}\} \text{ where } Y_{\boldsymbol{x}}(\boldsymbol{i}) = \begin{cases} \boldsymbol{x}_Y & \text{if } Y \in \boldsymbol{X} \\ h_Y^{(i)}\left(\{V_{\boldsymbol{x}}(\boldsymbol{i}) \mid V \in Pa_Y\}\right) & \text{otherwise} \end{cases}$$

618      *Moreover, let $\left\{\mathcal{R}_V^{(j)}\right\}_{j \in \boldsymbol{J}_V}$ is a covering of each canonical partition $\mathcal{U}_V^{(i)}$; and let $\boldsymbol{J} = \times_{V \in \boldsymbol{V}} \boldsymbol{J}_V$.*
619      *The above equation could be further written as, for any $\boldsymbol{i} \in \boldsymbol{I}$,*

$$P\left(\bigwedge_{V \in \boldsymbol{V}} \mathcal{U}_V^{(i)}\right) = \sum_{\boldsymbol{j} \in \boldsymbol{J}} P\left(\bigwedge_{V \in \boldsymbol{V}} \mathcal{R}_V^{(j)}\right) = \sum_{\boldsymbol{j} \in \boldsymbol{J}} \prod_{U \in \boldsymbol{U}} P\left(\bigwedge_{V \in ch(U)} \mathcal{R}_{V,U}^{(j)}\right), \tag{33}$$

620      *where $ch(U)$ are child nodes of $U$ in DAG $\mathcal{G}$, i.e., $ch(U) = \{\forall V \in \boldsymbol{V} \mid U \in U_V\}$.*

621      *Proof.* We first show that for any $\boldsymbol{Y}, \boldsymbol{X} \subseteq \boldsymbol{V}$, given any $\boldsymbol{u}, \boldsymbol{x}, *y$,

$$\mathbb{1}_{\boldsymbol{Y_x}(\boldsymbol{u})=\boldsymbol{y}} = \sum_{\boldsymbol{i} \in \boldsymbol{I}} \mathbb{1}_{\boldsymbol{Y_x}(\boldsymbol{i})=\boldsymbol{y}} \prod_{V \in \boldsymbol{V}} \mathbb{1}_{u_V \in \mathcal{U}_V^{(i)}}. \tag{34}$$

622      Let $\mathcal{G}_{\overline{\boldsymbol{X}}}$ be a subgraph obtained from the causal diagram $\mathcal{G}$ by removing all incoming arrows of $\boldsymbol{X}$.
623      For any $Y \in \boldsymbol{Y}$, let $An(Y)_{\mathcal{G}}$ be the set of ancestor nodes of $Y$ in a DAG $\mathcal{G}$, including $Y$ itself. We
624      will prove Eq. (34) by induction on $n = \max_{Y \in \boldsymbol{Y}} \left|An(Y)_{\mathcal{G}_{\overline{\boldsymbol{X}}}}\right|$.

625      **Base Case $n = 1$.** In this case, for $Y \in \boldsymbol{X} \cap \boldsymbol{Y}$, $\mathbb{1}_{Y_{\boldsymbol{x}}(\boldsymbol{u})=y} = \mathbb{1}_{y=\boldsymbol{x}_Y}$ where $\boldsymbol{x}_Y$ be the values
626      assigned to $Y$ in $\boldsymbol{x}$. For $Y \in \boldsymbol{Y} \setminus \boldsymbol{X}$, we must have $Pa_Y = \emptyset$. This implies

$$\mathbb{1}_{Y_{\boldsymbol{x}}(\boldsymbol{u})=y} = \mathbb{1}_{f_Y(u_Y)=y} \tag{35}$$

$$= \mathbb{1}_{y = \sum_{i \in I_Y} h_Y^{(i)} \mathbb{1}_{u_Y \in \mathcal{U}_Y^{(i)}}} \qquad\qquad \text{\# By Lem. 2} \tag{36}$$

$$= \sum_{i \in I_Y} \mathbb{1}_{h_Y^{(i)}=y} \mathbb{1}_{u_Y \in \mathcal{U}_Y^{(i)}} \tag{37}$$

---

[5]For any index sequence $\boldsymbol{i} \in \boldsymbol{I}$, we use $i_V$ to represent the element in $\boldsymbol{i}$ with restriction to $V \in \boldsymbol{V}$. We omit
the subscript $V$ when it is obvious; therefore, $\mathcal{U}_V^{(i)} = \mathcal{U}_V^{(i_V)}$, $h_V^{(i)} = h_V^{(i_V)}$. The same applies to $\boldsymbol{j} \in \boldsymbol{J}$.

The above equation implies

$$\mathbb{1}_{\boldsymbol{Y}_{\boldsymbol{x}}(\boldsymbol{u})=\boldsymbol{y}} = \prod_{Y\in\boldsymbol{Y}\cap\boldsymbol{X}}\mathbb{1}_{y=\boldsymbol{x}_Y}\prod_{Y\in(\boldsymbol{Y}\setminus\boldsymbol{X})}\sum_{i\in I_Y}\mathbb{1}_{h_Y^{(i)}=y}\mathbb{1}_{u_Y\in\mathcal{U}_Y^{(i)}} \tag{38}$$

$$= \sum_{i\in\boldsymbol{I}}\prod_{Y\in\boldsymbol{Y}\cap\boldsymbol{X}}\mathbb{1}_{y=\boldsymbol{x}_Y}\prod_{Y\in(\boldsymbol{Y}\setminus\boldsymbol{X})}\mathbb{1}_{h_Y^{(i)}=y}\prod_{V\in\boldsymbol{V}}\mathbb{1}_{u_V\in\mathcal{U}_V^{(i)}} \tag{39}$$

$$= \sum_{i\in\boldsymbol{I}}\mathbb{1}_{\boldsymbol{Y}_{\boldsymbol{x}}(\boldsymbol{i})=\boldsymbol{y}}\prod_{V\in\boldsymbol{V}}\mathbb{1}_{u_V\in\mathcal{U}_V^{(i)}}. \tag{40}$$

The last step follows from the definition of variables $\boldsymbol{Y}_{\boldsymbol{x}}(\boldsymbol{i})$ given index $\boldsymbol{i}\in\boldsymbol{I}$.

**Induction Case** $n = k + 1$. Assume that Eq. (34) hols for $n = k$. We will prove for the case $n = K + 1$. For $Y\in\boldsymbol{X}\cap\boldsymbol{Y}$, $\mathbb{1}_{Y_{\boldsymbol{x}}(\boldsymbol{u})=y} = \mathbb{1}_{y=\boldsymbol{x}_Y}$. For $Y\in\boldsymbol{Y}\setminus\boldsymbol{X}$, the decomposition in Lem. 2 implies:

$$\mathbb{1}_{Y_{\boldsymbol{x}}(\boldsymbol{u})=y} = \mathbb{1}_{f_Y(\{V_{\boldsymbol{x}}(\boldsymbol{u})|V\in Pa_Y\},u_Y)=y} \tag{41}$$

$$= \mathbb{1}_{y=\sum_{i\in I_Y}h_Y^{(i)}(\{V_{\boldsymbol{x}}(\boldsymbol{u})|V\in Pa_Y\})\mathbb{1}_{u_Y\in\mathcal{U}_Y^{(i)}}} \tag{42}$$

$$= \sum_{i\in I_Y}\sum_{pa_Y}\mathbb{1}_{h_Y^{(i)}(pa_Y)=y}\mathbb{1}_{\{V_{\boldsymbol{x}}(\boldsymbol{u})|V\in Pa_Y\}=pa_Y}\mathbb{1}_{u_Y\in\mathcal{U}_Y^{(i)}}. \tag{43}$$

Since Eq. (34) holds for Case $n = k$, the above equation could be further written as

$$\mathbb{1}_{Y_{\boldsymbol{x}}(\boldsymbol{u})=y} = \sum_{i\in I_Y}\sum_{pa_Y}\mathbb{1}_{h_Y^{(i)}(pa_Y)=y}\mathbb{1}_{u_Y\in\mathcal{U}_Y^{(i)}}\sum_{i\in\boldsymbol{I}}\mathbb{1}_{\{V_{\boldsymbol{x}}(\boldsymbol{i})|V\in Pa_Y\}=pa_Y}\prod_{V\in\boldsymbol{V}}\mathbb{1}_{u_V\in\mathcal{U}_V^{(i)}} \tag{44}$$

$$= \sum_{i\in\boldsymbol{I}}\sum_{pa_Y}\mathbb{1}_{h_Y^{(i)}(pa_Y)=y}\mathbb{1}_{\{V_{\boldsymbol{x}}(\boldsymbol{i})|V\in Pa_Y\}=pa_Y}\prod_{V\in\boldsymbol{V}}\mathbb{1}_{u_V\in\mathcal{U}_V^{(i)}} \tag{45}$$

$$= \sum_{i\in\boldsymbol{I}}\mathbb{1}_{h_Y^{(i)}(\{V_{\boldsymbol{x}}(\boldsymbol{i})|V\in Pa_Y\})=y}\prod_{V\in\boldsymbol{V}}\mathbb{1}_{u_V\in\mathcal{U}_V^{(i)}}. \tag{46}$$

We thus have

$$\mathbb{1}_{\boldsymbol{Y}_{\boldsymbol{x}}(\boldsymbol{u})=\boldsymbol{y}} = \prod_{Y\in\boldsymbol{Y}\cap\boldsymbol{X}}\mathbb{1}_{y=\boldsymbol{x}_Y}\prod_{Y\in(\boldsymbol{Y}\setminus\boldsymbol{X})}\sum_{i\in\boldsymbol{I}}\mathbb{1}_{h_Y^{(i)}(\{V_{\boldsymbol{x}}(\boldsymbol{i})|V\in Pa_Y\})=y}\prod_{V\in\boldsymbol{V}}\mathbb{1}_{u_V\in\mathcal{U}_V^{(i)}} \tag{47}$$

$$= \sum_{i\in\boldsymbol{I}}\prod_{Y\in\boldsymbol{Y}\cap\boldsymbol{X}}\mathbb{1}_{y=\boldsymbol{x}_Y}\prod_{Y\in(\boldsymbol{Y}\setminus\boldsymbol{X})}\mathbb{1}_{h_Y^{(i)}(\{V_{\boldsymbol{x}}(\boldsymbol{i})|V\in Pa_Y\})=y}\prod_{V\in\boldsymbol{V}}\mathbb{1}_{u_V\in\mathcal{U}_V^{(i)}} \tag{48}$$

$$= \sum_{i\in\boldsymbol{I}}\mathbb{1}_{\boldsymbol{Y}_{\boldsymbol{x}}(\boldsymbol{i})=\boldsymbol{y}}\prod_{V\in\boldsymbol{V}}\mathbb{1}_{u_V\in\mathcal{U}_V^{(i)}}. \tag{49}$$

The last step follows from the definition of variables $\boldsymbol{Y}_{\boldsymbol{x}}(\boldsymbol{i})$ given index $\boldsymbol{i}\in\boldsymbol{I}$.

We now consider the proof of Eq. (32). The statement of Eq. (34) implies that for any $\boldsymbol{Y},\dots,\boldsymbol{Z},\boldsymbol{X},\dots,\boldsymbol{W}\subseteq\boldsymbol{V}$,

$$P(\boldsymbol{y}_{\boldsymbol{x}},\dots,\boldsymbol{z}_{\boldsymbol{w}}) = \int_{\Omega_U}\mathbb{1}_{\boldsymbol{Y}_{\boldsymbol{x}}(\boldsymbol{u})=\boldsymbol{y}}\wedge\cdots\wedge\mathbb{1}_{\boldsymbol{Z}_{\boldsymbol{w}}(\boldsymbol{u})=\boldsymbol{z}}dP(\boldsymbol{u}) \tag{50}$$

$$= \int_{\Omega_U}\left(\sum_{i\in\boldsymbol{I}}\mathbb{1}_{\boldsymbol{Y}_{\boldsymbol{x}}(\boldsymbol{i})=\boldsymbol{y}}\prod_{V\in\boldsymbol{V}}\mathbb{1}_{u_V\in\mathcal{U}_V^{(i)}}\right)\wedge\cdots\wedge\left(\sum_{i\in\boldsymbol{I}}\mathbb{1}_{\boldsymbol{Z}_{\boldsymbol{w}}(\boldsymbol{i})=\boldsymbol{z}}\prod_{V\in\boldsymbol{V}}\mathbb{1}_{u_V\in\mathcal{U}_V^{(i)}}\right)dP(\boldsymbol{u}) \tag{51}$$

$$= \int_{\Omega_U}\sum_{i\in\boldsymbol{I}}\mathbb{1}_{\boldsymbol{Y}_{\boldsymbol{x}}(\boldsymbol{i})=\boldsymbol{y}}\wedge\cdots\wedge\mathbb{1}_{\boldsymbol{Z}_{\boldsymbol{w}}(\boldsymbol{i})=\boldsymbol{z}}\prod_{V\in\boldsymbol{V}}\mathbb{1}_{u_V\in\mathcal{U}_V^{(i)}}dP(\boldsymbol{u}) \tag{52}$$

$$= \sum_{i\in\boldsymbol{I}}\mathbb{1}_{\boldsymbol{Y}_{\boldsymbol{x}}(\boldsymbol{i})=\boldsymbol{y}}\wedge\cdots\wedge\mathbb{1}_{\boldsymbol{Z}_{\boldsymbol{w}}(\boldsymbol{i})=\boldsymbol{z}}\int_{\Omega_U}\prod_{V\in\boldsymbol{V}}\mathbb{1}_{u_V\in\mathcal{U}_V^{(i)}}dP(\boldsymbol{u}) \tag{53}$$

$$= \sum_{i\in\boldsymbol{I}}\mathbb{1}_{\boldsymbol{Y}_{\boldsymbol{x}}(\boldsymbol{i})=\boldsymbol{y}}\wedge\cdots\wedge\mathbb{1}_{\boldsymbol{Z}_{\boldsymbol{w}}(\boldsymbol{i})=\boldsymbol{z}}P\left(\bigwedge_{V\in\boldsymbol{V}}\mathcal{U}_V^{(i)}\right). \tag{54}$$

What remains is to prove Eq. (33). We first show that, for any $\mathcal{A} \in \mathcal{F}$,

$$P\left(\mathcal{U}_V^{(i)} \wedge \mathcal{A}\right) = \sum_{j \in J_V} P\left(\mathcal{R}_V^{(i)} \wedge \mathcal{A}\right). \tag{55}$$

Let $\mathcal{A}^\complement = \Omega \setminus \mathcal{A}$. Since $\left\{\mathcal{R}_V^{(j)}\right\}_{j \in J_V}$ is a covering of $\mathcal{U}_V^{(i)}$, we have $\mathcal{U}_V^{(i)} \subseteq \cup_{j \in J_V} \mathcal{R}_V^{(j)}$. This implies

$$P\left(\mathcal{U}_V^{(i)} \wedge \mathcal{A}\right) \leq \sum_{j \in J_V} P\left(\mathcal{R}_V^{(j)} \wedge \mathcal{A}\right), \qquad P\left(\mathcal{U}_V^{(i)} \wedge \mathcal{A}^\complement\right) \leq \sum_{j \in J_V} P\left(\mathcal{R}_V^{(j)} \wedge \mathcal{A}^\complement\right). \tag{56}$$

We will next show that the above inequality relationships are both tight. Suppose say, the inequality in Eq. (55) is strict. We must have

$$P\left(\mathcal{U}_V^{(i)}\right) = P\left(\mathcal{U}_V^{(i)} \wedge \mathcal{A}\right) + P\left(\mathcal{U}_V^{(i)} \wedge \mathcal{A}^\complement\right) \tag{57}$$

$$< \sum_{j \in J_V} P\left(\mathcal{R}_V^{(j)} \wedge \mathcal{A}\right) + \sum_{j \in J_V} P\left(\mathcal{R}_V^{(j)} \wedge \mathcal{A}^\complement\right). \tag{58}$$

The above equation implies

$$P\left(\mathcal{U}_V^{(i)}\right) < \sum_{j \in J_V} P\left(\mathcal{R}_V^{(j)}\right), \tag{59}$$

which is a contradiction. The property of Eq. (55) implies, for any $i \in I$,

$$P\left(\bigwedge_{V \in \boldsymbol{V}} \mathcal{U}_V^{(i)}\right) = \sum_{\boldsymbol{j} \in \boldsymbol{J}} P\left(\bigwedge_{V \in \boldsymbol{V}} \mathcal{R}_V^{(j)}\right). \tag{60}$$

Since each cell $\mathcal{R}_V^{(j)}$ is a Cartesian product of subsets $\times_{U \in U_V} \mathcal{R}_{V,U}^{(j)}$ of each exogenous domains and exogenous variables in $\boldsymbol{U}$ are mutually independent, we must have, for any $\boldsymbol{j} \in \boldsymbol{J}$,

$$P\left(\bigwedge_{V \in \boldsymbol{V}} \mathcal{R}_V^{(j)}\right) = \prod_{U \in \boldsymbol{U}} P\left(\bigwedge_{V \in ch(U)} \mathcal{R}_{V,U}^{(j)}\right). \tag{61}$$

The above equations together prove Eq. (33). $\qquad\square$

Consider again the SCM $M$ described in Eq. (22). Note that the only function in the hypothesis class $\mathcal{H}_Z$ compatible with event $Z = 1$ is $h_Z^{(2)} = 1$. Similarly, event $X_{z=0} = 0, X_{z=1} = 0$ corresponds to the function $h_X^{(1)}(z) = 0$ in $\mathcal{H}_X$. Applying the decomposition of Eq. (32) gives:

$$P\left(Z = 1, X_{z=0} = 0, X_{z=1} = 0\right) = \sum_{i=1,\dots,4} P\left(\mathcal{U}_Z^{(2)} \wedge \mathcal{U}_X^{(1)} \wedge \mathcal{U}_Y^{(i)}\right) = P\left(\mathcal{U}_Z^{(2)} \wedge \mathcal{U}_X^{(1)}\right). \tag{62}$$

Among above quantities, the canonical partition $\mathcal{U}_Z^{(2)} = \{u_1 \in [0, 1.5]\}$ is a cell. $\mathcal{U}_X^{(1)}$ has a covering of $\left\{(u_1, u_2) \in \mathcal{R}_X^{(j)} \mid j = 1, 2, 3\right\}$ where $\mathcal{R}_X^{(j)} = [j-1, j]^2$. Eq. (33) implies

$$P\left(\mathcal{U}_Z^{(2)} \wedge \mathcal{U}_X^{(1)}\right) = \sum_{j=1,2,3} P\left(U_1 \in [0, 1.5] \wedge (U_1, U_2) \in [j-1, j]^2\right)$$
$$= P\left(U_1 \in [0, 1]\right) P\left(U_2 \in [0, 1]\right) + P\left(U_1 \in [1, 1.5]\right) P\left(U_2 \in [1, 2]\right). \tag{63}$$

Computing Eqs. (62) and (63) gives $P\left(Z = 1, X_{z=0} = 0, X_{z=1} = 0\right) = 1/6$. One could verify this answer from the parametrization of SCM $M$ in Eq. (22) using the three-step algorithm introduced in [33] which consists of abduction, action, and prediction.

## A.3 Bounding Cardinalities of Exogenous Domains

The decomposition in Lem. 4 implies a discretization procedure that could reproduce all counterfactual distributions in any SCM $M = \langle \mathbf{V}, \mathbf{U}, \mathbf{F}, P \rangle$. First, we decompose the exogenous domain $\Omega_{U_V}$ for each $V \in \mathbf{V}$ into the canonical partitions. Second, we further decompose each canonical partition using its covering. By doing so, we obtain a partition over the exogenous domain $\Omega_{U_V}$ which consists of countably many (almost) disjoint cells; each cell is assigned with a function (say, $h_V$) in the hypothesis class $\mathcal{H}_V$. Finally, for each configuration $U_V = u_V$, we find the cell partition containing $u_V$ and generate values of $V$ using the associated function $h_V$. We formalize this data-generating process using a canonical family of SCMs described as follows.

**Definition 9.** An SCM $M = \langle \mathbf{V}, \mathbf{U}, \mathbf{F}, P \rangle$ is said to be a *canonical SCM* if for each $V \in \mathbf{V}$, let $\left\{ \mathcal{R}_V^{(j)} \right\}_{j \in \mathbf{J}_V}$ be a covering of $\Omega_{U_V}$; function $f_V \in \mathbf{F}$ is given by, for $i_j \in \{1, \ldots, m_V\}, j \in \mathbf{J}_V$,

$$f_V(pa_V, u_V) = \sum_{j \in \mathbf{J}_V} h_V^{(i_j)}(pa_V) \mathbb{1}_{u_V \in \mathcal{R}_V^{(j)}}. \tag{64}$$

Consider the SCM $M$ described in Eq. (22) as an example. Let $N$ be a canonical SCM compatible with the DAG of Fig. 1b; its covering cells (e.g., $\mathcal{R}_X^{(j)}$) and corresponding functions ($h_X^{(j_i)}(z)$) associated with $X, Y, Z$ are graphically described in Fig. 5 respectively. It immediately follows from Lem. 4 that $M$ and $N$ generate the same collection of counterfactual distributions $\mathbf{P}^*$.

**Lemma 5.** *For a DAG $\mathcal{G}$, let $M$ be an arbitrary SCM compatible with $\mathcal{G}$. There exists a canonical SCM $N$ compatible with $\mathcal{G}$ such that $\mathbf{P}_M^* = \mathbf{P}_N^*$, i.e., they coincide in all counterfactual distributions.*

*Proof.* For each $V \in \mathbf{V}$ in SCM $M$, let $\left\{ \mathcal{R}_V^{(j)} \right\}_{j \in \mathbf{J}_V^{(i)}}$ denote a covering for a canonical partition $\mathcal{U}_V^{(i)}, i \in \mathbf{I}_V$. Since $\{ \mathcal{U}_V^{(i)} \}_{i \in \mathbf{I}_V}$ forms a partition over the exogenous domain $\Omega_{U_V}$. The collection $\left\{ \mathcal{R}_V^{(j)} \mid j \in \mathbf{J}_V^{(i)}, V \in \mathbf{V} \right\}$ forms a covering over $\Omega_{U_V}$. Let $\mathbf{J}_V$ be the union of indexing set $\cup_{i \in \mathbf{I}_V} \mathbf{J}_V^{(i)}$. Naturally, any element $j \in \mathbf{J}_V$ must belong to a subset $\mathbf{J}_V^{(i)}$; let $i_j$ denote such index $i$. We construct a canonical SCM $N$ using coverings $\left\{ \mathcal{R}_V^{(j)} \right\}_{j \in \mathbf{J}_V}$ and index $i_j$ described previously. Let $\mathbf{J} = \times_{V \in \mathbf{V}} \mathbf{J}_V$. For any $\mathbf{Y}, \ldots, \mathbf{Z}, \mathbf{X}, \ldots, \mathbf{W} \subseteq \mathbf{V}$, the counterfactual distribution $P(\mathbf{y_x}, \ldots, \mathbf{z_w})$ in the canonical SCM $N$ is equal to

$$P(\mathbf{y_x}, \ldots, \mathbf{z_w}) = \sum_{\mathbf{j} \in \mathbf{J}} \mathbb{1}_{\mathbf{Y_x}(\mathbf{i_j}) = \mathbf{y}} \wedge \cdots \wedge \mathbb{1}_{\mathbf{Z_w}(\mathbf{i_j}) = \mathbf{z}} P \left( \bigwedge_{V \in \mathbf{V}} \mathcal{R}_V^{(j)} \right), \tag{65}$$

where $\mathbf{i_j}$ is the indexing sequence $(i_j)_{j \in \mathbf{j}}$. Lem. 4, together with some reordering over indices in $\mathbf{i_j}$, implies that $M$ and $N$ induce the same collection of counterfactual distributions. $\square$

Given a canonical SCM, one could immediately obtain a discrete SCM by discretizing exogenous domains following the covering cells. Since each cell is a Caresian product of subsets (Def. 7), the resulting discrete model must induce a causal diagram with the same network structure.

**Lemma 6.** *For a DAG $\mathcal{G}$, consider the following conditions: (1) $\mathcal{M}$ is the set of all SCMs compatible with $\mathcal{G}$; (2) $\mathcal{N}$ is the set of all discrete SCMs compatible with $\mathcal{G}$. Then, $\mathcal{M}$ and $\mathcal{N}$ are counterfactually equivalent.*

*Proof.* For any cell $\mathcal{R}_V^{(j)} = \times_{U \in U_V} \mathcal{R}_{V,U}^{(j)}$, we call $\mathcal{R}_{V,U}^{(j)}$ the projection of $\mathcal{R}_V^{(j)}$ to domains of $U$. We will describe a discretization procedure that discretize domains of each $U \in \mathbf{U}$ following the intersections of projections $\cap_{V \in ch(U)} \mathcal{R}_{V,U}^{(j)}, \forall j \in \mathbf{J}_V$. For each $V \in ch(U)$, for any infinite binary sequence $r_{V,U} \in \{0,1\}^{\mathbf{J}_V}$, let an event $\mathcal{A}_{r_{V,U}^{(j)}} \in \mathcal{F}_{U_k}$ be, for $j \in \mathbf{J}_V$,

$$\mathcal{A}_{r_{V,U}^{(j)}} = \begin{cases} \mathcal{R}_{V,U}^{(j)} & \text{if } r_{V,U}^{(j)} = 1 \\ \Omega_U \setminus \mathcal{R}_{V,U}^{(j)} & \text{if } r_{V,U}^{(j)} = 0. \end{cases} \tag{66}$$

For any $r_U = \{r_{V,U} : V \in ch(U)\}$, let a subset $\mathcal{A}_{r_U} \in \Omega_U$ be

$$\mathcal{A}_{r_U} = \bigcap_{V \in ch(U)} \bigcap_{j \in \boldsymbol{J}_V} \mathcal{A}_{r_{V,U}^{(j)}}. \tag{67}$$

Since $\mathcal{A}_{r_{V,U}}, \forall r_U$, enumerates all possible intersections of projections $\mathcal{R}_{V,U}^{(j)}$, we could obtain probabilities over any intervention $\cap_{V \in ch(U)} \mathcal{R}_{V,U}^{(j)}$ using the join probability $P(\mathcal{A}_{r_U})$.

It now suffices to show that distribution $P(\mathcal{A}_{r_U})$ has countable support, i.e., the set $\mathcal{A}_U = \{\mathcal{A}_{r_U} : P(\mathcal{A}_{r_U}) > 0\}$ has at most countably elements. Since $P$ is a probability measurable, $P(\mathcal{A}_{r_k}) \in [0,1]$. By the construction of Eq. (66), we must have $\sum_{r_U} P(\mathcal{A}_{r_U}) = 1$. If the sum over an uncountable set of reals is finite, then there exist at most countable number of events $\mathcal{A}_{r_U}$ such that $P(\mathcal{A}_{r_U}) > 0$, i.e, the set $\mathcal{A}_U$ is countable. $\qquad\square$

Lem. 6 implies that one could represent all counterfactual distributions in a causal diagram using a countably infinite number of exogenous states. To prove Thm. 1, what remains is to bound the cardinality of the exogenous domain. More specifically, we will show that any discrete SCM $M$ with cardinality $|\Omega_U| > \prod_{V \in \boldsymbol{C}_U} |\mathscr{H}_V|, \forall U \in \boldsymbol{U}, \boldsymbol{C}_U$ is the c-component that contains all child nodes of $U$, can be modified into a discrete SCM $N$ with $|\Omega_U| \leq \prod_{V \in \boldsymbol{C}_U} |\mathscr{H}_V|, \forall U \in \boldsymbol{U}$, while maintaining all counterfactual distributions $\boldsymbol{P}^*$ and the same network structure in the causal diagram.

**Theorem 1.** *For a DAG $\mathcal{G}$, consider the following conditions[6]: (1) $\mathscr{M}$ is the set of all SCMs compatible with $\mathcal{G}$; (2) $\mathscr{N}$ is the set of all discrete SCMs compatible with $\mathcal{G}$ where for every $U \in \boldsymbol{U}$, its cardinality $|\Omega_U| = \prod_{V \in \boldsymbol{C}_U} |\Omega_{Pa_V} \mapsto \Omega_V|$, i.e., the number of functions mapping from $Pa_V$ to $V$ for every variable $V$ in the c-component $\boldsymbol{C}_U$. Then, $\mathscr{M}$ and $\mathscr{N}$ are counterfactually equivalent.*

*Proof.* Lem. 4 implies that it suffices to prove that for any discrete SCM $M \in \mathcal{M}$, there exists a finite SCM $N \in \mathcal{N}$ such that $M$ and $N$ coincide in the joint distribution over canonical partitions $P\left(\bigwedge_{V \in \boldsymbol{V}} \mathcal{U}_V^{(i)}\right)$. C-components in $\mathcal{C}(\mathcal{G})$ implies the following decomposition

$$P\left(\bigwedge_{V \in \boldsymbol{V}} \mathcal{U}_V^{(i)}\right) = \prod_{\boldsymbol{C} \in \mathcal{C}(\mathcal{G})} P\left(\bigwedge_{V \in \boldsymbol{C}} \mathcal{U}_V^{(i)}\right). \tag{68}$$

We now focus on the consistency for the joint probability $P\left(\bigwedge_{V \in \boldsymbol{C}} \mathcal{U}_V^{(i)}\right)$ for each $\boldsymbol{C} \in \mathcal{C}(\mathcal{G})$.

Fix a c-component $\boldsymbol{C}$. Let $\vec{P}$ be a vector representing probabilities of $\left(P\left(\bigwedge_{V \in \boldsymbol{C}} \mathcal{U}_V^{(i)}\right)\right)_{\boldsymbol{i} \in \boldsymbol{I}}$, which could be seen as a point in $d-1$-dimensional real space where $d = \prod_{V \in \boldsymbol{C}} |\mathscr{H}_V|$[7]. Let $U_{\boldsymbol{C}}$ denote the collection $\cup_{V \in \boldsymbol{C}} U_V$. Fix an exogenous $U \in U_{\boldsymbol{C}}$. Let $P_u\left(\bigwedge_{V \in \boldsymbol{C}} \mathcal{U}_V^{(i)}\right)$ denote joint distributions over canonical partitions when $U$ is fixed as a constant $u \in \Omega_U$. More specifically,

$$P_u\left(\bigwedge_{V \in \boldsymbol{C}} \mathcal{U}_V^{(i)}\right) = \sum_{\boldsymbol{u} \setminus u} \prod_{V \in \boldsymbol{C}} \mathbb{1}_{u_v \in \mathcal{U}_V^{(i)}} \prod_{U' \in (\boldsymbol{U} \setminus U)} P(u'). \tag{69}$$

Similarly, let $\vec{P}_u$ be a vector in $\mathbb{R}^{d-1}$ representing probabilities of $P_u\left(\bigwedge_{V \in \boldsymbol{C}} \mathcal{U}_V^{(i)}\right)$. By basic probabilistic operations, we must have $\vec{P} = \sum_u \vec{P}_u P(u)$. That is, $\vec{P} \in \mathbb{R}^{d-1}$ is a point lies in the convex hull of a set $\left\{\vec{P}_u \mid \forall u \in \Omega_U\right\}$. The Carathéodory theorem [9, 13] implies that we could write $\vec{P}$ as a convex combination of at most $d$ points in $\left\{\vec{P}_u \mid \forall u \in \Omega_U\right\}$. That is, for $d$ distinct values $\{u_1, \ldots, u_d\}$ in $\Omega_U$,

$$\vec{P} = \sum_{k=1}^{d} w_d \vec{P}_{u_k}, \qquad \text{where } w_k > 0, \forall k = 1, \ldots, d, \text{ and } \sum_k w_k = 1. \tag{70}$$

---

[6]For every $V \in \boldsymbol{V}, \Omega_{Pa_V} \mapsto \Omega_V$ is the set of all functions mapping from domains $\Omega_{Pa_V}$ to $\Omega_V$.

[7]By definition, $\vec{P}$ is a vector with $d = \prod_{V \in \boldsymbol{C}} |\mathscr{H}_V|$ elements. Since $\sum_{\boldsymbol{i}} P\left(\bigwedge_{V \in \boldsymbol{C}} \mathcal{U}_V^{(i)}\right) = 1$, it only takes a vector with $d-1$ dimensions to uniquely determine $\vec{P}$.

We could replace $P(u)$ with a distribution $P'(u_k) = w_k$ over a finite discrete domain $\Omega'_U = \{u_1, \ldots, u_d\}$ and obtain a discrete SCM $N$ that reproduce all counterfactual distributions in $M$ with cardinality $|\Omega_U| \leq \prod_{V \in \boldsymbol{C}_U} |\mathcal{H}_V|$ for a fixed $U \in \boldsymbol{U}$. Finally, we complete the proof by repeatedly applying this replacement for every $U \in \boldsymbol{U}$. $\qquad\square$

## A.4 Partial identification of Counterfactual Distributions

To demonstrate the expressive power of discrete SCMs, we investigate the problem of partial identification of counterfactual distributions. For an SCM $M^* = \langle \boldsymbol{V}, \boldsymbol{U}, \boldsymbol{F}, P \rangle$, we are interested in evaluating an arbitrary counterfactual probability $P(\boldsymbol{y_x}, \ldots, \boldsymbol{z_w})$. The detailed parametrization of $M^*$ is unknown. Instead, the learner only has access to the causal diagram $\mathcal{G}$ and the observational distribution $P(\boldsymbol{v})$ induced by $M^*$. Our goal is to derive an informative bound $[l, r]$ from the combination of $\mathcal{G}$ and $P(\boldsymbol{v})$ that contains the actual counterfactual probability $P(\boldsymbol{y_x}, \ldots, \boldsymbol{z_w})$.

Let $\mathcal{N}$ denote the family of discrete SCMs defined in Thm. 1 which are compatible with the causal diagram $\mathcal{G}$. We derive a bound $[l, r]$ over $P(\boldsymbol{y_x}, \ldots, \boldsymbol{z_w})$ from the observational data $P(\boldsymbol{v})$ by solving the optimization problem in Eq. (6). It follows immediately from Thm. 1 that the solution $[l, r]$ of the optimization problem Eq. (6) is guaranteed to be a tight bound over the unknown counterfactual $P(\boldsymbol{y_x}, \ldots, \boldsymbol{z_w})$.

**Corollary 1** (Soundness). *Given a DAG $\mathcal{G}$ and an observational distribution $P(\boldsymbol{v})$, let $\mathcal{M}$ be the set of all SCMs compatible with $\mathcal{G}$ and let $\mathcal{M}_o = \{\forall M \in \mathcal{M} \mid P_M(\boldsymbol{v}) = P(\boldsymbol{v})\}$. For the solution $[l, r]$ of Eq. (6), $P_M(\boldsymbol{y_x}, \ldots, \boldsymbol{z_w}) \in [l, r]$ for any SCM $M \in \mathcal{M}_o$.*

*Proof.* Without loss of generality, we assume $\mathcal{M}_o \neq \emptyset$, i.e., $\mathcal{G}$ and $P(\boldsymbol{v})$ are compatible. For any $M \in \mathcal{M}_o$, Thm. 1 implies that there exists a discrete $N \in \mathcal{N}$ such that $P_N(\boldsymbol{v}) = P_M(\boldsymbol{v}) = P(\boldsymbol{v})$ and $P_N(\boldsymbol{y_x}, \ldots, \boldsymbol{z_w}) = P_M(\boldsymbol{y_x}, \ldots, \boldsymbol{z_w})$. The optimization problem of Eq. (6) ensures $P_N(\boldsymbol{y_x}, \ldots, \boldsymbol{z_w}) \in [l, r]$, which completes the proof. $\qquad\square$

**Corollary 2** (Tightness). *Given a DAG $\mathcal{G}$ and an observational distribution $P(\boldsymbol{v})$, let $\mathcal{M}$ be the set of all SCMs compatible with $\mathcal{G}$ and let $\mathcal{M}_o = \{\forall M \in \mathcal{M} \mid P_M(\boldsymbol{v}) = P(\boldsymbol{v})\}$. For the solution $[l, r]$ of Eq. (6), there exist SCMs $M_1, M_2 \in \mathcal{M}_o$ such that $P_{M_1}(\boldsymbol{y_x}, \ldots, \boldsymbol{z_w}) = l$, $P_{M_2}(\boldsymbol{y_x}, \ldots, \boldsymbol{z_w}) = r$.*

*Proof.* Let $\mathcal{N}_o = \{\forall N \in \mathcal{N} \mid P_N(\boldsymbol{v}) = P(\boldsymbol{v})\}$. The optimization problem of Eq. (6) ensures that there exist discrete SCMs $N_1, N_2 \in \mathcal{N}_o$ such that $P_{N_1}(\boldsymbol{y_x}, \ldots, \boldsymbol{z_w}) = l$, $P_{N_2}(\boldsymbol{y_x}, \ldots, \boldsymbol{z_w}) = r$. For any $N_i, i = 1, 2$, Thm. 1 implies that one could find an SCM $M_i \in \mathcal{M}_o$ such that $P_{M_i}(\boldsymbol{y_x}, \ldots, \boldsymbol{z_w}) = P_{N_i}(\boldsymbol{y_x}, \ldots, \boldsymbol{z_w})$. This completes the proof. $\qquad\square$

## A.5 Acyclic Directed Mixed Graphs

In the causal inference literature [43, 45], a causal diagram could also be represented by an acyclic directed mixed graph (ADMG), where exogenous variables are not explicitly shown. Formally, an ADMG associated with an SCM $M = \langle \boldsymbol{V}, \boldsymbol{U}, \boldsymbol{F}, P \rangle$ is an augmented DAG where nodes represent $\boldsymbol{V}$; arrows represent arguments $Pa_V$ of each function $f_V$; and a bi-directed arrow between nodes $V_i$ and $V_j$ indicates the presence of unobserved confounders (UCs) affecting both $V_i$ and $V_j$, i.e., $U_{V_i} \cap U_{V_j} \neq \emptyset$[8]. For instance, Fig. 6b shows an ADMG compatible with SCMs described in Fig. 6a. Similarly, it is also compatible with SCMs graphically described in Fig. 6c. That is, an ADMG describes an equivalence class of DAGs (more than 1). [43, Def. 5] introduce an algorithm to project a DAG to an ADMG which maintains the same causal relationships over endogenous variables.

We will study an inverse algorithm that translates an ADMG into a DAG while maintaining all counterfactual distributions. Our construction rests on a novel object called the *confounded clique*.

**Definition 10** (c-clique). For an ADMG $\mathcal{G}$, a subset $\boldsymbol{C} \subseteq \boldsymbol{V}$ is a c-clique if any pair $V_i, V_j \in \boldsymbol{C}$ is connected by a *bi-directed arrow* in $\mathcal{G}$, i.e., $V_i \leftrightarrow V_j \in \mathcal{G}$.

---

[8]The definition of ADMG used here differs from the one studied in [15]. According to [15], the ADMG in Fig. 6b uniquely corresponds to the DAG in Fig. 6a; the ADMG for the DAG of Fig. 6c is not defined.

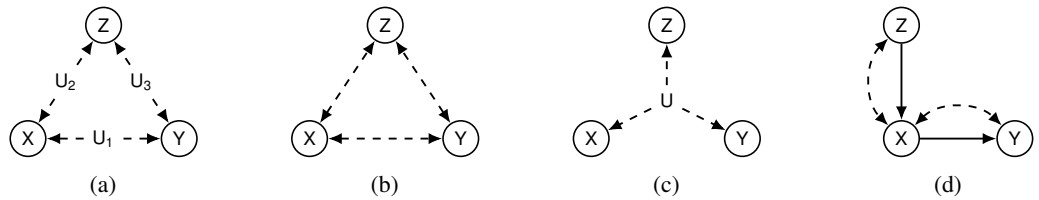

Figure 6: DAGs (a, c) containing a treatment $X$, an outcome $Y$, and a covariate $Z$; and (b) their corresponding ADMG; (d) an ADMG that is counterfactually equivalent to the DAG in Fig. 1b.

A c-clique $C$ in $\mathcal{G}$ is *maximal* if there exists no other c-clique that contains $C$. We denote by $c(\mathcal{G})$ the set of all maximal c-cliques in an ADMG $\mathcal{G}$. For instance, the ADMG of Fig. 6c has a single c-clique $C = \{X, Y, Z\}$. Fig. 6d contains two c-cliques $C_1 = \{X, Z\}$ and $C_2 = \{X, Y\}$; while it only contains a single c-component $\{X, Y, Z\}$.

Our algorithm INVERSEPROJECT, described in Alg. 2, translates an ADMG into a DAG by replacing bi-directed arrows in each c-clique with arrows from a new exogenous variable. As an

---

**Algorithm 2:** INVERSEPROJECT

1: **Input:** An ADMG $\mathcal{G}$
2: **Output:** A DAG $\mathcal{H}$.
3: Let $\mathcal{H} = \mathcal{G}$.
4: **for** each c-clique $C$ in $c(\mathcal{G})$ **do**
5:     For every pair $V_i, V_j \in C$, remove $V_i \leftrightarrow V_j$ from $\mathcal{H}$.
6:     Add an exogenous node $U$ in $\mathcal{H}$.
7:     For every $V \in C$, add $U \to V$ in $\mathcal{H}$.
8: **end for**

---

example, Fig. 6c shows an DAG obtained from the ADMG of Fig. 6b where exogenous variable $U$ corresponds to the c-clique $C = \{X, Y, Z\}$. Fig. 1b shows a DAG obtained from applying INVERSEPROJECT to the ADMG of Fig. 6d. The following proposition shows that INVERSEPROJECT constructs a DAG that generates the same counterfactual distributions in the given ADMG.

**Lemma 7.** *For an ADMG $\mathcal{G}$, let $\mathcal{H}$ be a DAG obtained from INVERSEPROJECT($\mathcal{G}$), consider the following conditions: (1) $\mathcal{M}$ is the set of all SCMs associated with $\mathcal{G}$; (2) $\mathcal{N}$ is the set of all SCMs associated with $\mathcal{H}$. Then $\mathcal{M}$ and $\mathcal{N}$ are counterfactually equivalent.*

*Proof.* By the definition of ADMGs, a backdoor path $V_i \leftarrow U_k \to V_j \in \mathcal{H}$ indicates the presence of a bi-directed arrow $V_i \leftrightarrow V_j \in \mathcal{G}$. Therefore, any SCM $N$ compatible with the DAG $\mathcal{H}$ is also compatible with the ADMG $\mathcal{G}$. That is, $N \in \mathcal{N}$ implies $N \in \mathcal{M}$.

It suffices to show that for any SCM $M$ compatible with the ADMG $\mathcal{G}$, there exists an SCM $N$ compatible the DAG $\mathcal{H}$ such that for any $\boldsymbol{X} \subset \boldsymbol{V}$, $P_M(\boldsymbol{v}|\mathrm{do}(\boldsymbol{x})) = P_N(\boldsymbol{v}|\mathrm{do}(\boldsymbol{x}))$. Let $c^2$-components $c(\mathcal{G}) = \{\boldsymbol{C}_1, \ldots, \boldsymbol{C}_n\}$. We will construct a partition $\tilde{U}_1, \ldots, \tilde{U}_n$ over exogenous variables $\boldsymbol{U}$ in $M$. Let $\tilde{U}_1 = \cup_{V \in \boldsymbol{C}_i} U_V$ and $\tilde{U}_i = \cup_{V \in \boldsymbol{C}_i} U_V \setminus \left( \cup_{j<i} \tilde{U}_i \right)$ for $i = 2, \ldots, n$. By construction, we must have $\tilde{U}_i \subseteq \cup_{V \in \boldsymbol{C}_i} U_V$. Finally, we obtain an SCM $N$ compatible with DAG $\mathcal{H}$ by (1) simply grouping exogenous variables $\boldsymbol{U}$ in $M$ into the partition $\tilde{\boldsymbol{U}} = \{\tilde{U}_1, \ldots, \tilde{U}_n\}$ and (2) use $\tilde{\boldsymbol{U}}$ as the exogenous variables in the modified model $N$. Since structural functions $\boldsymbol{F}$ and exogenous distribution $P$ remain the same, $M$ and $N$ must coincide in all counterfactual distributions. $\square$

To characterize counterfactual distributions in an ADMG $\mathcal{G}$, we could apply procedure INVERSEPROJECT to obtain a DAG $\mathcal{H}$. Lem. 7 and Thm. 1 imply that one could assume exogenous variables in $\mathcal{G}$ to be exogenous variables in $\mathcal{H}$ with finite domains, without loss of generality.

 **B    Monte Carlo Estimation of Credible Intervals**

In this section, we provide proofs for the large deviation bounds for empirical estimates of $100(1 - \alpha)\%$ credible intervals introduced in Sec. 3.2.

**Lemma 1.** *Fix $T > 0$ and $\delta \in (0, 1)$. Let function $f(T, \delta) = \sqrt{2T^{-1} \ln(4/\delta)}$. With probability at least $1 - \delta$, estimators $\hat{l}_\alpha(T), \hat{r}_\alpha(T)$ for any $\alpha \in [0, 1)$ is bounded by*

$$\hat{l}_\alpha(T) \in \left[l_{\alpha - f(T,\delta)}, l_{\alpha + f(T,\delta)}\right], \qquad \hat{r}_\alpha(T) \in \left[r_{\alpha + f(T,\delta)}, r_{\alpha - f(T,\delta)}\right]. \tag{17}$$

*Proof.* Fix $\epsilon > 0$. If $\hat{l}_\alpha(T) > l_{\alpha + \epsilon}$, this means that there are at most $\lceil(\alpha/2)T\rceil - 1$ instances in $\left\{\theta_{\text{ctf}}^{(t)}\right\}_{t=1}^T$ that are smaller than or equal to $l_{\alpha + \epsilon}$. That is,

$$P\left(\hat{l}_\alpha(T) > l_{\alpha + \epsilon}\right) \leq P\left(\sum_{t=1}^T \mathbb{1}_{\theta_{\text{ctf}}^{(t)} \leq l_{\alpha + \epsilon}} \leq \lceil(\alpha/2)T\rceil - 1\right) \tag{71}$$

$$\leq P\left(\sum_{t=1}^T \mathbb{1}_{\theta_{\text{ctf}}^{(t)} \leq l_{\alpha + \epsilon}} \leq (\alpha/2)T\right) \tag{72}$$

$$\leq P\left(\frac{1}{T}\sum_{t=1}^T \mathbb{1}_{\theta_{\text{ctf}}^{(t)} \leq l_{\alpha + \epsilon}} \leq \frac{\alpha + \epsilon}{2} - \frac{\epsilon}{2}\right) \tag{73}$$

$$\leq \exp\left(-\frac{T\epsilon^2}{2}\right). \tag{74}$$

The last step in the above equation follows from the standard Hoeffding's inequality.

If $\hat{l}_\alpha(T) < l_{\alpha - \epsilon}$, this implies that there are at least $\lceil(\alpha/2)T\rceil$ instances in $\left\{\theta_{\text{ctf}}^{(t)}\right\}_{t=1}^T$ that are larger than or equal to $l_{\alpha + \epsilon}$. That is,

$$P\left(\hat{l}_\alpha(T) < l_{\alpha - \epsilon}\right) \leq P\left(\sum_{t=1}^T \mathbb{1}_{\theta_{\text{ctf}}^{(t)} \leq l_{\alpha - \epsilon}} \geq \lceil(\alpha/2)T\rceil\right) \tag{75}$$

$$\leq P\left(\sum_{t=1}^T \mathbb{1}_{\theta_{\text{ctf}}^{(t)} \leq l_{\alpha - \epsilon}} \geq (\alpha/2)T\right) \tag{76}$$

$$\leq P\left(\frac{1}{T}\sum_{t=1}^T \mathbb{1}_{\theta_{\text{ctf}}^{(t)} \leq l_{\alpha - \epsilon}} \geq \frac{\alpha - \epsilon}{2} + \frac{\epsilon}{2}\right) \tag{77}$$

$$\leq \exp\left(-\frac{T\epsilon^2}{2}\right). \tag{78}$$

The last step follows from the standard Hoeffding's inequality. Similarly, we could also show that

$$P\left(\hat{h}_\alpha(T) < h_{\alpha + \epsilon}\right) \leq \exp\left(-\frac{T\epsilon^2}{2}\right), \qquad P\left(\hat{h}_\alpha(T) > h_{\alpha - \epsilon}\right) \leq \exp\left(-\frac{T\epsilon^2}{2}\right). \tag{79}$$

Finally, bounding the error rate by $\delta/4$ gives:

$$\exp\left(-\frac{T\epsilon^2}{2}\right) = \frac{\delta}{4} \Rightarrow \epsilon = \sqrt{2T^{-1}\ln(4/\delta)}. \tag{80}$$

Replacing the error rate $\epsilon$ with $f(T, \delta) = \sqrt{2T^{-1}\ln(4/\delta)}$ completes the proof. $\qquad\square$

**Corollary 3.** *Fix $\delta \in (0, 1)$ and $\epsilon > 0$. With probability at least $1 - \delta$, the interval $[\hat{l}, \hat{r}] = $ CREDIBLEINTERVAL$(\alpha, \delta, \epsilon)$ for any $\alpha \in [0, 1)$ is bounded by $\hat{l} \in [l_{\alpha - \epsilon}, l_{\alpha + \epsilon}]$ and $\hat{r} \in [r_{\alpha + \epsilon}, r_{\alpha - \epsilon}]$.*

*Proof.* The statement follows immediately from Lem. 1 by setting $\sqrt{2T^{-1}\ln(4/\delta)} \leq \epsilon$. $\qquad\square$

# C   Simulation Setups and Additional Experiments

In this section, we will provide details on the simulation setups and preprocessing of datasets. We also conduct additional experiments on other more involved causal diagrams and using skewed hyperparameters for prior distributions. For all experiments, we will focus on stick-breaking priors in Eq. (8) with hyperparameters $\alpha_U^{(u)} = \alpha_U/d_U$ and $\beta_U^{(u)} = (d_U - u)\alpha_U/d_U$ for some real $\alpha_U > 0$. This is equivalent to drawing probabilities $\theta_U = \{\theta_u \mid \forall u\}$ from a Dirichlet distribution defined as:

$$\theta_U \sim \texttt{Dirichlet}\left(\frac{\alpha_1}{d_U}, \cdots, \frac{\alpha_{d_U}}{d_U}\right), \text{ where } \alpha_i = \alpha_U, \forall i = 1, \ldots, d_U. \tag{81}$$

All experiments were performed on a computer with 32GB memory, implemented in MATLAB. We are in the process of translating the source code to other open-source platforms (e.g., Julia). We will release them if the paper is accepted.

**Experiment 1: Frontdoor**   We collect $N = 10^4$ observational data $\bar{V} = \{X^{(n)}, Y^{(n)}, W^{(n)}\}_{n=1}^N$ from an SCM compatible with the "Frontdoor" graph in Fig. 3, defined as follows:

$$\begin{aligned}
&U_1 \sim \texttt{Unif}(0,1), \quad U_2 \sim \mathcal{N}(0,1),\\
&X \sim \texttt{Binomial}(1, p_X), \text{ where } p_W = U_1,\\
&W \sim \texttt{Binomial}(1, p_W), \text{ where } p_W = \frac{1}{1 + \exp(-X - U_2)},\\
&Y \sim \texttt{Binomial}(1, p_Y), \text{ where } p_Y = \frac{1}{1 + \exp(W - U_1)}.
\end{aligned} \tag{82}$$

In this experiment, we set hyperparameters $\alpha_{U_1} = d_{U_1} = 8$ and $\alpha_{U_1} = d_{U_2} = 4$.

**Experiment 2: Instrumental Variables (IV)**   We collect $N = 10^4$ observational samples $\bar{V} = \{X^{(n)}, Y^{(n)}, Z^{(n)}\}_{n=1}^N$ from an SCM compatible with the "IV" graph in Fig. 1a, defined as follows:

$$\begin{aligned}
&U_1 \sim \mathcal{N}(0,1), \quad U_2 \sim \mathcal{N}(0,1),\\
&Z \sim \texttt{Binomial}(1, p_Z), \text{ where } p_Z = \frac{1}{1 + \exp(-U_1)},\\
&X \sim \texttt{Binomial}(1, p_X), \text{ where } p_X = \frac{1}{1 + \exp(-Z - U_2)},\\
&Y \sim \texttt{Binomial}(1, p_Y), \text{ where } p_Y = \frac{1}{1 + \exp(X - U_2 + 0.5)}.
\end{aligned} \tag{83}$$

In this experiment, we set hyperparameters $\alpha_{U_1} = d_{U_1} = 2$ and $\alpha_{U_1} = d_{U_2} = 16$.

**Experiment 3: Probability of Necessity and Sufficiency (PNS)**   We collect $N = 10^4$ observational samples $\bar{V} = \{X^{(n)}, Y^{(n)}\}_{n=1}^N$ from an SCM compatible with the "Bow" graph in Fig. 1d, defined as follows:

$$\begin{aligned}
&U \sim \mathcal{N}(0,1), \quad E \sim \texttt{Logistic}(0,1)\\
&X \sim \texttt{Binomial}(1, p_X), \text{ where } p_X = \frac{1}{1 + \exp(U)},\\
&Y \leftarrow \mathbb{1}_{X - U + E + 0.1 > 0}.
\end{aligned} \tag{84}$$

In this experiment, we set hyperparameters $\alpha_U = d_U = 8$.

**Experiment 4: International Stroke Trials (IST)**   IST was a large, randomized, open trial of up to 14 days of antithrombotic therapy after stroke onset [10]. The aim was to provide reliable evidence on the efficacy of aspirin and of heparin. The dataset is released under Open Data Commons Attribution License (ODC-By). In particular, the treatment $X$ is a pair $(i,j)$ where $i = 0$ stands for no aspirin allocation, 1 otherwise; $j = 0$ stands for no heparin allocation, 1 for median-dosage, and 2 for high-dosage. The primary outcome $Y \in \{0, \ldots, 3\}$ is the health of the patient 6 months after the treatment, where 0 stands for death, 1 for being dependent on the family, 2 for the partial recovery, and 3 for the full recovery.

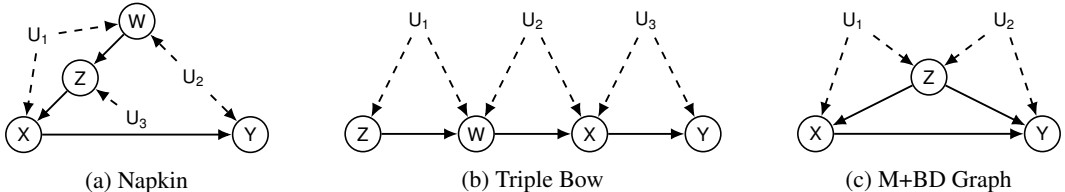

<p style="text-align:center">(a) Napkin     (b) Triple Bow     (c) M+BD Graph</p>

Figure 7: DAGs for Experiment 5 (a), Experiment 7 (b), and Experiment 8 (d), containing a treatment $X$, an outcome $Y$, ancestors $Z, W$, and exogenous variables $U$.

To emulate the presence of unobserved confounding, we filter the experimental data with selection rules $f_X^{(Z)}$, $Z \in \{0, \ldots, 9\}$, following a procedure in [49]. More specifically, given a collection of IST samples $\{X^{(n)}, Y^{(n)}, U_2^{(n)}\}_{n=1}^N$ where $U_2^{(n)}$ is the age of the $n$th patient. For each data point $\left(X^{(n)}, Y^{(n)}, U_2^{(n)}\right)$, we introduce an instrumental variable $Z^{(n)} \in \{0, \ldots, 9\}$. Values of the instrument $Z^{(n)}$ for $n$th patient are decided by

$$Z^{(n)} = \lfloor 10 \times U_1 \rfloor, \text{ where } U_1^{(n)} \sim \mathtt{Unif}(0, 1). \tag{85}$$

We then check if $X^{(n)}$ satisfies the following condition

$$X^{(n)} = \lfloor 6 \times p_X \rfloor, \text{ where } p_X = \frac{1}{1 + \exp\left(-U_1^{(n)} \times U_2^{(n)}/100 - Z^{(n)}/10\right)} \tag{86}$$

If the above condition is satisfied, we keep the data point $\left(X^{(n)}, Y^{(n)}, Z^{(n)}, U_1^{(n)}, U_2^{(n)}\right)$ in the dataset; otherwise, the data point is dropped. After this data selection process is complete, we hide columns of variables $U_1^{(n)}, U_2^{(n)}$. Doing so allows us to obtain $N = 3 \times 10^3$ synthetic observational samples $\bar{V} = \left\{X^{(n)}, Y^{(n)}, Z^{(n)}\right\}_{n=1}^N$ that are compatible with the "Double bow" diagram of Fig. 1b. In this experiment, we set hyperparameters $\alpha_{U_1} = 10$ and $\alpha_{U_2} = 10$. As a baseline, we estimate the treatment effect $E[Y_{x=(1,0)}] = 1.3418$ for only assigning aspirin $X = (1, 0)$ from the randomized trial data containing $1.9285 \times 10^4$ subjects.

## C.1 Additional Simulations on Other Causal Diagrams

We also evaluate our algorithms on various simulated SCM instances in other more involved causal diagrams. Overall, we found that simulation results match the findings in the manuscript. For identifiable settings (Experiment 5), our algorithms are able to recover the actual, unknown counterfactual probabilities. For other more general cases where the target distribution is non-identifiable (Experiments 6, 7 and 8), our algorithms consistently dominate state-of-art bounding strategies.

**Experiment 5: Napkin Graph** This experiment evaluates our sampling algorithm on interventional probabilities that are identifiable from the observational data. In this case, the bounds over the target probability should collapse to a point estimate. Consider the "Napkin" graph in Fig. 10a where $X, Y, Z, W$ are binary variables in $\{0, 1\}$; $U_1, U_2, U_3$ take values in real $\mathbb{R}$. The identifiability of the interventional distribution $P(y_x)$ from the observational data $P(x, y, w, z)$ could be derived by iteratively applying inference rules of "do-calculus" [33, Thm. 4.3.1]. We collect $N = 10^4$ observational samples $\bar{V} = \{X^{(n)}, Y^{(n)}, Z^{(n)}, W^{(n)}\}_{n=1}^N$ from an SCM defined as follows:

$$U_1 \sim \mathcal{N}(0, 1), \quad U_2 \sim \mathcal{N}(0, 1), \quad U_3 \sim \mathcal{N}(0, 1)$$

$$W \sim \mathtt{Binomial}(1, p_W), \text{ where } p_W = \frac{1}{1 + \exp(U_1 - U_2)},$$

$$Z \sim \mathtt{Binomial}(1, p_Z), \text{ where } p_Z = \frac{1}{1 + \exp(W - U_3)},$$

$$X \sim \mathtt{Binomial}(1, p_X), \text{ where } p_X = \frac{1}{1 + \exp(-Z - U_1)}, \tag{87}$$

$$Y \sim \mathtt{Binomial}(1, p_Y), \text{ where } p_Y = \frac{1}{1 + \exp(X - U_2 - 0.5)}.$$

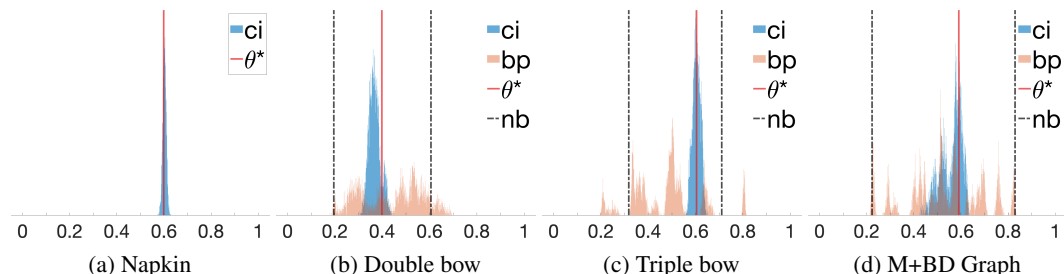

Figure 8: Histogram plots for samples drawn from the posterior distribution over target counterfactual probabilities. For all plots (a - d), *ci* represents our proposed algorithms; *bp* stands for Gibbs samplers using the representation of canonical partitions [2]; $\theta^*$ is the actual counterfactual probability; *nb* stands for the natural bounds [30].

In this experiment, we set hyperparameters $\alpha_{U_1} = d_{U_1} = 32$, $\alpha_{U_2} = d_{U_1} = 32$, and $\alpha_{U_3} = d_{U_3} = 4$. Fig. 8a shows a histogram containing samples drawn from the posterior distribution of $\left(P(Y_{x=0} = 1) \mid \bar{V}\right)$. Our analysis reveals that these samples converges to the actual interventional probability $P(Y_{x=0} = 1) = 0.6098$, which confirms the identifiability of $P(y_x)$ in the napkin graph.

**Experiment 6: Double Bow** This experiment evaluates our bounding strategy in non-identifiable settings where the optimal bounding strategy does not exist. In this case, our proposed algorithm should improve over state-of-art bounds. Consider again the "Double Bow" diagram in Fig. 1b where $X, Y, Z \in \{0, 1\}$ and $U_1, U_2 \in \mathbb{R}$. We collect $N = 10^4$ observational samples $\bar{V} = \{X^{(n)}, Y^{(n)}, Z^{(n)}\}_{n=1}^N$ from an SCM instance defined as follows:

$$U_1 \sim \mathcal{N}(0, 1), \quad U_2 \sim \mathcal{N}(0, 1),$$

$$Z \sim \text{Binomial}(1, p_Z), \text{ where } p_Z = \frac{1}{1 + \exp(-U_1)},$$

$$X \sim \text{Binomial}(1, p_X), \text{ where } p_X = \frac{1}{1 + \exp(-Z - U_1 - U_2)}, \quad (88)$$

$$Y \sim \text{Binomial}(1, p_Y), \text{ where } p_Y = \frac{1}{1 + \exp(X - U_2 + 0.5)}.$$

In this experiment, we set hyperparameters $\alpha_{U_1} = d_{U_1} = 32$ and $\alpha_{U_2} = d_{U_1} = 32$. Fig. 8b shows samples drawn from the posterior distribution of $\left(P(Y_{x=0} = 1) \mid \bar{V}\right)$. As a baseline, we also include the natural bounds [36, 30] (*nb*), and posterior samples obtained from the Gibbs sampler using a naïve generalization of the discretization procedure (*bp*) in [2]. Our analysis reveals that all algorithms achieve bounds that contain the actual, target causal effect $P(Y_{x=0} = 1) = 0.3954$. Our algorithm obtains a 100% credible interval $l_{ci} = 0.3054, r_{ci} = 0.4456$, which dominates all the other algorithms ($l_{bp} = 0.1778, r_{bp} = 0.6923, l_{nb} = 0.1949, r_{nb} = 0.6061$).

**Experiment 7: Triple Bow** Consider the "Triple Bow" diagram in Fig. 10b where $X, Y, Z \in \{0, 1\}$ and $U_1, U_2, U_3 \in \mathbb{R}$. We collect $N = 10^4$ observational samples $\bar{V} = \{X^{(n)}, Y^{(n)}, Z^{(n)}\}_{n=1}^N$ from an SCM defined as follows:

$$U_1 \sim \mathcal{N}(0, 1), \quad U_2 \sim \mathcal{N}(0, 1), \quad U_3 \sim \mathcal{N}(0, 1),$$

$$Z \sim \text{Binomial}(1, p_Z), \text{ where } p_Z = \frac{1}{1 + \exp(-U_1)},$$

$$W \sim \text{Binomial}(1, p_W), \text{ where } p_W = \frac{1}{1 + \exp(-Z - U_1 - U_2)}, \quad (89)$$

$$X \sim \text{Binomial}(1, p_X), \text{ where } p_X = \frac{1}{1 + \exp(-W - U_2 - U_3)},$$

$$Y \sim \text{Binomial}(1, p_Y), \text{ where } p_Y = \frac{1}{1 + \exp(X - U_3 - 0.5)}.$$

In this experiment, we set hyperparameters $\alpha_{U_1} = 0.001 \times d_{U_1} = 0.032$ and $\alpha_{U_2} = 0.001 \times d_{U_1} = 0.032$. Fig. 8c shows samples drawn from the posterior distribution of $\left(P(Y_{x=0} = 1) \mid \bar{V}\right)$. As a

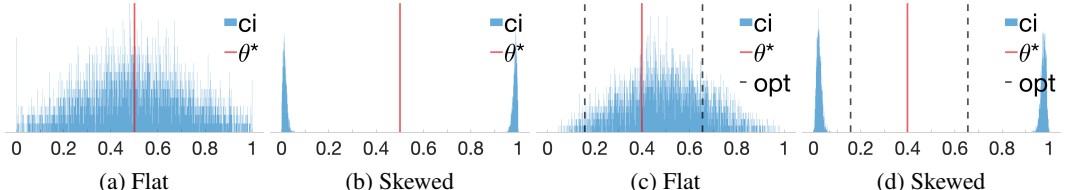

Figure 9: Prior distributions for (a, b) Experiment 9 and (c, d) Experiment 10.

baseline, we also include the natural bounds [36, 30] (*nb*), and posterior samples obtained from the Gibbs sampler using a naïve generalization of the discretization procedure (*bp*) in [2]. Our analysis reveals that while all algorithms achieve valid bounds ($l_{bp} = 0.1964, r_{bp} = 0.8148, l_{nb} = 0.3179, r_{nb} = 0.7105$), our algorithm obtains a $100\%$ credible interval $l_{ci} = 0.5608, r_{ci} = 0.6515$, which is the tightest bound over the target probability $P(Y_{x=0} = 1) = 0.6098$.

**Experiment 8: M+BD Graph**   Consider the "M+BD" graph in Fig. 10c where $X, Y, Z \in \{0, 1\}$ and $U_1, U_2 \in \mathbb{R}$. In this case, the counterfactual distribution $P(y_x)$ is non-identifiable due to the presence of the collider path $X \leftarrow U_1 \rightarrow Z \leftarrow U_2 \rightarrow Y$. We collect $N = 10^4$ observational samples $\bar{\boldsymbol{V}} = \{X^{(n)}, Y^{(n)}, Z^{(n)}\}_{n=1}^N$ from an SCM instance defined as follows:

$$
\begin{aligned}
U_1 &\sim \mathcal{N}(0, 1), \quad U_2 \sim \mathcal{N}(0, 1), \\
Z &\sim \texttt{Binomial}(1, p_Z), \text{ where } p_Z = \frac{1}{1 + \exp(-U_1)}, \\
X &\sim \texttt{Binomial}(1, p_X), \text{ where } p_X = \frac{1}{1 + \exp(-Z - U_1 - U_2)}, \\
Y &\sim \texttt{Binomial}(1, p_Y), \text{ where } p_Y = \frac{1}{1 + \exp(X - Z - U_2)}.
\end{aligned}
\tag{90}
$$

In this experiment, we set hyperparameters $\alpha_{U_1} = 0.01 \times d_{U_1} = 0.32$ and $\alpha_{U_2} = 0.01 \times d_{U_1} = 0.32$. Fig. 8d shows samples drawn from the posterior distribution of $\left(P(Y_{x=0} = 1) \mid \bar{\boldsymbol{V}}\right)$. As a baseline, we also include the natural bounds [36, 30] (*nb*), and posterior samples obtained from the Gibbs sampler using a naïve generalization of the discretization procedure (*bp*) in [2]. Our analysis reveals that all algorithms achieve bounds that contain the actual, target causal effect $P(Y_{x=0} = 1) = 0.5910$. Our algorithm obtains a $100\%$ credible interval $l_{ci} = 0.4247, r_{ci} = 0.6345$, which dominates all the other algorithms ($l_{bp} = 0.2140, r_{bp} = 0.8344, l_{nb} = 0.2230, r_{nb} = 0.8296$).

## C.2   The Effect of Sample Size and Prior Distributions

We will evaluate our algorithms using skewed prior distributions. We found that increasing the size of observational samples was able to wash away the bias introduced by prior distributions. That is, despite the influence of prior distributions, our algorithms eventually converge to sharp bounds over unknown counterfactual probabilities as the number of observational sample grows (to infinite).

**Experiment 9: Frontdoor**   Consider first the "Frontdoor" graph in Fig. 3 where the counterfactual distribution $P(y_x)$ is identifiable from the observational data $P(x, y, w)$. The detailed parametrization of the underlying SCM is described in Eq. (82). We present our results using two different priors. The first is a flat (uniform) distribution over probabilities of $U_1$ and $U_2$ respectively, i.e., $\alpha_{U_1} = d_{U_1} = 8$ and $\alpha_{U_1} = d_{U_2} = 4$. The second is skewed to present a strong preference on the deterministic relationships between $X$ and $Y$; in this case, $\alpha_1 = 300 \times d_{U_i}$, $i = 1, 2$, for prior distributions associated with both $U_1$ and $U_2$. Figs. 9a and 9b shows the distribution of $P(Y_{x=0})$ induced by these two priors (in the absence of any observational data). We see that the skewed prior of Fig. 9b assigns almost all weights to deterministic probabilities $P(Y_{x=0} = 1) = 1$ or $P(Y_{x=0} = 0) = 1$.

Fig. 10 shows posterior samples obtained by our Gibbs sampler when applied to observational data of various sizes, using both the flat prior (Figs. 10a to 10d) and the skewed prior (Figs. 10e to 10h). Both priors eventually collapse to the actual, unknown probability $P(Y_{x=0} = 1) = 0.5085$. As expected, more observational data are needed for the skewed prior before the posterior distribution converges, since the skewed prior is concentrated further away from the value $0.5085$ than the uniform prior.

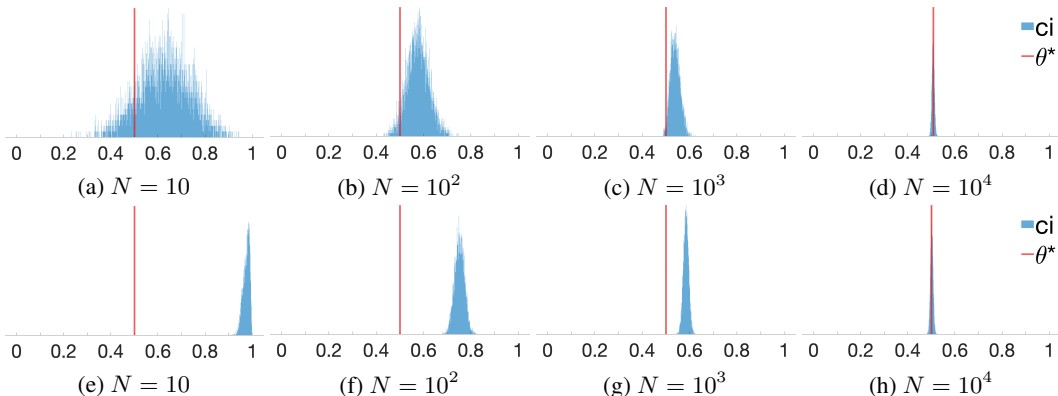

Figure 10: Histogram plots for samples drawn from the posterior distribution over probability $P(Y_{x=0} = 0)$ in "Frontdoor" graph of Fig. 3 using two priors. (a - d) shows the posteriors using the flat prior and observational data of size $N = 10, 10^2, 10^3$ and $10^4$ respectively; (e - h) shows the posetriors using the skewed prior and the same respective observational datasets.

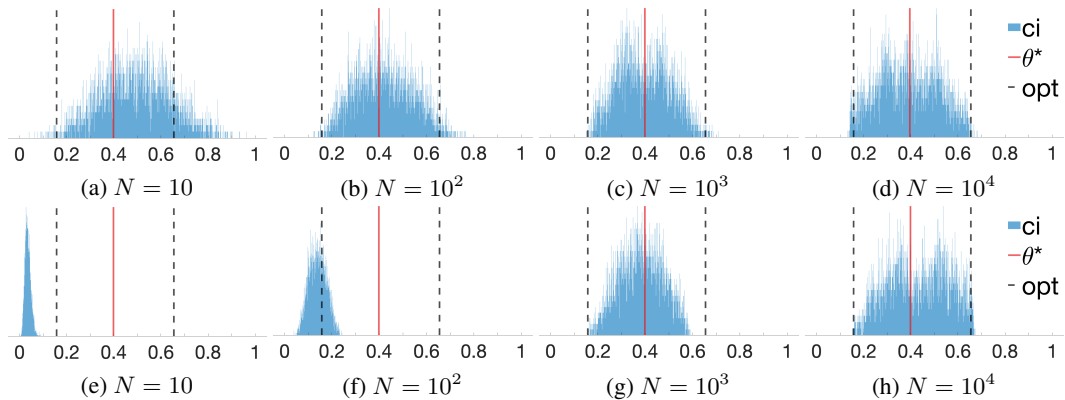

Figure 11: Histogram plots for samples drawn from the posterior distribution over probability $P(Y_{x=0} = 0)$ in "IV" graph of Fig. 1a using two priors. (a - d) shows the posteriors using the flat prior and observational data of size $N = 10, 10^2, 10^3$ and $10^4$ respectively; (e - h) shows the posetriors using the skewed prior and the same respective observational datasets.

**Experiment 10: IV** Consider the "IV" graph in Fig. 1b where $X, Y, Z$ are binary variables in $\{0, 1\}$. The detailed parametrization of the underlying SCM is described in Eq. (83). In this case, the counterfactual distribution $P(y_x)$ is not identifiable from the observational data $P(x, y, z)$ [5]. Sharp bounds over $P(y_x)$ from $P(x, y, z)$ were derived in [2] (labelled as *opt*). We present our results using two different priors. The first is a flat (uniform) distribution over probabilities of $U_1$ and $U_2$ respectively, i.e., $\alpha_{U_1} = d_{U_1} = 2$ and $\alpha_{U_1} = d_{U_2} = 16$. The second is skewed to present a strong preference on the deterministic relationships between $X$ and $Y$; in this case, $\alpha_1 = 300 \times d_{U_i}$, $i = 1, 2$, for prior distributions associated with both $U_1$ and $U_2$. Figs. 9c and 9d shows the distribution of $P(Y_{x=0})$ induced by these two prior distributions (in the absence of any observational data). We see that the skewed prior of Fig. 9d assigns almost all weights to deterministic probabilities $P(Y_{x=0} = 1) = 1$ or $P(Y_{x=0} = 0) = 1$.

Fig. 11 shows posterior samples obtained by our Gibbs sampler when applied to observational data of various sizes, using both the flat prior (Figs. 11a to 11d) and the skewed prior (Figs. 11e to 11h). Our analysis reveals that $100\%$ credible intervals of both priors eventually converge to the sharp IV bound $l = 0.1468, r = 0.6617$ over the unknown counterfactual probability $P(Y_{x=0} = 1) = 0.3954$. It is interesting to note that, in this experiment, while the choice of prior distribution does not influence the final counterfactual bound, it still has an effect on the shape of posterior distributions.

## D Naïve Generalization of (Balke and Pearl, 1995)

In this section, we will describe a naïve generalization of the canonical partitioning approach in [3] to the causal diagram of Fig. 1b. In particular, given any SCM $M$ compatible with Fig. 1b, we will construct a discrete SCM $N$ compatible with the diagram of Fig. 1c such that $M$ and $N$ coincide in all counterfactual distributions $\boldsymbol{P}^*$.

We first introduce some useful notations. Let $f_Z, f_X, f_Y$ denote functions associated with $Z, X, Y$ in SCM $M$. Let constants $h_Z^{(1)} = 0$ and $h_Z^{(2)} = 1$. Note that given any $U_1 = u_1$, $f_Z(u_1)$ must equate to a binary value in $\{0, 1\}$. We define a partition $\mathcal{U}_Z^{(i)}$, $i = 1, 2$, over domains of $U_1$ such that $u_1 \in \mathcal{U}_Z^{(i)}$ if and only if $f_Z(u_1) = h_Z^{(i)}$. Given any $u_1, u_2$, $f_X(\cdot, u_1, u_2)$ defines a function mapping from domains of $Z$ to $X$. Let functions in the hypothesis class $\Omega_Z \mapsto \Omega_X$ be ordered by

$$h_X^{(1)}(z) = 0, \qquad h_X^{(2)}(z) = z, \qquad h_X^{(3)}(z) = \neg z, \qquad h_X^{(4)}(z) = 1. \qquad (91)$$

Similarly, we define a partition $\mathcal{U}_X^{(i)}$, $i = 1, 2, 3, 4$ over the domain $\Omega_{U_1} \times \Omega_{U_2}$ such that $(u_1, u_2) \in \mathcal{U}_X^{(i)}$ if and only if the induced function $f_X(\cdot, u_1, u_2) = h_X^{(i)}$. Finally, let functions mapping from domains of $X$ to $Y$ be ordered by

$$h_Y^{(1)}(x) = 0, \qquad h_Y^{(2)}(x) = x, \qquad h_Y^{(3)}(x) = \neg x, \qquad h_Y^{(4)}(x) = 1. \qquad (92)$$

For any $u_2$, the induced function $f_Y(\cdot, u_2)$ must coincide with only of the above elements. Let $\mathcal{U}_Y^{(i)}$, $i = 1, 2, 3, 4$ be a partition over $\Omega_{U_2}$ such that $u_2 \in \mathcal{U}_Y^{(i)}$ if any only if $f_Y(\cdot, u_2) = h_Y^{(i)}$.

We now construct a discrete SCM $N$ compatible with the casual diagram of Fig. 1c. Let the exogenous variable $U$ in $N$ be a tuple $(U_Z, U_X, U_Y)$, where $U_Z \in \{1, 2\}$, $U_X \in \{1, 2, 3, 4\}$ and $U_Y \in \{1, 2, 3, 4\}$. For any $u_Z$, values of $Z$ are decided by $h_Z^{(u_Z)}$ where $h_Z^{(1)} = 0$, $h_Z^{(2)} = 1$. Given input $z, u_X$, values of $X$ are given by

$$x \leftarrow \xi_X^{(z, u_X)} = h_X^{(u_X)}(z), \qquad (93)$$

where $h_X^{(i)}(z)$, $i = 1, 2, 3, 4$, are functions defined in Eq. (91). Similarly, given input $x, u_Y$, values of $Y$ are given by

$$y \leftarrow \xi_Y^{(x, u_Y)} = h_Y^{(u_Y)}(x), \qquad (94)$$

where $h_Y^{(i)}(x)$, $i = 1, 2, 3, 4$, are functions defined in Eq. (92). Finally, we define the exogenous probability $P(u_Z, u_X, u_Y)$ in $N$ as the joint probability over partitions $\mathcal{U}_Z^{(i)}, \mathcal{U}_X^{(j)}, \mathcal{U}_Y^{(k)}$, $i = 1, 2$, $j = 1, 2, 3, 4$, $k = 1, 2, 3, 4$. That is,

$$P_N\left(U_Z = i, U_X = j, U_Y = k\right) = P_M\left((U_1, U_2) \in \mathcal{U}_Z^{(i)} \wedge \mathcal{U}_X^{(j)} \wedge \mathcal{U}_Y^{(k)}\right). \qquad (95)$$

It follows from the decomposition in Lem. 4 that $N$ and $M$ must coincide in all counterfactual distributions over binary $X, Y, Z$. The total cardinality of the exogenous domains in $N$ is $|\Omega_{U_Z} \times \Omega_{U_X} \times \Omega_{U_Y}| = 2 \times 4 \times 4 = 32$.

However, the construction for the reverse direction does not hold true. That is, given an arbitrary discrete $N$ compatible with the causal diagram in Fig. 1c, one could not construct an SCM $M$ compatible with the "Double bow" diagram in Fig. 1b such that $M$ and $N$ coincide in all counterfactual distributions. To witness, consider a discrete SCM $N$ where $P(U_Z = U_Y) = 1$, i.e., variables $U_Z$ and $U_Y$ are always the same, taking values in $\{1, 2\}$. Since in SCM $N$, values of $Z(u_Z)$ and $Y_{x=1}(u_Y)$ are given by

$$Z(u_Z) = h_Z^{(u_Z)} = 0 \times \mathbb{1}_{u_Z=1} + 1 \times \mathbb{1}_{u_Z=2},$$

$$Y_{x=1}(u_Y) = h_Y^{(u_Y)}(1) = 0 \times \mathbb{1}_{u_Y=1} + 1 \times \mathbb{1}_{u_Y=2}.$$

This means that counterfactual variables $Z$ and $Y_{x=0}$ must coincide, i.e., $P(Z = Y_{x=1}) = 1$. However, for any SCM $M$ compatible with Fig. 1b, counterfactual variables $Z$ and $Y_x$ must be independent due to the independence restriction [33, Ch. 7.3.2], which is a contradiction.

# E  Polynomial Optimization for Bounding Counterfactual Probabilities

In this section, we demonstrate how the optimization problem in Eq. (6) could be reduced to an equivalent polynomial program. The main challenge here is to write the counterfactual distribution $P(\boldsymbol{y_x}, \ldots, \boldsymbol{z_w})$ in discrete SCMs as a polynomial function of parameters $\xi_V^{(pa_V, u_V)}, \theta_u$. Since for binary $a, b \in \{0, 1\}$, $a \wedge b = ab$, this means that counterfactual distributions $P(\boldsymbol{y_x}, \ldots, \boldsymbol{z_w})$ in a discrete SCM could be written as:

$$P(\boldsymbol{y_x}, \ldots, \boldsymbol{z_w}) = \sum_{U \in \boldsymbol{U}} \sum_{u=1, \ldots, d_U} \mathbb{1}_{\boldsymbol{Y_x}(\boldsymbol{u})=y} \cdots \mathbb{1}_{\boldsymbol{Z_w}(\boldsymbol{u})=z} \prod_{U \in \boldsymbol{U}} \theta_u. \tag{96}$$

For convenience, we will represent parameters $\xi_V^{(pa_V, u_V)}$, for every $V \in \boldsymbol{V}$, any $pa_V, u_V$, as a binary sequence $\left\{ \xi_v^{(pa_V, u_V)} \mid \forall v \in \Omega_V \right\}$ such that $\xi_v^{(pa_V, u_V)} \in \{0, 1\}$ and $\sum_{v \in D_V} \xi_v^{(pa_V, u_V)} = 1$. The following proposition translates indicator functions of the form $\mathbb{1}_{\boldsymbol{Y_x}(\boldsymbol{u})=\boldsymbol{y}}$ into a polynomial function with regard to parameters $\xi_v^{(pa_V, u_V)}, \theta_u$.

**Lemma 8.** *For a discrete SCM $M = \langle \boldsymbol{V}, \boldsymbol{U}, \boldsymbol{F}, P \rangle$, for any $\boldsymbol{X}, \boldsymbol{Y} \subseteq \boldsymbol{V}$, fix $\boldsymbol{x}, \boldsymbol{y}, \boldsymbol{u}$. The indicator function $\mathbb{1}_{\boldsymbol{Y_x}(\boldsymbol{u})=\boldsymbol{y}}$ could be written as*

$$\mathbb{1}_{\boldsymbol{Y_x}(\boldsymbol{u})=\boldsymbol{y}} = \prod_{Y \in \boldsymbol{Y}} \mathbb{1}_{Y_x(\boldsymbol{u})=y}, \tag{97}$$

$$\text{where} \quad \mathbb{1}_{Y_x(\boldsymbol{u})=y} = \begin{cases} \mathbb{1}_{y=\boldsymbol{x}_Y} & \text{if } Y \in \boldsymbol{X} \\ \sum_{pa_Y} \xi_y^{(pa_Y, u_Y)} \mathbb{1}_{\{V_x(\boldsymbol{u}) \mid \forall V \in Pa_Y\}=pa_Y} & \text{otherwise} \end{cases} \tag{98}$$

*Proof.* By the basic property of indicator function, we must have, for any $\boldsymbol{Y}, \boldsymbol{X} \subseteq \boldsymbol{V}$,

$$\mathbb{1}_{\boldsymbol{Y_x}(\boldsymbol{u})=\boldsymbol{y}} = \prod_{Y \in \boldsymbol{Y}} \mathbb{1}_{Y_x(\boldsymbol{u})=y}. \tag{99}$$

Among quantities in the above equation, if $Y \subseteq \boldsymbol{X}$, $\mathbb{1}_{Y_x(\boldsymbol{u})=y}$ is equal to $\mathbb{1}_{\boldsymbol{x}_Y=y}$ where $\boldsymbol{x}_Y$ is the assignment to variable $Y$ in constants $\boldsymbol{x}$. Otherwise, for $Y \notin \boldsymbol{X}$, Eq. (4) implies

$$\mathbb{1}_{Y_x(\boldsymbol{u})=y} = \mathbb{1}_{\xi_Y^{(\{V_x(\boldsymbol{u}) \mid V \in Pa_Y\}, u_Y)}=y} \tag{100}$$

The indicator $\mathbb{1}_{Y_x(\boldsymbol{u})=y}$ could be further written as:

$$\mathbb{1}_{Y_x(\boldsymbol{u})=y} = \xi_y^{(\{V_x(\boldsymbol{u}) \mid V \in Pa_Y\}, u_Y)} = \sum_{pa_Y \in \Omega_{Pa_Y}} \xi_y^{(pa_Y, u_Y)} \mathbb{1}_{\{V_x(\boldsymbol{u}) \mid \forall V \in Pa_Y\}=pa_Y} \tag{101}$$

The last step follows from the fact that values of counterfactual variables $\{V_x(\boldsymbol{u}) \mid \forall V \in Pa_Y\}$ given $\boldsymbol{U} = \boldsymbol{u}$ must equate to an element in the domain $\Omega_{Pa_Y}$. $\qquad\square$

Recursively applying Lem. 8 to indicator functions $\mathbb{1}_{\boldsymbol{Y_x}(\boldsymbol{u})=\boldsymbol{y}}, \ldots, \mathbb{1}_{\boldsymbol{Z_w}(\boldsymbol{u})=\boldsymbol{z}}$ in Eq. (96) allows us to write any counterfactual distribution $P(\boldsymbol{y_x}, \ldots, \boldsymbol{z_w})$ as a polynomial function w.r.t. parameters $\theta_u, \xi_v^{(pa_V, u_V)}$. Therefore, the optimization problem in Eq. (6) is reducible to a series of polynomial programs which maximizes the objective $P(\boldsymbol{y_x}, \ldots, \boldsymbol{z_w})$ subject to the observational constraints in $P(\boldsymbol{v})$ and other basic parameter constraints over $\theta_u, \xi_v^{(pa_V, u_V)}$. We will illustrate our algorithm using various examples, summarized as follows.

**Example 1: Double Bow**  Consider again the "Double bow" diagram in Fig. 1b. We could derive a tight bound $[l, r]$ over the counterfactual probability $P(z, x_{z'}, y_{x'})$ from the observational distribution

$P(x, y, z)$ by solving the following polynomial program:

$$\min / \max \ P(z, x_{z'}, y_{x'}) = \sum_{u_1, u_2=1}^{d} \xi_z^{(u_1)} \xi_x^{(z', u_1, u_2)} \xi_y^{(x', u_2)} \theta_{u_1} \theta_{u_2}$$

$$\text{subject to} \ \ P(x, y, z) = \sum_{u_1, u_2=1}^{d} \xi_z^{(u_1)} \xi_x^{(z, u_1, u_2)} \xi_y^{(x, u_2)} \theta_{u_1} \theta_{u_2}$$

$$\forall z, u_1, \ \ \xi_z^{(u_1)} \left( 1 - \xi_z^{(u_1)} \right) = 0, \ \ \sum_z \xi_z^{(u_1)} = 1,$$

$$\forall x, z, u_1, u_2, \ \ \xi_x^{(z, u_1, u_2)} \left( 1 - \xi_x^{(z, u_1, u_2)} \right) = 0, \ \ \sum_x \xi_x^{(z, u_1, u_2)} = 1, \quad (102)$$

$$\forall y, x, u_2, \ \ \xi_y^{(x, u_2)} \left( 1 - \xi_y^{(x, u_2)} \right) = 0, \ \ \sum_y \xi_y^{(x, u_2)} = 1,$$

$$\forall u_1, \ \ 0 \leq \theta_{u_1} \leq 1, \ \ \sum_{u_1} \theta_{u_1} = 1,$$

$$\forall u_2, \ \ 0 \leq \theta_{u_2} \leq 1, \ \ \sum_{u_2} \theta_{u_2} = 1.$$

where the cardinality $d = |\Omega_Z| \times |\Omega_Z \mapsto \Omega_X| \times |\Omega_X \mapsto \Omega_Y|$.

**Example 2: IV** Consider the "IV" diagram in Fig. 1a. We could derive a tight bound $[l, r]$ over the counterfactual probability $P(y'_{x'}, x, y) \equiv P(Y_{x=x'} = y', X = x, Y = y)$ from the observational distribution $P(x, y, z)$ by solving the following polynomial program:

$$\min / \max \ P(y'_{x'}, x, y) = \sum_{u_1=1}^{d_1} \sum_{u_2=1}^{d_2} \xi_{y'}^{(x', u_2)} \xi_y^{(x, u_2)} \sum_z \xi_x^{(z, u_2)} \xi_z^{(u_1)} \theta_{u_1} \theta_{u_2}$$

$$\text{subject to} \ \ P(x, y, z) = \sum_{u_1=1}^{d_1} \sum_{u_2=1}^{d_2} \xi_z^{(u_1)} \xi_x^{(z, u_2)} \xi_y^{(x, u_2)} \theta_{u_1} \theta_{u_2}$$

$$\forall z, u_1, \ \ \xi_z^{(u_1)} \left( 1 - \xi_z^{(u_1)} \right) = 0, \ \ \sum_z \xi_z^{(u_1)} = 1,$$

$$\forall x, z, u_2, \ \ \xi_x^{(z, u_2)} \left( 1 - \xi_x^{(z, u_2)} \right) = 0, \ \ \sum_x \xi_x^{(z, u_2)} = 1, \quad (103)$$

$$\forall y, x, u_2, \ \ \xi_y^{(x, u_2)} \left( 1 - \xi_y^{(x, u_2)} \right) = 0, \ \ \sum_y \xi_y^{(x, u_2)} = 1,$$

$$\forall u_1, \ \ 0 \leq \theta_{u_1} \leq 1, \ \ \sum_{u_1} \theta_{u_1} = 1,$$

$$\forall u_2, \ \ 0 \leq \theta_{u_2} \leq 1, \ \ \sum_{u_2} \theta_{u_2} = 1.$$

where the cardinality $d_1 = |\Omega_Z|$ and $d_2 = |\Omega_Z \mapsto \Omega_X| \times |\Omega_X \mapsto \Omega_Y|$.

**Example 3: Bow** Consider the "Bow" diagram in Fig. 1d. We could derive a tight bound $[l, r]$ over the counterfactual probability $P(y_x, y'_{x'}) \equiv P(Y_x = y, Y_{x=x'} = y')$ from the observational

distribution $P(x, y)$ by solving the following polynomial program:

$$\min / \max \ P(y_x, y'_{x'}) = \sum_{u=1}^{d} \xi_y^{(x,u)} \xi_{y'}^{(x',u)} \theta_u$$

$$\text{subject to} \ \ P(x, y) = \sum_{u=1}^{d} \xi_x^{(u)} \xi_y^{(x,u)} \theta_u$$

$$\forall x, u, \ \ \xi_x^{(u)} \left( 1 - \xi_x^{(u)} \right) = 0, \ \ \sum_x \xi_x^{(u)} = 1, \tag{104}$$

$$\forall y, x, u, \ \ \xi_y^{(x,u)} \left( 1 - \xi_y^{(x,u)} \right) = 0, \ \ \sum_y \xi_y^{(x,u)} = 1,$$

$$\forall u, \ \ 0 \leq \theta_u \leq 1, \ \ \sum_u \theta_u = 1$$

where the cardinality $d = |\Omega_Z \mapsto \Omega_X|$.

**Example 4: Frontdoor** Consider the "Frontdoor" diagram in Fig. 3. We could derive a tight bound $[l, r]$ over the interventional probability $P(y_x)$ from the observational distribution $P(x, y, z)$ by solving the following polynomial program:

$$\min / \max \ P(y_x) = \sum_{u_1=1}^{d_1} \sum_{u_1=1}^{d_2} \sum_w \xi_y^{(w,u_1)} \xi_w^{(x,u_2)} \theta_{u_1} \theta_{u_2}$$

$$\text{subject to} \ \ P(x, y, w) = \sum_{u_1=1}^{d} \sum_{u_1=1}^{d_2} \sum_w \xi_x^{(u)} \xi_y^{(w,u_1)} \xi_w^{(x,u_2)} \theta_{u_1} \theta_{u_2}$$

$$\forall x, u_1, \ \ \xi_x^{(u)} \left( 1 - \xi_x^{(u)} \right) = 0, \ \ \sum_x \xi_x^{(u)} = 1,$$

$$\forall y, w, u_1, \ \ \xi_y^{(w,u_1)} \left( 1 - \xi_y^{(w,u_1)} \right) = 0, \ \ \sum_y \xi_y^{(w,u_1)} = 1, \tag{105}$$

$$\forall w, x, u_2, \ \ \xi_w^{(x,u_2)} \left( 1 - \xi_w^{(x,u_w)} \right) = 0, \ \ \sum_w \xi_w^{(x,u_w)} = 1,$$

$$\forall u_1, \ \ 0 \leq \theta_{u_1} \leq 1, \ \ \sum_{u_1} \theta_{u_1} = 1,$$

$$\forall u_2, \ \ 0 \leq \theta_{u_2} \leq 1, \ \ \sum_{u_2} \theta_{u_2} = 1.$$

where the cardinality $d_1 = |\Omega_X| \times |\Omega_W \mapsto \Omega_Y|$ and $d_2 = |\Omega_X \mapsto \Omega_W|$.

# F   Derivations of Complete Conditional Distributions

1017 In this section, we will provide detailed derivations for complete conditional distributions used in our
1018 proposed Gibbs samplers in Sec. 3.

1019 **Sampling $P\left(\bar{\boldsymbol{u}} \mid \bar{\boldsymbol{v}}, \boldsymbol{\theta}, \boldsymbol{\xi}\right)$.**   Variables $\boldsymbol{U}^{(n)}, \boldsymbol{V}^{(n)}$, $n = 1, \ldots, N$, are mutually independent given
1020 parameters $\boldsymbol{\theta}, \boldsymbol{\xi}$. This implies

$$P\left(\bar{\boldsymbol{u}} \mid \bar{\boldsymbol{v}}, \boldsymbol{\theta}, \boldsymbol{\xi}\right) = \prod_{U \in \boldsymbol{U}} P\left(\boldsymbol{u}^{(n)} \mid \bar{\boldsymbol{v}}, \boldsymbol{\theta}, \boldsymbol{\xi}\right) \tag{106}$$

$$= \prod_{U \in \boldsymbol{U}} P\left(\boldsymbol{u}^{(n)} \mid \boldsymbol{v}^{(n)}, \boldsymbol{\theta}, \boldsymbol{\xi}\right) \tag{107}$$

1021 The complete conditional for $\left(\boldsymbol{U}^{(n)} \mid \boldsymbol{V}^{(n)}, \boldsymbol{\theta}, \boldsymbol{\xi}\right)$, $n = 1, \ldots, N$, is given by

$$P\left(\boldsymbol{u}^{(n)} \mid \boldsymbol{v}^{(n)}, \boldsymbol{\theta}, \boldsymbol{\xi}\right) \propto P\left(\boldsymbol{u}^{(n)} \boldsymbol{v}^{(n)} \mid \boldsymbol{\theta}, \boldsymbol{\xi}\right) \tag{108}$$

$$\propto \prod_{V \in \boldsymbol{V}} P\left(v^{(n)} \mid pa_V^{(n)}, u_V^{(n)}, \boldsymbol{\theta}, \boldsymbol{\xi}\right) \prod_{U \in \boldsymbol{U}} P\left(u_V^{(n)} \mid \boldsymbol{\theta}, \boldsymbol{\xi}\right). \tag{109}$$

1022 Among quantities in the above equation, $P\left(u_V^{(n)} \mid \boldsymbol{\theta}, \boldsymbol{\xi}\right) = \theta_u$ for $u = u_V^{(n)}$; and

$$P\left(v^{(n)} \mid pa_V^{(n)}, u_V^{(n)}, \boldsymbol{\theta}, \boldsymbol{\xi}\right) = \mathbb{1}_{\xi_V^{\left(pa_V^{(n)}, u_V^{(n)}\right)} = v^{(n)}}. \tag{110}$$

1023 **Sampling $P\left(\boldsymbol{\xi}, \boldsymbol{\theta} \mid \bar{\boldsymbol{v}}, \bar{\boldsymbol{u}}\right)$.**   For every exogenous variable $U \in \boldsymbol{U}$, $\theta_U = \{\theta_u \mid \forall u\}$. For every
1024 endogenous variable $V \in \boldsymbol{V}$, $\xi_V = \left\{\xi_V^{(pa_V, u_V)} \mid \forall pa_V, u_V\right\}$. Since parameters $\boldsymbol{\xi}_V$, for every
1025 $V \in \boldsymbol{V}$, $\boldsymbol{\theta}_U$, for every $U \in \boldsymbol{U}$ are mutually independent, and they do not have common child nodes,
1026 we must have

$$P\left(\boldsymbol{\xi}, \boldsymbol{\theta} \mid \bar{\boldsymbol{v}}, \bar{\boldsymbol{u}}\right) = \prod_{V \in \boldsymbol{V}} P\left(\xi_V \mid \bar{\boldsymbol{v}}, \bar{\boldsymbol{u}}\right) \prod_{U \in \boldsymbol{U}} P\left(\theta_U \mid \bar{\boldsymbol{v}}, \bar{\boldsymbol{u}}\right). \tag{111}$$

1027 The above independence relationships imply that we could draw samples of posterior distributions
1028 over $\left(\xi_V \mid \bar{\boldsymbol{V}}, \bar{\boldsymbol{U}}\right)$ and $\left(\theta_U \mid \bar{\boldsymbol{V}}, \bar{\boldsymbol{U}}\right)$ for every $V \in \boldsymbol{V}, U \in \boldsymbol{U}$ separately.

1029 The complete conditional over $\left(\xi_V \mid \bar{\boldsymbol{V}}, \bar{\boldsymbol{U}}\right)$, defined in Eq. (10), follows from the fact that in discrete
1030 SCMs, the $n$th observation of variable $V \in \boldsymbol{V}$ is decided by $v^{(n)} \leftarrow \xi_V^{(pa_V, u_V)}$ given $pa_V^{(n)} = pa_V$,
1031 $u_V^{(n)} = u_V$. The complete conditional over $\left(\theta_U \mid \bar{\boldsymbol{V}}, \bar{\boldsymbol{U}}\right)$ in Eq. (11), follows from the conjugacy of
1032 the generalized Dirichlet distribution to multinomial sampling (e.g., see [22, Sec. 5.2]).

1033 **Sampling $P\left(\boldsymbol{u}^{(n)} \mid \bar{\boldsymbol{v}}, \bar{\boldsymbol{u}}_{-n}\right)$.**   At each iteration, draw $\boldsymbol{U}^{(n)}$ from the conditional given by

$$P\left(\boldsymbol{u}^{(n)} \mid \bar{\boldsymbol{v}}, \bar{\boldsymbol{u}}_{-n}\right) \propto \prod_{V \in \boldsymbol{V}} P\left(v^{(n)} \mid pa_V^{(n)}, u_V^{(n)}, \bar{\boldsymbol{v}}_{-n}, \bar{\boldsymbol{u}}_{-n}\right) \prod_{U \in \boldsymbol{U}} P\left(u^{(n)} \mid \bar{\boldsymbol{v}}_{-n}, \bar{\boldsymbol{u}}_{-n}\right). \tag{112}$$

1034 Among quantities in the above equation, for every $V \in \boldsymbol{V}$,

$$P\left(v^{(n)} \mid pa_V^{(n)}, u_V^{(n)}, \bar{\boldsymbol{v}}_{-n}, \bar{\boldsymbol{u}}_{-n}\right)$$

$$= \sum_{\xi_V^{\left(pa_V^{(n)}, u_V^{(n)}\right)} \in \Omega_V} \mathbb{1}_{\xi_V^{\left(pa_V^{(n)}, u_V^{(n)}\right)} = v^{(n)}} P\left(\xi_V^{\left(pa_V^{(n)}, u_V^{(n)}\right)} \mid \bar{\boldsymbol{v}}_{-n}, \bar{\boldsymbol{u}}_{-n}\right). \tag{113}$$

1035 The complete conditional distribution over $\left(\xi_V^{(pa_V, u_V)} \mid \bar{\boldsymbol{V}}_{-n}, \bar{\boldsymbol{V}}_{-n}\right)$, $\forall pa_V, u_V$, follows from the
1036 definition of discrete SCMs, i.e., the $n$th observation of variable $V \in \boldsymbol{V}$ is decided by $v^{(n)} \leftarrow$
1037 $\xi_V^{(pa_V, u_V)}$ given $pa_V^{(n)} = pa_V$, $u_V^{(n)} = u_V$. Formally,

$$P\left(\xi_V^{(pa_V, u_V)} \mid \bar{\boldsymbol{V}}_{-n}, \bar{\boldsymbol{V}}_{-n}\right) = \begin{cases} \mathbb{1}_{\xi_V^{(pa_V, u_V)} = v^{(i)}} & \text{if } \exists i \neq n, \, pa_V^{(i)} = pa_V, u_V^{(i)} = u_V, \\ 1/|\Omega_V| & \text{otherwise.} \end{cases} \tag{114}$$

1038  Marginalizing over the domain $\Omega_V$ in Eq. (113) gives the complete conditional in Eq. (13). For every
1039  $U \in \boldsymbol{U}$, the complete conditional of $P\left(u^{(n)} \mid \bar{\boldsymbol{v}}_{-n}, \bar{\boldsymbol{u}}_{-n}\right)$, defined in Eq. (14), follows from the
1040  Pólya urn characterization of generalized Dirichlet distributions (e.g., see [22, Sec. 4]).