# OpenReview forum: "Partial Identification of Counterfactual Distributions"
_NeurIPS.cc/2021/Conference — NeurIPS 2021 Submitted_

### Official Review · Reviewer_bpXs · 2021-07-12

**Rating:** 5
**Confidence:** 5

**Summary:**

EDIT:  I thank the authors for their clarifications and corrections to my previous line of thought, and I am prepared to revise my score up to 5 as a result.  I take the point about polynomial vs linear programming, and I can see that Lemma 4 is a non-trivial result.  The authors, however, make no effort to point to Lemma 4 as being the primary source of the novelty in Theorem 1!

Overall, the content of the paper seems too rich for a conference with a page limit like this one, it might be better off submitted to a journal with much more explanation and more examples.

----

This paper gives a Monte Carlo method for obtaining bounds from any DAG model with hidden variables.   The authors illustrate the algorithm on the front door model, as well as the instrumental variables model; they give several more examples in the supplementary material.

I think the paper could be improved by explaining more explicitly how the Bayesian algorithm for obtaining samples helps, when presumably one could solve the linear program (6).

**Limitations And Societal Impact:**

I'm not sure what to write here!  The paper seems fine other than the remarks I have made above.

**Main Review:**

### Main Points

1. I do not think that Theorem 1 is quite as much of a contribution as the authors claim.  First, note that Rosset et al. do not explicitly exclude edges within the graph, and I don't think their result is restricted in the way that the authors here imply.  Secondly there is an unpublished manuscript of Sainz and Wolfe that shows it is sufficient to consider graphs where the random vertices (i.e. those inside a given C-component) do not have children (it is referenced in Navascués and Wolfe, 2020).

    The addition of counterfactually equivalent distributions seems like only a small technical tweak.  I also note that the proof is now essentially identical to that of Rosset.

2.  The MCMC algorithm seems quite good, though it is never tested on a really high dimensional example.  In fact the only example given seems to be the instrumental variables model.  In addition, does the tighter credible interval you obtain simply reflect a more concentrated prior distribution?

3.  I note that there is no comparison with Richardson et al (2011), who do something very similar.

4.  I am curious about the authors' statement that the Balke-Pearl bounds do not apply to Figure 1(b).  Can you elaborate?

### Minor Points

 - l14: 'finite number of observational data' $\rightarrow$ 'finite quantity of observational data'?  Or 'observational data points'?

 - l41: 'as the principal stratification'- should this just be 'as principal stratification'?  (I am genuinely not sure.)

 - l345: '100% credible interval' - should this be '95%'?

 - l354: 'counterfactual equivalent' $\rightarrow$ 'counterfactually equivalent'

### References

Navascués and Wolfe. The inflation technique completely solves the causal compatibility problem. _Journal of Causal Inference_ 8(1)70-91, 2020.

Richardson, Evans and Robins.  Transparent parameterizations of models for potential outcomes. _Bayesian Statistics 9_, 2011.

**Time Spent Reviewing:**

2 hours

---

> ### Author Response · Authors · 2021-08-10
> **Response to Reviewer bpXs**
>
> We thank the reviewer for the feedback. We believe that a few misreadings of our work made some of the evaluations overly harsh, and respectfully ask the reviewer to reconsider our paper in the light of clarifications provided below.
>
> ---
> > Q0. _"I think the paper could be improved by explaining more explicitly how the Bayesian algorithm for obtaining samples helps, when presumably one could solve the linear program (6)."_
>
> This statement that “presumably one could solve the linear program (6)” is factually wrong. The optimization problem in Eq. (6) is generally equivalent to a polynomial program, instead of a linear program. As we mentioned in Line 193-195: "The optimization problem of Eq. (6) is reducible to equivalent polynomial programs (see Appendix E). Despite the soundness and tightness of derived bounds, solving such programs may take exponentially long in the most general case." For example, we applied a state-of-art polynomial program solver, gloptipoly3 (Henrion and Lasserre 2003), to the "Double bow" diagram in Fig. 1(b) with binary $X, Y, Z$. The program ran out of memory on a 32GB computer and was unable to find the exact solution for a model with 3 binary endogenous and 2 exogenous variables. To address this computational challenge, we develop Bayesian algorithms to approximate the optimal bounds.
>
> ---
> > Q1a. _“I do not think that Theorem 1 is quite as much of a contribution as the authors claim. First, note that Rosset et al. do not explicitly exclude edges within the graph, and I don't think their result is restricted in the way that the authors here imply. Secondly there is an unpublished manuscript of Sainz and Wolfe that shows it is sufficient to consider graphs where the random vertices (i.e. those inside a given C-component) do not have children (it is referenced in Navascués and Wolfe, 2020). "_
>
> We are unaware of results introduced in "an unpublished manuscript of Sainz and Wolfe". We also checked (Navascués and Wolfe, 2020) and could not find any reference to (Sainz and Wolfe), nor could Google search find any relevant manuscript by (Sainz and Wolfe). It would be appreciated if the reviewer could provide the exact reference, including the title of the manuscript.
>
> It is possible that the reviewer might be referring to the "unpacking" procedure (Navascués and Wolfe, 2020). This procedure could be seen as recursively applying *composition* axiom [33] (Pearl, 2000, Chap. 7.3.1), which translates the observational distribution $P(\boldsymbol{v})$ into a counterfactual distribution. If this is the case, we will remove "When there is no direct arrow between endogenous variables" in the future draft. Still, this procedure is only applicable to the observational distribution and does not extend to an arbitrary counterfactual distribution, as explained in the following.
>
> ---
> > Q1b. _"The addition of counterfactually equivalent distributions seems like only a small technical tweak. I also note that the proof is now essentially identical to that of Rosset."_
>
> We respectfully disagree with the statement that Theorem 1 "seems like only a small technical tweak" and its proof "is essentially identical to" (Rosset et al., 2018). Theorem 1 generalizes (Rosset et al., 2018) in fundamental ways, including some critical assumptions.
>
> First, we note (Rosset et al., 2018) focus on representing one observational distribution, and causality is neither considered nor invoked. The generalization to counterfactual distributions is not immediate or obvious at all. For instance, when considering an arbitrary counterfactual distribution $P(y_{x}, \dots, z_w)$, the unpacking procedure in (Eq. 37, Navascués and Wolfe, 2020) is not applicable. To prove Theorem 1, a critical step is to show that any counterfactual distribution $P(y_{x}, \dots, z_w)$ could be written as a function over joint probabilities over canonical partitions (Def. 6). Proving this result is non-trivial and is shown in Lem. 4, Appendix A.2.
>
> Second, (Rosset et al., 2018) rely on the critical assumption that probability density functions $p(u)$ over every exogenous variable $U$ are well-defined, which is not always realizable, e.g., the Cantor distribution. When density functions $p(u)$ exist, one could discretize the exogenous domains by repeatedly applying Caratheodory’s theorem. However, this is no longer case when considering the more general setting where each $U$ is associated with a probability space $\langle \Omega, F, P \rangle$. To address this challenge, we show in Lem. 3 that each canonical partition could be decomposed into a countable set of cells (Def. 8). Such decomposition allows one to translate any SCM $M$ into an equivalent discrete SCM $N$ in terms of counterfactual probabilities and causal structures (Lems. 5 & 6). The construction of such discrete SCMs is non-trivial and is arguably the most critical insight behind Theorem 1. Finally, one could bound the cardinality of the discrete SCM $N$ using Caratheodory’s theorem.
>
> ---
> > Q2a. _"The MCMC algorithm seems quite good, though it is never tested on a really high dimensional example. In fact the only example given seems to be the instrumental variables model.”_
>
> In addition to the IV model in Fig. 1(a) (Experiment 2), we tested on the “Bow” diagram in Fig. 1d (Experiment 3), and the "Double bow" in Fig. 1(b) (Experiment 4), where the cardinality of $X$ is $6$, $Y$ is $4$, and $Z$ is $10$. We also refer readers to Appendix C for additional experiments (mentioned in Line 296). We do not claim that our algorithm completely solves challenges posed by really high-dimensional data but it provides a practical solution for settings that weren’t solvable before (i.e., any causal diagram and any counterfactual distribution with the exception of the IV diagram which has been fully solved). Simulation results show that our algorithm consistently dominates existing bounding strategies in various types of causal diagrams with regular endogenous domains. We believe this result is highly non-trivial and could have a significant impact on the practice of causal inference.
>
> ---
> > Q2b. _“In addition, does the tighter credible interval you obtain simply reflect a more concentrated prior distribution?"_
>
> The tighter credible interval **DOES NOT** reflect a more concentrated prior distribution. For instance, see Fig. 8(b) in Appendix C, all Bayesian solvers utilize **uniformative priors** (i.e., flat) over parameters. Our algorithm (ci) still obtains tighter bounds compared to state-of-art methods. We also include experiments in Appendix C.2 comparing credible intervals obtained using an uninformative prior (flat) and a skewed prior (concentrated). Simulation results (Figs. 10 & 11) show that both priors eventually converge to the optimal bound. That is, the choice of prior distribution does not influence the final counterfactual bound.
>
> ---
> > Q3. _"I note that there is no comparison with Richardson et al (2011), who do something very similar."_
>
> (Richardson et al. 2011) presents a reparametrization of the canonical partitions of [3] (Balke & Pearl, 1995), applicable for the IV diagram in Fig. 1(a), but were not about deriving any new bounds. We already include the asymptotic optimal bound (opt) for the IV case introduced in [4] (Balke & Pearl, 1997). The simulation results show that our credible intervals (ci) match the optimal bounds. Again, the cited reference has a different goal of presenting a more “transparent” re-parametrizations of the canonical partitions in IV models that separate the identified and non-identified parts of the parameter.
>
> ---
> > Q4. _"I am curious about the authors' statement that the Balke-Pearl bounds do not apply to Figure 1(b). Can you elaborate?"_
>
> Thank you for the opportunity to clarify the issue. The Balke-Pearl bounds rely on a discretization procedure designed only for the IV diagram in Fig. 1(a). The "Double bow" diagram in Fig. 1(b) differs from Fig. 1(a) in that there now exists an unobserved confounder $U_1$ between $Z$ and $X$. This difference, albeit apparently small, has significant implications on the discretization procedure because now the Balke-Pearl canonical partition of $(U_1, U_2)$ based on $f_X$ needs to match with the canonical partition of $U_1$ based on $f_Z$ and that of $U_2$ based on $f_Y$ (see discussion in Appendix D, page 30). As mentioned in Line 145-156:
>
> >"One could naively apply the discretization procedure in (Balke & Pearl, 1995) and obtain a family of discrete SCMs that are sufficient in representing distributions in a causal diagram. However, such parametrization is not necessarily complete. To witness, consider again the causal diagram in Fig. 1(b) with binary $X, Y, Z$. Applying the discretization in (Balke & Pearl, 1995)  leads to a family of discrete SCMs compatible with a different diagram in Fig. 1(c), where the cardinality of exogenous variable $U$ is equal to $d = 32$ (see Appendix D for details, Line 966). However, this parametrization fails to capture some critical constraints over counterfactual distributions since it does **not** maintain the original structure of the causal diagram. For instance, counterfactual variables $Z$ and $Y_x$ in the original diagram of Fig. 1(b) are independent due to independence restrictions; while $Z$ and $Y_x$ in Fig. 1(c) are generally correlated due to the presence of unobserved confounder $U$."
>
> After all, we hope that the answers provided above help to clarify the issues raised, but we would be happy to provide further elaboration in case some point is still not clear.

---

> > ### Comment · Reviewer_bpXs · 2021-08-11
> > **I meant the bounds of Balke and Pearl (1997)**
> >
> > >> Q4. "I am curious about the authors' statement that the Balke-Pearl bounds do not apply to Figure 1(b). Can you elaborate?"
> >
> > > Thank you for the opportunity to clarify the issue. The Balke-Pearl bounds rely...
> > >> "One could naively apply the discretization procedure in (Balke & Pearl, 1995) and obtain a family of discrete SCMs...
> >
> > The bounds I was referring to are derived in the 1997 paper by Balke and Pearl (your reference [4]).  Can you explain why these bounds are not the optimal ones for the double bow model?  Or am I missing something more subtle here?

---

> > > ### Author Response · Authors · 2021-08-12
> > > **Applying (Balke & Pearl, 1997) bounds to “Double-bow”**
> > >
> > > > _”The bounds I was referring to are derived in the 1997 paper by Balke and Pearl (your reference [4]). Can you explain why these bounds are not the optimal ones for the double bow model? Or am I missing something more subtle here?”_
> > >
> > > Thank you for the opportunity to clarify this issue. Balke & Pearl’s (1997) bounds were derived for the IV diagram in Fig. 1(a), which encodes the following critical assumptions (p. 2 of the cited reference). (1) variable $Z$ does not affect outcome $Y$ directly, but only through treatment $X$; (2) $Z$ is independent of all latent factors $U$ that could affect $X$ and $Y$. However, in the "Double bow" diagram (Fig. 1(b)), Assumption 2 no longer holds: variable $Z$ is now generally correlated with latent factors $U_1, U_2$ affecting $X$ and $Y$. In other words, Balke & Pearl's (1997) bounds are not applicable to the "Double bow" diagram or other diagrams, which means blindly applying them to other diagrams may generate invalid answers.
> > >
> > > More specifically, Balke-Pearl’s (1997) bounds were derived based on a discretization of unobserved confounders technique. The goal of this work is to extend the discretization technique to **any** causal diagrams to facilitate the derivation of bounds, which turned out to be quite challenging, even for the simple "Double bow" diagram in Fig. 1(b). In our previous response, we explained why it is non-trivial to extend the discretization technique. The particular bounds derived in Balke-Pearl (1997) are only theoretically sound for the IV diagram (Fig. 1(a)) but not for any other diagram.

---

> > > > ### Comment · Reviewer_bpXs · 2021-08-12
> > > > **Applying (Balke & Pearl, 1997) bounds to “Double-bow”**
> > > >
> > > > So, are you claiming that the bounds don't hold for the double bow example?  Because I'm pretty sure they do hold (and are tight).  The actual statement in Balke and Pearl (1997) is that, for the DAG $Z \to X \to Y \gets U_2 \to X$ (renaming their $U$ as $U_2$) we need:
> > > >  - $Z \perp Y \mid X, U_2$; and
> > > >  - $Z \perp U_2$.
> > > >
> > > > These conditions certainly both hold in the double bow.  Importantly, $Y_x$ is clearly independent of $Z$ (and $U_1$) in the double bow example as well.
> > > >
> > > > I'm also pretty sure that if you wrote down the linear program for your double bow then it would just amount to setting $U_1 = Z$ without loss of generality, and then having $U_2 = (X_z, Y_x)$ without loss of generality.  The _reason_ that $Z$ is correlated with $X$ is immaterial for this particular causal effect.
> > > >
> > > > Therefore I would be very confident that the Balke-Pearl bounds on $P(Y_x)$ also hold for the double bow example.

---

> > > > > ### Author Response · Authors · 2021-08-13
> > > > > **Regarding (Balke & Pearl, 1997)**
> > > > >
> > > > > We start by reiterating that the goal of our paper is to develop a general bounding strategy for **any** causal diagram -- beyond the “IV” diagram (Fig. 1(a)) and IV assumptions in Balke & Pearl's (1997) -- and for **any** counterfactual distribution -- beyond the effect of interventions, $P(y_x)$, and the effect of the treatment on the treated (ETT), $P(y_x|x’)$. To be clear, we stand by our statement, “Balke & Pearl's (1997) bounds are not applicable to the "Double bow" diagram or other diagrams, which means blindly applying them to other diagrams may generate invalid answers.” We provide below further elaboration to explain the possible subtlety of such a statement. For the sake of clarity of exposition, we will highlight the specific pair (causal diagram, query) we are referring to throughout the discussion whenever this is not clear.
> > > > >
> > > > > ---
> > > > > #### a. IV Assumptions for Bounding Interventional Distributions (i.e., (IV Graph, $P(y_x)$) + (Double Bow Graph, $P(y_x)$)
> > > > >
> > > > > BP-97 introduced new machinery for bounding the interventional distribution $P(y_x)$ when the endogenous variables are discrete under the assumptions of the canonical IV graph, as shown in Fig. 1(a). Further, as suggested by the reviewer, we can confirm that the same constraints are evoked when bounding $P(y_x)$ over binary domains in the double-bow diagram, which is shown in Fig. 1(b). These constraints will be called IV constraints and imply the bounds will coincide in the IV and Double-bow graphs.
> > > > >
> > > > > ---
> > > > > #### b. IV Assumptions for Bounding Other Counterfactual Distributions (Double-bow graph, $P(z, x_{z’}, y_{x’})$)
> > > > >
> > > > > To understand the subtlety, we note now that the bounds and linear programming (LP) formulation in BP-97 cannot in general be used to bound other counterfactual distributions even in the “double-bow” diagram. For instance, when bounding counterfactual distribution given in Eq. 5 in the paper, $P(z, x_{z’}, y_{x’})$, one could no longer obtain a LP reduction by setting $Z = U_1$, $U_2  = (X_{z=0}, X_{z=1}, Y_{x=0}, Y_{x=1})$, as suggested by the reviewer for the $P(Y_x)$ case. This would imply $P(x, y|z) = P(x, y|do(z))$, which does not hold in general in the “Double-bow” graph. The resulting optimization program is polynomial, i.e., see Eq. 102, Appendix E.
> > > > >
> > > > > ---
> > > > > #### c. (IV graph + $Z \leftarrow U_1 \rightarrow Y$, $P(y_x)$)
> > > > >
> > > > > To further understand the limitations of BP-97’s approach, we consider the same interventional query $P(Y_x)$ but with a slight modification of the IV graph.  Specifically, consider the generating model described by the SCM $M$ below compatible with the diagram $G$: $Z \rightarrow X \rightarrow Y$, $Z \leftarrow U_1 \rightarrow Y$, and $X \leftarrow U_2 \rightarrow Y$, which is the “IV” diagram (Fig. 1(a)) + $Z \leftarrow U_1 \rightarrow Y$. $U_1, U_2$ are ternary variables independently drawn from a distribution $P(u_i)$, $i = 1, 2$.
> > > > >
> > > > > |        | $U_i = 0$ | $U_i = 1$ | $U_i = 2$ |
> > > > > |--------|---------|---------|---------|
> > > > > | $P(U_i)$ | 0.3     | 0.3     | 0.4     |
> > > > >
> > > > >
> > > > > Values of $Z, X, Y$ are decided by functions $f_Z, f_X, f_Y$ given by, respectively,
> > > > >
> > > > > |          | $U_1 = 1$ | $U_1 = 1$ | $U_1 = 2$ |
> > > > > |----------|---------|---------|---------|
> > > > > | $f_Z(U_1)$ | 0       | 1       | 0       |
> > > > >
> > > > > |               | $U_2 = 0 $|  $U_2 = 1$ | $U_2 = 2 $|
> > > > > |---------------|---------|---------|---------|
> > > > > | $f_X(Z=0, U_2)$| 0       | 0       | 1       |
> > > > > | $f_X(Z=1, U_2)$ | 1       | 0       | 0       |
> > > > >
> > > > >
> > > > > |                             | $U_2 = 0$ | $U_2 = 1$ | $U_2 = 2$ |
> > > > > |-----------------------------|---------|---------|---------|
> > > > > | $f_Y(X = 0 \text{ or }1, U_1 = 0, U_2) $| 0       | 1       | 0       |
> > > > > | $f_Y(X = 0\text{ or }1, U_1 = 1, U_2) $| 1       | 0       | 0       |
> > > > > | $f_Y(X = 0\text{ or }1, U_1 = 2, U_2) $| 0       | 0       | 1       |
> > > > >
> > > > >
> > > > > In this SCM $M$, the treatment effect $E[Y|do(X=0)] = 0.34$.
> > > > >
> > > > > However, if apply the bounds in (Balke & Pearl, 1997), we obtain the following answer:
> > > > >
> > > > > |                 | Ground Truth | (Balke & Pearl, 1997) | Our bounds       |
> > > > > |-----------------|--------------|-----------------------|------------------|
> > > > > | $E[Y\|do(X = 0)] $| $0.34$         | $[0.1286, 0.3000] $    | $[0.1260, 0.4188] $|
> > > > >
> > > > >
> > > > > This means that Balke & Pearl’ (1997) bound does not contain the actual effect $E[Y|do(X = 0)]$, i.e., it is not a valid bound. We also show in the above table the credible intervals obtained using our algorithm, which turns out to be a valid answer (i.e., contains the ground truth effect), as expected.
> > > > >
> > > > > ---
> > > > > #### d. (IV +  $Z \leftarrow U_1 \rightarrow Y$, $P(z, x_{z’}, y_{x’})$)
> > > > >
> > > > > Now we consider the same SCM $M$ and graph $G$ but with a different counterfactual distribution $P(z, x_{z’}, y_{x’})$). If we apply the same program to bound the counterfactual probability $P(Z = 1, X_{z = 0}=1, Y_{x=0} = 0)$ by simply replacing the objective function, this gives the following answer:
> > > > >
> > > > > |                                      | Ground Truth | Our bounds       |
> > > > > |--------------------------------------|--------------|------------------|
> > > > > | $P(Z = 1, X_{z = 0}=1, Y_{x = 0} = 0) $| $0.12 $        | $[0.0171, 0.1465]$ |
> > > > >
> > > > > One reasonable question at this point is whether these bounds could be obtained by solving the optimization problem in Eq. (6), and an efficient linear program (LP) reduction might apply. However, as far as we are aware, there is no claim made in the literature about the complexity of computing such bounds or whether a simpler LP exists. Still, the independence restrictions among counterfactuals $Z \perp X_{z’}$ seems to be required in such computation, which in turn lead to polynomial constraints $P(Z, X_{z}) = P(Z)P(X_{z})$. Given this state of affairs, we tried to use a state-of-art polynomial program solver (Henrion and Lasserre 2003) to solve such an instance. As expected, unfortunately, this program ran out of memory on a 32GB computer and could not find an exact solution.
> > > > >
> > > > >
> > > > > ---
> > > > > #### Summary
> > > > > All in all, (Balke & Pearl, 1997) bounds were specifically defined for estimating the treatment effect $P(y_x)$ under IV assumptions, and Figs. 1(a) & (b) are two instances of such assumptions. Furthermore, under the same IV assumptions, Balke’s results also apply to the counterfactual distribution $P(y_x | x’)$. By and large, blindly applying (Balke & Pearl, 1997) to other pairs of diagrams and quantities could lead to invalid answers, as shown in the examples provided above. The goal of this paper is to develop a method to bound any general counterfactual probabilities in any causal diagram. We show this bounding problem is reducible to a polynomial programming problem that, in general, cannot be reduced to a LP. To circumvent the computational challenge presented by polynomial optimization, we develop a Bayesian approach (including algorithms) to approximate the optimal bounds. Empirical results show that our algorithm is able to derive novel counterfactual bounds in various causal diagrams, where traditional approaches do not apply.

---

> > > > > > ### Comment · Reviewer_bpXs · 2021-08-25
> > > > > > **Applying (Balke & Pearl, 1997) bounds to “Double-bow”**
> > > > > >
> > > > > > OK, that makes more sense.  The full explanation was unnecessary - I think you just need to rephrase that sentence so as not to specifically refer to a graph where the inequalities are (in fact) the same.

---

> > > > > > > ### Author Response · Authors · 2021-08-25
> > > > > > > **Applying (Balke & Pearl, 1997) bounds to “Double-bow”**
> > > > > > >
> > > > > > > Thank you for the feedback. We will incorporate your suggestions in the future manuscript.

---

> > ### Comment · Reviewer_bpXs · 2021-08-12
> > **'Interruption' Technique**
> >
> > > We are unaware of results introduced in "an unpublished manuscript of Sainz and Wolfe"....
> >
> > Sorry, I realize that it was in a [previous version](https://arxiv.org/pdf/1707.06476v1.pdf) of the arXiv manuscript.  The manuscript was apparently called "Interruption Technique for Causal Inference and the Quantum Instrumental Scenario".  The results are actually published, I now realize (see below).
> >
> > > It is possible that the reviewer might be referring to the "unpacking" procedure (Navascués and Wolfe, 2020).
> >
> > No, in fact I was referring to what Wolfe terms 'interruption'.  After a little more research, I found it explained in this paper [Quantum Inflation: A General Approach to Quantum Causal Compatibility](https://journals.aps.org/prx/pdf/10.1103/PhysRevX.11.021043) (see Section 5).

---

> > > ### Author Response · Authors · 2021-08-13
> > > **'Interruption' Technique and Theorem 1**
> > >
> > > >_”No, in fact I was referring to what Wolfe terms 'interruption'. After a little more research, I found it explained in this paper Quantum Inflation: A General Approach to Quantum Causal Compatibility (see Section 5).”_
> > >
> > > Thanks for providing the reference, which we read with interest. We will cite the new reference and remove "When there is no direct arrow between endogenous variables" before “[38] showed that the observational distribution in a diagram could be represented using finite-state exogenous variables.“ in the future draft. Based on our reading, the “interruption” technique allows one to translate an observational distribution of the form $P(Y = y | X = x, Z = z)$ into a specific counterfactual distribution $P(Y_{x, z} = y| X = x, Z_x = z)$. It could be seen as recursively applying the *composition* axiom [33] (Pearl, 2000, Sec. 7.3.1), which is one of the working horses of modern causal inference.
> > >
> > > Still, such a technique (or axiom) does not render the generalization to counterfactual distributions in Theorem 1 immediate or obvious. For instance, when considering a counterfactual distribution of the form $P(Y_{x} =y, Y_{w} = y’, Z_{x’} = z, Z_{w} = z’)$, the interruption technique (Eq. 13, Wolfe et al, 2021) is not applicable. On the other hand, a critical step to prove Theorem 1 in our paper is to show that any counterfactual distribution (e.g., $P(Y_{x} =y,Y_{w} = y’, Z_{x’} = z, Z_{w} = z’)$) could be written as a function over joint probabilities over canonical partitions (Def. 6). Proving this result in general is non-trivial and is shown in Lem. 4, Appendix A.2.

---

### Official Review · Reviewer_bRA7 · 2021-07-15

**Rating:** 6
**Confidence:** 3

**Summary:**

The paper presents a methodology to obtain bounds on counterfactual probabilities from unknown SCMs using observational data. The authors consider a setting where the causal structure (DAG) of the data-generating model is known and all the endogenous variables are discrete and finite.


In this setting, the authors show that the set of all SCMs compatible with the known DAG, G, and observational data can be mapped to a special class of SCMs, which they call discrete SCMs. These discrete SCMs have the nice property that they are compatible with DAG G and the observational distribution and all exogenous variables are discrete and finite (with known maximum value). The authors show that there must exist a discrete SCM within this family which agrees with the original (unknown) SCM on all counterfactual probabilities. Using this as the main idea, the authors attempt to bound the counterfactual probabilities by maximising/minimising the counterfactual probabilities over all discrete SCMs in this family.


The paper makes no assumptions about the missing exogenous variables in the SCM model under question. This makes for a very powerful idea which can be used to bound counterfactual queries.

**Limitations And Societal Impact:**

The authors have stated the assumptions clearly, however I believe more can be done to justify the Bayesian approach for obtaining Credible Intervals (see main review for more details).


I do not see any potential negative impacts of their work on the society.


**Main Review:**


The paper tackles the question of gaining insights into the counterfactual world using the factual observed data with missing confounders. This is a fundamentally challenging question to answer without making additional assumptions. The authors propose a methodology which does not make any assumptions about the missing (exogenous) variables, which makes this methodology quite appealing.


Setting:
However, the setting under consideration is very specific: the authors are only considering SCMs with discrete and finite endogenous variables. This rarely occurs in most real world problems. Additionally, the authors work on the assumption that the DAG of the data-generating model is known (including causal links between unobserved variables and observed variables). This may be a reasonable assumption in some settings, but depending on the problem at hand, we may not know the number of missing (exogenous) variables and how they causally affect endogenous variables in great granular detail. In such cases, we may have to resort to a very general DAG, which could potentially come at the cost of the tightness of the bounds on counterfactual probabilities.


Notation:
The work is generally high quality, however, in my opinion the notation gets very dense at times and could be further simplified. While the authors generally did a good job of defining the notation, it was sometimes difficult to get an intuitive feel of what a concept meant for a model under consideration. For example, it would have been nice to have a visual representation of c-components within a DAG, as it would allow readers to immediately see why they form partitions over both endogenous and exogenous variables. Moreover, the notation U_V is unclear as this is the first time the authors use endogenous variables as subscripts for exogenous variables. I believe either the definition of this notation is missing, or this is a typo.


Method:
With regards to the methodology itself, I think the general idea is very nice, and the authors have provided a reasonable theoretical backing to show that it works. However, one of the questions that has not been addressed theoretically in sufficient detail is: how good is the Bayesian approach in finding the family of discrete SCMs that agree with the observational distribution? While the authors show that the problem of finding the bounds can be reduced to a polynomial program in Appendix E, they do not use this approach to find the counterfactual bounds. I think it is very important to include a comparison of credible intervals obtained using Bayesian approach with the bounds obtained by solving these polynomial programs. This would be useful to gauge the reliability of credible intervals against the ground truth.


Experiments:
The experimental results generally show that the credible intervals obtained contain the ground truth, which is promising, and in the case where the counterfactual probability is identifiable, the credible interval is very tight. However, one thing which is concerning is that the observational data size used in first three experiments is very high (N=10^5) even though the endogenous variables are binary and models are very small (with at most 3 endogenous variables). I would like to see the effect of having smaller datasets on credible intervals. Additionally, the credible intervals in experiment 2 and 3 are very wide (and largely uninformative). In experiment 4, the authors have used age as a confounder, however, having looked at the IST details on https://pubmed.ncbi.nlm.nih.gov/9174558/, it seems that the trial is randomized ("Half the patients were allocated unfractionated heparin (5000 or 12,500 IU bd [twice daily]), and half were allocated "avoid heparin"…"). This suggests that age is not a confounder. Please clarify this if I am wrong and have overlooked something.


Overall, the idea is original and nice, but I believe more needs to be done to justify the use of this methodology in partial identification of counterfactual outcomes.


I have given it a rating of 6 because of some of the above questions. I will be happy to reconsider my rating if these questions are addressed.

**Time Spent Reviewing:**

4-5

---

> ### Author Response · Authors · 2021-08-10
> **Response to Reviewer bRA7**
>
> We thank the reviewer for the careful, thought, and constructive comments, and we tried our best to address them accordingly.
>
> ---
> > Q1. _“the setting under consideration is very specific: the authors are only considering SCMs with discrete and finite endogenous variables. This rarely occurs in most real world problems.”_
>
> We believe SCMs with discrete endogenous variables are an important class of models both theoretically and practically, and is the main class of models studied in Causality (Pearl, 2000), with a few exceptions (e.g., most of Ch. 5). Broadly speaking, categorical variables could exist in many practical applications, e.g., the IST dataset in experiments. Also, bounding continuous counterfactual probabilities from the observational data is generally infeasible. The target probability could take any real value. As a result, almost all bounding algorithms for counterfactuals assume endogenous domains to be finite. This work extends and generalizes the existing results to an arbitrary causal diagram.
>
> Note that before this work, the only well-studied bounding case is the IV model almost 25 years ago [3](Balke & Pearl, 1995). We hope that this paper is a step in the right direction towards more general partial identification results, which are in most need in the literature since precise identification is not always obtainable.
>
> Also, we agree with the reviewer that when the detailed causal diagram is unknown, "we may have to resort to a very general DAG," e.g., the Bow graph. In this case, the resulting counterfactual bound may not be tight, but it is still informative (i.e., smaller than the entire interval $[0, 1]$).
>
> ---
> > Q2. _"the notation $U_V$ is unclear"_
>
> This is defined In Line 78-79, values of $V$ are decided by a function "$v \gets f_V(pa_V, u_V)$" where $U_V$ is a subset of exogenous variables.
>
> ---
> > Q3. _"one of the questions that has not been addressed theoretically in sufficient detail is: How good is the Bayesian approach in finding the family of discrete SCMs that agree with the observational distribution? While the authors show that the problem of finding the bounds can be reduced to a polynomial program in Appendix E, they do not use this approach to find the counterfactual bounds. I think it is very important to include a comparison of credible intervals obtained using Bayesian approach with the bounds obtained by solving these polynomial programs. This would be useful to gauge the reliability of credible intervals against the ground truth."_
>
> In principle, our Bayesian approach converges to the optimal counterfactual bound, provided with uninformative priors and sufficient observational samples (Chickering & Pearl, 1997). It follows from a simple application of Bayes’ rule and the law of large numbers. For example, in the IV graph (Fig. 1(a)), the simulation result (Fig. 4(b)) shows that our algorithm (ci) achieves a credible interval that matches the asymptotic optimal bound (opt). The optimal bound is derived from the polynomial program in Eq. (6), which, in this case,  turns out to be a linear program and could be efficiently solved. Both the credible interval and the optimal bound contain the actual counterfactual probability $\theta^*$ (the ground truth), which supports the reliability of the Bayesian approach.
>
> In general solving polynomial programs in Appendix E may take exponentially long as mentioned in Line 193-195.  For example, we applied a state-of-art polynomial program solver, gloptipoly3 (Henrion and Lasserre 2003), to the "Double bow" diagram in Fig. 1(b) with binary $X, Y, Z$. The program ran out of memory on a 32GB computer and was unable to find the exact solution.
>
> ---
> > Q4. _"one thing which is concerning is that the observational data size used in first three experiments is very high ($N=10^5$) … I would like to see the effect of having smaller datasets on credible intervals."_
>
> In the first three experiments, we compare our algorithm with the optimal bounds already known from the literature. These bounds are computed from the observational distribution P(v), estimated using a large observational dataset. Still, our algorithm could converge using a much smaller observational dataset. For example, in Fig. 11 (a-d) in Appendix C.2 (p. 29), we show the bounds obtained by our Bayesian strategy with various sizes of observational data. Indeed, our algorithm efficiently approximates the asymptotic optimal bound using only $N = 10^2$ observational samples.
>
> ---
> > Q5. _"The credible intervals in experiment 2 and 3 are very wide (and largely uninformative)."_
>
> As shown in Figs. 4(b,c), credible intervals in Experiments 2 and 3 converge to the optimal counterfactual bound. Since the target counterfactual probability is not identifiable in these settings, it could take any value inside the bound. In other words, while credible intervals in Experiments 2 and 3 may appear uninformative, they are the best achievable estimation that one could obtain from the observational data without additional assumptions.
>
> ---
> > Q6. _“In experiment 4, the authors have used age as a confounder, however, having looked at the IST details..., it seems that the trial is randomized. ... This suggests that age is not a confounder. Please clarify this if I am wrong and have overlooked something.”_
>
> The details of the experiments could be found in Appendix C (p. 25). In summary, we simulate the unobserved confounding in IST data using a preprocessing procedure introduced in Sec. 5 of [23] (Kallus & Zhou, 2018). First, we filter the dataset using a selection function that determines a treatment $X = x'$ based on a patient's age. We compare value $x'$ with the actual treatment value $x$ that the patient received. If $x' = x$, we keep the sample in the dataset; otherwise, the sample is ignored. Finally, we remove the "age" column in the dataset and use the remaining samples as the confounded observational data. This followed a procedure in [49] (Zhang & Bareinboim, 2021), which has a more detailed explanation in its Appendix III.
>
> ---
> > Q7. _“more can be done to justify the Bayesian approach for obtaining Credible Intervals”_
>
> The main motivation for developing a Bayesian approach is that solving the exact polynomial programs introduced in Sec. 2.1 is in general computationally prohibitive as mentioned in Lines 193-195 (unless it’s a linear programming problem). The credible intervals based on the proposed Bayesian approach will converge to the optimal bounds in principle and were shown empirically to be effective (see Sec. 4 and Appendix C).

---

> > ### Comment · Reviewer_bRA7 · 2021-08-27
> > **Response**
> >
> > Thank you for addressing my queries.
> >
> > I think in general authors could make the notation more clear, and explain the many different components of this paper more clearly.
> >
> > Also, as the reviewer bpXs pointed out, the proof of proposition 2 in Rosset et al seems very similar to the proof of Theorem 1 in this paper. I think a reference to this proof should be included.

---

> ### Author Response · Authors · 2021-08-26
> **Follow-up with the rebuttal**
>
> Dear Reviewer bRA7,
>
> We appreciate your review and your positive assessment of our paper, which considered our ideas "original" and "nice".
>
> We are hopeful that you seem open to change your opinion and rating based on the discussion and new evidence. Since the reviewing period is coming to an end, and we feel we have fully addressed your concerns, we wonder whether we can provide any further clarification or additional information to help elucidate any point that may still be unclear.
>
> Thanks again for your time for providing good feedback to our work.
>
> Authors of paper #6136

---

### Official Review · Reviewer_BaLP · 2021-07-16

**Rating:** 6
**Confidence:** 3

**Summary:**

The main result of this paper is to establish that any SCM with discrete endogenous variables can be transformed into a counterfactually equivalent SCM where also the exogenous variables are discrete (finite domains). This result is used to find exact bounds on counterfactual probabilities based on the observational distribution, or, using an MCMC procedure, approximate bounds based on observational data.

**Limitations And Societal Impact:**

The exact method is computationally prohibitive, possibly to the point of not being practical, but this is not quantified (at least not in the main paper)

**Main Review:**

The main theorem is an impressive result that was apparently very challenging to prove. Knowing that infinite domains need not be considered for exogenous variables is of great theoretical interest, and could open the way to computational approaches for problems that were previously not accessible in such a way.  Two such new approaches are given in the paper, and the empirical results confirm their usefulness. But to me, Theorem 1 is the main contribution of this paper. I did not check the proof, though.

Minor comments:

Line 112: runaway "f"
Line 113-114: If I'm not mistaken, this condition follows from the general definitions of SCMs + the preceding condition + the assumption that endogenous variables have finite domains (which was stated earlier). It does apparently serve the role of introducing the $\xi$-notation, and maybe to put the finiteness assumption in a clearer place. But as it is, it looks like a fairly complicated condition when the actual message is much simpler.
Line 144: Why mention $2d=64$ here? Wouldn't $d^2$ be more relevant computationally?

UPDATE: I thank the authors for engaging in the discussion. Based on the other reviews and the discussion, I have revised my score to a 6: while there is a novel contribution in this paper, the result is in fact closer to existing results than I thought at first.

**Time Spent Reviewing:**

2

---

> ### Author Response · Authors · 2021-08-10
> **Response to Reviewer BaLP**
>
> We thank the reviewer for the constructive comments, and we have addressed them accordingly as follows.
>
> ---
> > Q1. _"Line 144: Why mention 2d=64 here? Wouldn't d^2 be more relevant computationally?"_
>
> Since U1 and U2 are independent, $P(u_1, u_2) = P(u_1)P(u_2)$; each $P(u_i)$ has $d$ parameters. The total number of parameters for $P(u_1, u_2)$ in this case is $2d=64$.
>
> ---
> > Q2. _“The exact method is computationally prohibitive, possibly to the point of not being practical, but this is not quantified“_
>
> We provided a qualitative assessment in Line 193-194, "The optimization problem of Eq. (6) is reducible to equivalent polynomial programs (see Appendix E). Despite the soundness and tightness of derived bounds, solving such programs may take exponentially long in the most general case." This means that the exact method is NP-hard in general. This is indeed our primary motivation to study Bayesian algorithms to approximate the optimal bounds in the second part of the paper.
>
> We also applied a state-of-art polynomial program solver, gloptipoly3 (Henrion and Lasserre 2003), to the "Double bow" diagram in Fig. 1(b) with binary $X, Y, Z$. The program ran out of memory on a 32GB computer and was unable to find the exact solution.

---

### Official Review · Reviewer_R5pL · 2021-07-17

**Rating:** 4
**Confidence:** 3

**Summary:**

Bounding unknown counterfactual probabilities using observational data and assumptions in a causal graph is the core problem addressed here. The main result is formally presented in theorem 1 which states that all counterfactual distributions in an arbitrary causal diagram could be generated using discrete exogenous variables taking values from a finite domain.


**Main Review:**

Too many results crammed in together makes the paper extremely hard to understand and follow. Too few examples and complicated notations make it harder. Broadly speaking, there are solutions to two problems that are jointly presented here, the first is identification and the second is estimation.  Also, in this work, proofs are the main contribution. Proofs are in the appendix and the appendix is filled with errors and typos.

An example for equation 2 where V is a small set V will be helpful. I found one in the appendix but there are issues with it that I detail below.

There is an ‘f’ at the end of line 112 on page 3.

(3) in definition 3: Please clarify what you mean. Is it the case that an U variable can only occur once in the path?

“SCM lives can be mapped… well-behaved space”: Try to avoid using such vague and ill-defined terms.

“Since G has a single c-component...family of discrete SCMs N”(page 4). It is not immediately clear why U1 and U2 must share the same cardinality.

Appendix A

Why is there a difference in the definitions of SCM in section 1.1 and definition 5?  h_{u_v} in definition 5 appears to be essentially the same as f_v in section 1.1.  Also in the first line of definition 5, you have M=<V,U,F,P>. ‘F’ is not mentioned anywhere in the rest of the definition. In line 557, the authors revert to the use of ‘f_v’.

“Since Y is not a descendant of X” (line 531): Are you still referring to Fig. 1d? If so, Y is a child of X and hence a descendant.

In 533, the notation pertaining to ‘h’ changed. I am assuming the ‘h’ referred here is the same as in line 3 of definition 5. What does the superscript denote? Also shouldn’t the subscript denote the exogenous variable that is a parent of X? The notation of h in line 540 makes better sense. This example described from lines 535-545 is crucial. It ought to be corrected and moved to the main paper.


There are similar issues throughout the paper. For the results to be useful to the community this paper needs to be split into two, carefully revised (rewritten with more examples and by including more content from the appendix) and reviewed again.


**Time Spent Reviewing:**

15

---

> ### Author Response · Authors · 2021-08-10
> **Response to Reviewer R5pL**
>
> We thank the reviewer for the thoughtful feedback. We believe that a few misreadings of our work made some of the evaluations overly harsh. We respectively ask the reviewer to reconsider our paper in the light of clarifications provided below.
>
> ---
> > Q1. _"There are solutions to two problems that are jointly presented here, the first is identification and the second is estimation. Also, in this work, proofs are the main contribution."_
>
> This paper extends existing works (Balke & Pearl, 1995; Chickering & Pearl 1997) on the **partial identification** of counterfactual probabilities to arbitrary causal diagrams. Similar to the previous works, our solution contains two parts: the first is the discretization of the exogenous domains, and the second is a novel bounding algorithm based on the discretization. Certainly, the proof of discretization procedure in Theorem 1 is an important contribution, but it only constitutes part of our solution. This paper also develops the first bounding algorithm that allows one to approximate optimal bounds over unknown counterfactuals from the observational data in **any** causal diagram. We empirically show that the proposed bounding strategy consistently dominates state-of-art algorithms. Overall, we believe the bounding algorithm is a significant contribution to the field. We would invite the reviewer to re-evaluate the rest of the manuscript to see whether our statements are factual.
>
> ---
> > Q2. _“Proofs are in the appendix and the appendix is filled with errors and typos.”_
>
> Overall, we believe typos are rare and we are confident there are no errors in the proof. We believe that a few misreadings of our work made the evaluation overly harsh. We provide clarification to specific concerns in the following.
>
> ---
> > Q3. _“For the results to be useful to the community this paper needs to be split into two, carefully revised (rewritten with more examples and by including more content from the appendix) and reviewed again.”_
>
> The paper addresses the problem of bounding counterfactual probabilities given observational data in a causal diagram. Our solution consists of two parts: (1) a reduction of the original SCM to a SCM with discrete exogenous domains (Theorem 1), and (2) an effective Monte Carlo algorithm to approximate the optimal bounds based on the discretization in Theorem 1. We believe the two parts together constitute an integral solution to partial identification of counterfactuals in arbitrary causal diagrams. This task is recognized as significant by the community and discussed at length in Ch. 8 of Causality (Pearl, 2000). We chose to put the whole proof of Theorem 1 to the Appendix with detailed explanation and examples because it is is non-trivial, involves complex mathematical constructs, and may hinder the flow of the main text. We will elaborate more on Theorem 1 by moving content from Appendix to the additional page if the paper is accepted.
>
>  ---
> > Q4. _"(3) in definition 3: Please clarify what you mean. Is it the case that an U variable can only occur once in the path?"_
>
> Yes, $V_i \leftrightarrow V_j$ is a shorthand for $V_i \leftarrow U_k \rightarrow V_j$ for some $U_k$. We consider simple paths where every node appears at most once. For instance, in Fig. 1(b), $Z \leftarrow U_1 \rightarrow X \leftarrow U_2 \rightarrow Y$ is a bi-directed path from $Z$ to $Y$.
>
> ---
> > Q5. _"Since G has a single c-component...family of discrete SCMs N”(page 4). It is not immediately clear why U1 and U2 must share the same cardinality."_
>
> This is an important technical point. By Theorem 1, the cardinality of every exogenous $U$ in the proposed family of discrete SCMs N is given by "the number of functions mapping from $Pa_V$ to $V$ for every variable $V$ in the c-component" associated with $U$. Since $U_1$ and $U_2$ correspond to the same c-component $\{Z, X, Y\}$ in Fig. 1(b), they must also share the same cardinality.
>
> ---
> > Q6. _"Why is there a difference in the definitions of SCM in section 1.1 and definition 5? h_{u_v} in definition 5 appears to be essentially the same as f_v in section 1.1. Also in the first line of definition 5, you have M=<V,U,F,P>. ‘F’ is not mentioned anywhere in the rest of the definition. In line 557, the authors revert to the use of ‘f_v’."_
>
> Definition 5 defines a discrete SCM based on the definition of SCM in Section 1.1. As mentioned in lines 518-519, Def. 5 is an "equivalent definition of discrete SCMs which will facilitate the understanding of the proof".
>
> The set $\boldsymbol{F}$ follows the general definition of SCMs, which is given in the main manuscript (Line 77-78): "$\boldsymbol{F}$ is a set of functions where each $f_V \in \boldsymbol{F}$ decides values of an endogenous variable $V \in \boldsymbol{V}$ taking as argument a combination of other variables in the system."
>
> In Line 557, we use $f_V$ since we are referring to a general SCM, instead of a discrete one. The goal of the proof is to construct a discrete SCM $N$ from a general SCM $M$ while preserving $M$'s counterfactual distributions and causal diagram (i.e., all the counterfactual constraints ascertained by the graph should be shared across $M$ and $N$).
>
> ---
> > Q7. _“Since Y is not a descendant of X” (line 531): Are you still referring to Fig. 1d? If so, Y is a child of X and hence a descendant."_
>
> This is a typo. It should be "$X$ is not a descendant of $Y$".
>
> ---
> > Q8. _"In 533, the notation pertaining to ‘h’ changed. I am assuming the ‘h’ referred here is the same as in line 3 of definition 5. What does the superscript denote?"_
>
> The notation of function $h_V^{(u_V)}$ in Line 533 is equivalent to the function $h_{u_V}$ in Def. 5. We appreciate suggestions on the notations and will incorporate in the next version of the manuscript.
>
> We hope things are more clear now but will be happy to provide additional clarifications and elaborations otherwise during the discussion stage.  Thank you.

---

> ### Author Response · Authors · 2021-08-26
> **Follow-up with the rebuttal**
>
> Dear Reviewer R5pL
>
> The discussion period is coming towards its end. We wonder whether you had the chance to check our rebuttal and see if it clarified the interesting issues raised in your review (as we hoped). Otherwise, we will be happy to follow up and provide further elaboration on unanswered concerns and burning questions.
>
> We certainly appreciate your time and attention. Thank you!
>
> Authors of paper #6136

---

### Decision · Program_Chairs · 2021-09-27

**Decision:**

Reject

**Comment:**

A summary from one of the reviews:

"This paper gives a Monte Carlo method for obtaining bounds from any DAG model with hidden variables. The authors illustrate the algorithm on the front door model, as well as the instrumental variables model; they give several more examples in the supplementary material."

The paper received extensive discussion, both with the authors, and among the reviewers.  This discussed alleviated some of the concerns reviewers had, which resulted in overall improved scores.  Nevertheless, concerns about an appropriate way of explicating the novelty of the contribution given constraints of conference proceedings, remained.

Theorem 1, prior work and overall length.  The author's manuscript, as currently written, made it very challenging to see the distinction in the proof strategy between their contribution (via Lemma 4 in the supplement), and related prior work.  While discussions with the authors clarified this to some extent, making this connection clearer would entail nontrivial changes.

In general, while many NeurIPS submissions have extensive appendices, at the conclusion of the discussion reviewers felt this paper in particular relied on results and narrative in the supplement in a way that made reading challenging without expanding the main body of the paper.  Thus, it was felt that the paper in its current was better suited for a journal.